# ALPHAFLOW: UNDERSTANDING AND IMPROVING MEANFLOW MODELS

**Huijie Zhang**[1,2,*]     **Aliaksandr Siarohin**[1]     **Willi Menapace**[1]

**Michael Vasilkovsky**[1]     **Sergey Tulyakov**[1]     **Qing Qu**[2]     **Ivan Skorokhodov**[1]

[1]Snap Inc.   [2]Department of EECS, University of Michigan

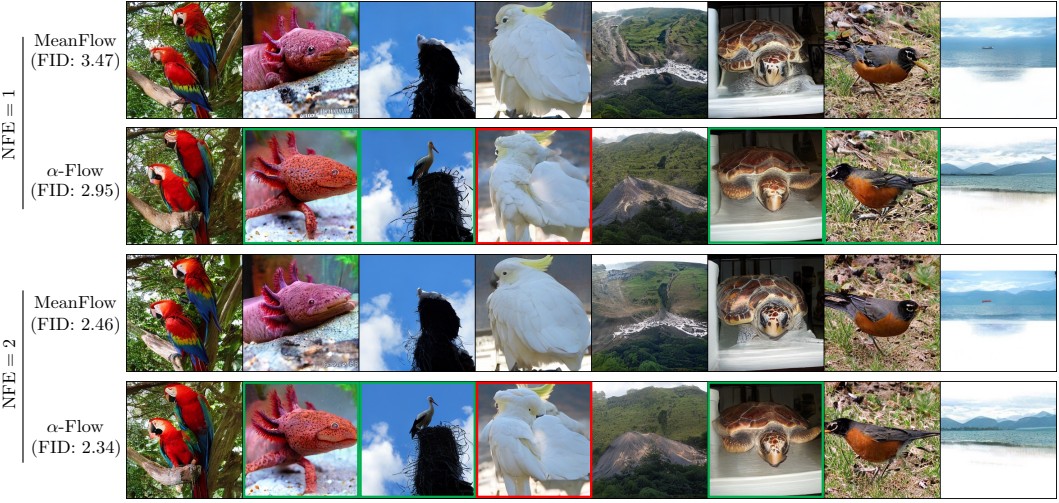

Figure 1: *Uncurated* samples (seeds 1-8) from the DiT-XL/2 model for MeanFlow Geng et al. (2025a) and $\alpha$-Flow (our proposed method) produced with 1 (upper) and 2 (lower) sampling steps for ImageNet-1K $256^2$.

## ABSTRACT

MeanFlow has recently emerged as a powerful framework for few-step generative modeling trained from scratch, but its success is not yet fully understood. In this work, we show that the MeanFlow objective naturally decomposes into two parts: trajectory flow matching and trajectory consistency. Through gradient analysis, we find that these terms are strongly negatively correlated, causing optimization conflict and slow convergence. Motivated by these insights, we introduce $\alpha$-Flow, a broad family of objectives that unifies trajectory flow matching, Shortcut Model, and MeanFlow under one formulation. By adopting a curriculum strategy that smoothly anneals from trajectory flow matching to MeanFlow, $\alpha$-Flow disentangles the conflicting objectives, and achieves better convergence. When trained from scratch on class-conditional ImageNet-1K 256×256 with vanilla DiT backbones, $\alpha$-Flow consistently outperforms Mean-Flow across scales and settings. Our largest $\alpha$-Flow-XL/2+ model achieves new state-of-the-art results using vanilla DiT backbones, with FID scores of 2.58 (1-NFE) and 2.15 (2-NFE). The source code and pre-trained checkpoints are available on `https://github.com/snap-research/alphaflow`.

## 1 INTRODUCTION

Diffusion models (Sohl-Dickstein et al., 2015) have emerged as the leading paradigm for generative modeling of visual data (Dhariwal & Nichol, 2021; Rombach et al., 2022; Brooks et al., 2024). However, their widespread use is limited by slow inference, as generating high-fidelity samples

---

[*]Work done during an internship at Snap Inc.

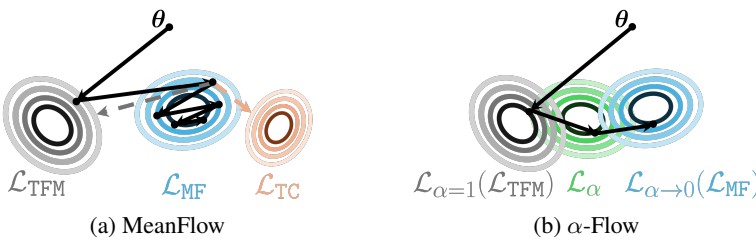

Figure 2: Diagrams of training dynamics between MeanFlow and $\alpha$-Flow. (a) MeanFlow training exhibits a gradient conflict. (b) $\alpha$-Flow resolves the conflict by first minimizing a trajectory flow matching loss, then following an annealing trajectory to approach the MeanFlow optimal solution.

typically requires a large number of denoising steps. This computational bottleneck has spurred extensive research into designing efficient diffusion-based generators that are able to operate in very few steps while preserving high generation quality (Salimans & Ho, 2022; Sauer et al., 2024; Song et al., 2023; Song & Dhariwal, 2024; Lu & Song, 2025; Geng et al., 2025b; Frans et al., 2025; Geng et al., 2025a).

Early attempts reduce the inference time of diffusion models through distilling a pre-trained multi-step model into a few-step one (Salimans & Ho, 2022; Sauer et al., 2024). The subsequent development of consistency models (Song et al., 2023; Song & Dhariwal, 2024; Lu & Song, 2025) enabled training from scratch for few-step generative models. However, a significant performance gap still remains between existing few-step and multi-step diffusion models. The recently introduced MeanFlow framework (Geng et al., 2025a) enables more stable training and better classifier-free guidance (Ho & Salimans, 2022) integration, significantly bridging the gap between few-step and multi-step from-scratch trained diffusion models. Despite its practical success, there still lacks a clear understanding of why MeanFlow performs better, which hinders further improvements and the design of stronger few-step models.

In this work, we provide a deeper understanding of why MeanFlow works, revealing that its training objective can be decomposed into two components: trajectory flow matching and trajectory consistency. Our gradient analysis shows that these two components are strongly negatively correlated during training, leading to instability and slow convergence in joint optimization. We further demonstrate that the previous heuristic adoption of border-case flow matching supervision is crucial: it actually acts as a surrogate loss for trajectory flow matching and mitigates gradient conflict. However, over 75% of MeanFlow's computation is spent on this border-case supervision, which is not its primary focus. This raises an open question: *can we design more efficient techniques to optimize MeanFlow objective, without such computational overhead?*

Motivated by these observations, we introduce $\alpha$-Flow, a new broad family of objectives for few-step flow models. This framework unifies trajectory flow matching, Shortcut Models Frans et al. (2025), and MeanFlow under a single unified formulation. As visualized in Figure 2, by employing a curriculum learning strategy that smoothly transitions from trajectory flow matching to MeanFlow, $\alpha$-Flow better disentangles the optimization of trajectory flow matching and trajectory consistency, reduces reliance on border-case flow matching supervision, and achieves better convergence.

By training vanilla DiT-(Peebles & Xie, 2023) models from scratch with $\alpha$-Flow on class-conditional ImageNet-1K $256^2$, we obtain consistently stronger performance across both small- and large-scale settings compared with MeanFlow, for both one-step and few-step generation. Our largest DiT-XL/2+ model establishes new state-of-the-art results among all from-scratch trained models with the vanilla DiT backbone and training pipeline, achieving FID scores of 2.58 (1-NFE) and 2.15 (2-NFE).

## 2 PRELIMINARIES

**Diffusion models and flow matching.** Diffusion model (Ho et al., 2020; Song & Ermon, 2019; Rombach et al., 2022) define a forward process that progressively adds noise to a data sample $x \sim$

$p_{\text{data}}(\boldsymbol{x})$ over a continuous timestep $t \in [0, 1]$. Specifically, given training data, the forward process perturbs $\boldsymbol{x}$ into a noisy version $\boldsymbol{z}_t = \beta_t \boldsymbol{x} + \sigma_t \boldsymbol{\epsilon}$ where $\boldsymbol{\epsilon} \sim \mathcal{N}(\mathbf{0}, \boldsymbol{I})$, $\beta_t$ and $\sigma_t$ are pre-defined scheduler parameters that depend on $t$, such that $\boldsymbol{z}_0 = \boldsymbol{x}$ and $\boldsymbol{z}_1 = \boldsymbol{\epsilon}$. Flow matching (Liu et al., 2023; Lipman et al., 2023) is a deterministic alternative that defines the forward process as a straight-line path between the noise distribution and the data distribution, setting $\beta_t = 1 - t$ and $\sigma_t = t$. A neural network $\boldsymbol{v_\theta}(\boldsymbol{z}_t, t)$ is trained to model the ground-truth vector field $\mathrm{d}\boldsymbol{z}_t/\mathrm{d}t$ along this trajectory $\boldsymbol{z}_t$ by minimizing the objective:

$$\mathcal{L}_{\text{FM}}(\boldsymbol{\theta}) = \mathbb{E}_{t, \boldsymbol{x}, \boldsymbol{z}_t}[||\boldsymbol{v_\theta}(\boldsymbol{z}_t, t) - \boldsymbol{v}_t||^2] \tag{1}$$

where $\boldsymbol{v}_t \triangleq \boldsymbol{v}(\boldsymbol{z}_t, t | \boldsymbol{x}) = \mathrm{d}\boldsymbol{z}_t / \mathrm{d}t \big|_{\boldsymbol{x}} = \boldsymbol{\epsilon} - \boldsymbol{x}$. To generate a new sample, the probability flow ODE (PF-ODE) $\mathrm{d}\boldsymbol{z}/\mathrm{d}t = \boldsymbol{v_\theta}(\boldsymbol{z}_t, t)$ is solved from $t = 1$ to $t = 0$, starting with an initial value $\boldsymbol{z}_1 \sim \mathcal{N}(\mathbf{0}, \boldsymbol{I})$.

One primary challenge of diffusion models is the slow sampling speed. To address this, several methods have been proposed to enable high-quality generation with significantly fewer steps.

**Consistency model (CM).** (Song et al., 2023) enables one-step generation by training a neural network $\boldsymbol{f_\theta}(\boldsymbol{z}_t, t)$ to directly map the noisy input $\boldsymbol{z}_t$ to clean samples $\boldsymbol{x}$. The core idea is to enforce a consistency property at any two nearby timesteps $t$ and $s$, by minimizing the difference between the model's output. Depending on the $\Delta t := t - s$, the training objective can be categorized into:

- *Discrete-time Consistency Training (CT)* (Geng et al., 2025b; Song et al., 2023; Song & Dhariwal, 2024) minimizes the following discrete time CT loss $\mathcal{L}_{\text{CT}_\text{d}}$:

$$\mathcal{L}_{\text{CT}_\text{d}}(\boldsymbol{\theta}) = \mathbb{E}_{t, s, \boldsymbol{z}_t}\left[\|\boldsymbol{f_\theta}(\boldsymbol{z}_t, t) - \boldsymbol{f}_{\boldsymbol{\theta}^-}(\boldsymbol{z}_s, s)\|_2^2\right], \tag{2}$$

  where $0 \le s < t \le 1$, $\boldsymbol{z}_s = \boldsymbol{z}_t - \Delta t \cdot \boldsymbol{v}$ and $\boldsymbol{f}_{\boldsymbol{\theta}^-} := \texttt{stopgrad}(\boldsymbol{f_\theta})$. While smaller values of $\Delta t$ reduce the discretization error and improve performance, they might also lead to training instability (Song et al., 2023; Geng et al., 2025b). This necessitates a carefully designed scheduler for $\Delta t$ to ensure good performance and stability during training.

- *Continuous-time CT* (Lu & Song, 2025; Song et al., 2023) eliminates the discretization error by the continuous time CT loss $\mathcal{L}_{\text{CT}_\text{c}}$:

$$\mathcal{L}_{\text{CT}_\text{c}}(\boldsymbol{\theta}) = 2\mathbb{E}_{t, \boldsymbol{z}_t}\left[\boldsymbol{f_\theta}^\top(\boldsymbol{z}_t, t)\frac{\mathrm{d}\boldsymbol{f}_{\boldsymbol{\theta}^-}(\boldsymbol{z}_t, t)}{\mathrm{d}t}\right], \tag{3}$$

  Song et al. (2023) theoretically show that $\nabla_{\boldsymbol{\theta}}\mathcal{L}_{\text{CT}_\text{c}}(\boldsymbol{\theta}) = \lim_{\Delta t \to 0} \nabla_{\boldsymbol{\theta}}\mathcal{L}_{\text{CT}_\text{d}}(\boldsymbol{\theta})/\Delta t$. However, estimating $\frac{\mathrm{d}\boldsymbol{f}_{\boldsymbol{\theta}^-}(\boldsymbol{z}_t, t)}{\mathrm{d}t}$ relies on the Jacobian-vector product (JVP) operation, which causes potential issues of scalability and efficiency in modern deep learning frameworks (Wang et al., 2025b; Peng et al., 2025).

**Consistency trajectory model (CTM).** (Kim et al., 2024; Zhou et al., 2025; Frans et al., 2025; Geng et al., 2025a) generalize Consistency Models (CMs) by training a neural network $\boldsymbol{u_\theta}(\boldsymbol{z}_t, r, t)$ to enforce consistency across a trajectory from $t$ to $r$ with $0 \le r \le t \le 1$. This allows jumping from any $t \in (0, 1]$ to any $r < t$ during inference, enabling multi-step generation. To train CTM from scratch:

- *Shortcut model* (Frans et al., 2025) enforces consistency by ensuring that a single "shortcut" step from $t$ to $r$ is consistent with two consecutive shortcut steps of half the size. The training objective is:

$$\mathcal{L}_{\text{SC}}(\boldsymbol{\theta}) = \mathop{\mathbb{E}}_{t, r, \boldsymbol{z}_t}\left[\|\boldsymbol{u_\theta}(\boldsymbol{z}_t, r, t) - \boldsymbol{u}_{\boldsymbol{\theta}^-}(\boldsymbol{z}_t, s, t)/2 - \boldsymbol{u}_{\boldsymbol{\theta}^-}(\boldsymbol{z}_s, r, s)/2\|_2^2\right], \tag{4}$$

  where $\boldsymbol{z}_s = \boldsymbol{z}_t - (t - s) \cdot \boldsymbol{u}_{\boldsymbol{\theta}^-}(\boldsymbol{z}_t, s, t)$ and $s = (t + r)/2$.

- *MeanFlow* (Geng et al., 2025a) trains the model $\boldsymbol{u_\theta}(\boldsymbol{z}_t, r, t)$ to estimate the mean velocity $\frac{1}{t-r}\int_r^t \boldsymbol{v}(\boldsymbol{z}_\tau, \tau)\mathrm{d}\tau$, with training objective given by:

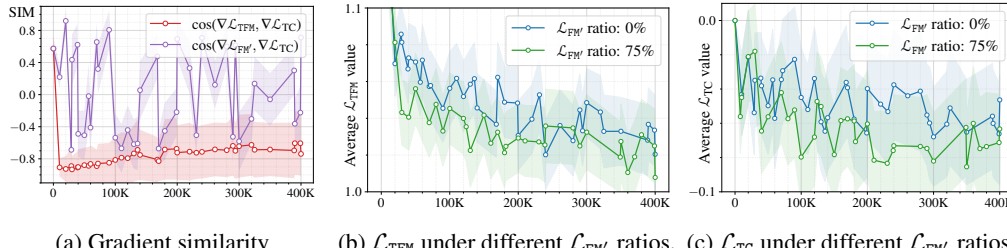

(a) Gradient similarity (b) $\mathcal{L}_{\text{TFM}}$ under different $\mathcal{L}_{\text{FM}'}$ ratios. (c) $\mathcal{L}_{\text{TC}}$ under different $\mathcal{L}_{\text{FM}'}$ ratios.

Figure 3: **MeanFlow training analysis**. (a) Shows the cosine similarity between the gradients of two loss pairs ($\nabla\mathcal{L}_{\text{TC}}$ vs. $\nabla\mathcal{L}_{\text{TFM}}$ and $\nabla\mathcal{L}_{\text{TC}}$ vs. $\nabla\mathcal{L}_{\text{FM}'}$) throughout training. (b) Evaluated $\mathcal{L}_{\text{TFM}}$ when MeanFlow trained with 0% and 75% of $\mathcal{L}_{\text{FM}'}$. (c) Evaluated $\mathcal{L}_{\text{TC}}$ when MeanFlow trained with 0% and 75% of $\mathcal{L}_{\text{FM}'}$.

$$\mathcal{L}_{\text{MF}}(\boldsymbol{\theta}) = \mathop{\mathbb{E}}_{t,r,\boldsymbol{z}_t}\left[\left\|\boldsymbol{u}_{\boldsymbol{\theta}}(\boldsymbol{z}_t, r, t) - \boldsymbol{v}_t + (t - r)\frac{\mathrm{d}\boldsymbol{u}_{\boldsymbol{\theta}^-}(\boldsymbol{z}_t, r, t)}{\mathrm{d}t}\right\|_2^2\right]. \tag{5}$$

In practice, MeanFlow significantly outperforms other one/few-step diffusion and flow models. Yet, there has been little analysis explaining why it works so effectively. To shed light on this, we analyze MeanFlow training in the next section.

## 3 ANALYZING MEANFLOW TRAINING

An intriguing aspect of MeanFlow is the noise distribution used during training: Geng et al. (2025a) empirically found that the best results are achieved when setting $r = t$ for 75% of the samples. This might look counter-intuitive, since we are interested in learning the average velocity on a $[r, t]$ interval to perform large trajectory leaps during inference, so why spending the majority of the training computation on fitting this border case that corresponds to vanilla flow matching supervision? In this section, we show that the MeanFlow loss on its own can be interpreted as velocity consistency training with extra flow matching supervision, and analyze the interaction of these two objectives.

### 3.1 UNDERSTANDING THE OBJECTIVE

Through algebraic manipulations, the original MeanFlow loss $\mathcal{L}_{\text{MF}}$ in Eq. (5) can be rewritten into the following equivalent form (see Section D.1):

$$\mathcal{L}_{\text{MF}}(\boldsymbol{\theta}) = \underbrace{\mathop{\mathbb{E}}_{t,r,\boldsymbol{z}_t}\left[\|\boldsymbol{u}_{\boldsymbol{\theta}}(\boldsymbol{z}_t, r, t) - \boldsymbol{v}_t\|_2^2\right]}_{\text{Trajectory flow matching } \mathcal{L}_{\text{TFM}}} + \underbrace{\mathop{\mathbb{E}}_{t,r,\boldsymbol{z}_t}\left[2(t - r) \cdot \boldsymbol{u}_{\boldsymbol{\theta}}^\top(\boldsymbol{z}_t, r, t)\frac{\mathrm{d}\boldsymbol{u}_{\boldsymbol{\theta}^-}(\boldsymbol{z}_t, r, t)}{\mathrm{d}t}\right]}_{\text{Trajectory consistency } \mathcal{L}_{\text{TC}}} + C, \tag{6}$$

where $C$ is a constant independent of $\boldsymbol{\theta}$. In this decomposition, the first term $\mathcal{L}_{\text{TFM}}$ corresponds to a flow matching loss but with an additional modeling input parameter $r$, so we refer to it as *trajectory flow matching*. The second term $\mathcal{L}_{\text{TC}}$, denoted as *trajectory consistency* loss, acts as a $(t - r)$-reweighted continuous consistency loss [1], but also without a boundary condition (Song et al., 2023). This decomposition highlights that the MeanFlow objective can be interpreted as a consistency (trajectory) model with extra flow matching supervision.

An interesting property of this decomposition is that $\mathcal{L}_{\text{TC}}$ does not have any boundary condition. In comparison, Song et al. (2023) enforces such a condition for vanilla consistency models using a $\boldsymbol{z}_0$-prediction parameterization: without it, the model would quickly converge to a trivial solution (e.g., a constant output). In the MeanFlow case, this collapse does not occur, which suggests that $\mathcal{L}_{\text{TFM}}$ implicitly provides the boundary condition for $\mathcal{L}_{\text{TC}}$. We hypothesize that the absence of an explicit boundary condition makes $\mathcal{L}_{\text{TC}}$ easier to optimize and enables a significantly larger solution space. Further analysis of the solution space is provided in Section E.1.

---

[1]Similarly to the proof in Remark 10 of Song et al. (2023), one can show that this term is equivalent to minimizing the difference between $\boldsymbol{u}_{\boldsymbol{\theta}}(\boldsymbol{z}_t, r, t)$ and $\boldsymbol{u}_{\boldsymbol{\theta}^-}(\boldsymbol{z}_{t-\Delta t}, r, t - \Delta t)$ as $\Delta t \to 0$.

Another important observation here is that trajectory flow matching involves random $r \leqslant t$, which differs from the $r = t$ case used during training by Geng et al. (2025a). To clarify this distinction, we directly compare trajectory flow matching ($\mathcal{L}_{\text{TFM}}$) with vanilla flow matching, which we denote as $\mathcal{L}_{\text{FM}'}$ when using the $u$-prediction parameterization:

$$\mathcal{L}_{\text{TFM}} \triangleq \mathop{\mathbb{E}}_{t,r,\boldsymbol{z}_t} \left[ \|\boldsymbol{u}_{\boldsymbol{\theta}}(\boldsymbol{z}_t, r, t) - \boldsymbol{v}_t\|_2^2 \right], \qquad \mathcal{L}_{\text{FM}'} \triangleq \mathop{\mathbb{E}}_{t,r,\boldsymbol{z}_t | r=t} \left[ \|\boldsymbol{u}_{\boldsymbol{\theta}}(\boldsymbol{z}_t, r, t) - \boldsymbol{v}_t\|_2^2 \right] \qquad (7)$$

Here, $\mathcal{L}_{\text{TFM}}$ arises from the decomposition of the MeanFlow loss, while $\mathcal{L}_{\text{FM}'}$ corresponds to the objective used in Geng et al. (2025a) for joint training. From this formulation, several observations follow. First, $\mathcal{L}_{\text{FM}'}$ is a "part" of $\mathcal{L}_{\text{TFM}}$, active only on the $p(t, r \mid r = t)$ slice of the joint distribution $p(t, r)$. Second, if the network is independent of $r$, then marginalizing out $r$ yields $\mathcal{L}_{\text{TFM}} \equiv \mathcal{L}_{\text{FM}'}$, reducing the objective to vanilla flow matching.

## 3.2 Empirical Analysis

With the decomposition in Equation (6), how does $\mathcal{L}_{\text{FM}'}$ interact with the two decomposed terms? In this section, we analyze the gradients of these losses and examine how extra $\mathcal{L}_{\text{FM}'}$ minimization affects $\mathcal{L}_{\text{TFM}}$ and $\mathcal{L}_{\text{TC}}$ individually. We conduct detailed experiments by training MeanFlow with the DiT-B/2 (Peebles & Xie, 2023) architecture on ImageNet-1K $256^2$ (Deng et al., 2009) for 400K steps. Additional experiment settings are in Section F.

We first analyze the training dynamics by measuring the cosine similarity between the gradients $\nabla \mathcal{L}_{\text{TFM}}$ and $\nabla \mathcal{L}_{\text{TC}}$ during training. As shown in Figure 3a, these two gradients are strongly negatively correlated, with a similarity typically below $-0.4$. This reveals that optimizing $\mathcal{L}_{\text{TFM}}$ and $\mathcal{L}_{\text{TC}}$ jointly is inherently difficult. We hypothesize this stems from the fact that $\mathcal{L}_{\text{TC}}$, without any boundary condition, has a very large optimal solution manifold, compared to $\mathcal{L}_{\text{TFM}}$ whose manifold is very narrow. Thus the optimization process is getting pulled towards the $\mathcal{L}_{\text{TC}}$ manifold, distracting from reaching a narrow intersection.

Given this gradient conflict, the question arises: why does joint training with $\mathcal{L}_{\text{FM}'}$ help? We identify two key reasons: First, as a subset of $\mathcal{L}_{\text{TFM}}$, $\mathcal{L}_{\text{FM}'}$ directly reduces $\mathcal{L}_{\text{TFM}}$. This is empirically confirmed in Figure 3b, where allocating 75% of the training budget to $\mathcal{L}_{\text{FM}'}$ significantly lowers the overall $\mathcal{L}_{\text{TFM}}$ compared to pure MeanFlow training. Second, $\mathcal{L}_{\text{FM}'}$ applies only at $r = t$, where $\mathcal{L}_{\text{TC}} = 0$. Consequently, the gradient $\nabla \mathcal{L}_{\text{FM}'}$ interferes less with $\nabla \mathcal{L}_{\text{TC}}$ than the $\nabla \mathcal{L}_{\text{TFM}}$ gradient. This is demonstrated in Figure 3a, which shows that $\cos(\nabla \mathcal{L}_{\text{FM}'}, \nabla \mathcal{L}_{\text{TC}})$ is consistently higher than $\cos(\nabla \mathcal{L}_{\text{TFM}}, \nabla \mathcal{L}_{\text{TC}})$, that is strongly negative for more than 95% of the training. Surprisingly, $\mathcal{L}_{\text{TC}}$ component doesn't seem to be affected and can even be lower when allocating 75% of the training budget to $\mathcal{L}_{\text{FM}'}$, as shown in Figure 3c. Which again hints at the fact that $\mathcal{L}_{\text{TC}}$ is relatively easy to optimize, even near the $\mathcal{L}_{\text{TFM}}$ optimum.

In conclusion, our analysis reveals three important observations:

> ▷ $\mathcal{L}_{\text{MF}}$ can be decomposed into trajectory flow matching $\mathcal{L}_{\text{TFM}}$ and trajectory consistency $\mathcal{L}_{\text{TC}}$ objectives, whose gradients are strongly negatively correlated during training.
> ▷ $\mathcal{L}_{\text{TC}}$ does not have a necessary boundary condition on its own, implying that $\mathcal{L}_{\text{TFM}}$ serves as an implicit boundary condition for it.
> ▷ $\mathcal{L}_{\text{FM}'}$ acts as a surrogate loss for $\mathcal{L}_{\text{TFM}}$, but with significantly less gradient conflict with the Trajectory consistency loss $\mathcal{L}_{\text{TC}}$.

## 4 $\alpha$-Flow Models

As we showed in the previous section, the $\mathcal{L}_{\text{TFM}}$ loss is difficult to optimize jointly with the $\mathcal{L}_{\text{TC}}$. While the introduction of the $\mathcal{L}_{\text{FM}'}$ loss serves as an effective surrogate for optimizing $\mathcal{L}_{\text{TFM}}$, this approach dedicates a significant portion of training to an objective that is not of our primary interest. This raises a key question: *Can we more efficiently optimize $\mathcal{L}_{\text{TFM}}$ when optimizing $\mathcal{L}_{\text{MF}}$ without this computational overhead?* To answer this, we introduce our $\alpha$-Flow loss, a new family of training objectives for flow-based models.

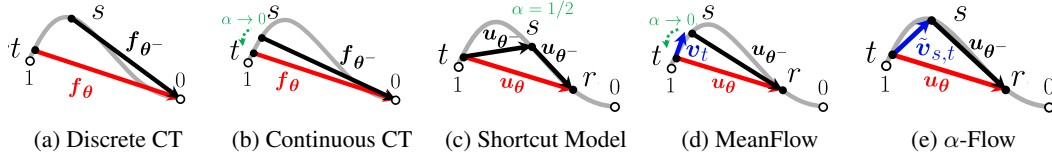

Figure 4: Comparison of training trajectories for various few-step diffusion and flow-based models.

### 4.1 $\alpha$-FLOW: UNIFYING ONE, FEW, AND MANY-STEP FLOW-BASED MODELS

**Definition 1.** *The $\alpha$-Flow loss $\mathcal{L}_\alpha$ is defined as:*

$$\mathcal{L}_\alpha(\boldsymbol{\theta}) \triangleq \mathbb{E}_{t,r,\boldsymbol{z}_t} \left[ \alpha^{-1} \cdot \|\boldsymbol{u}_{\boldsymbol{\theta}}(\boldsymbol{z}_t, r, t) - (\alpha \cdot \tilde{\boldsymbol{v}}_{s,t} + (1-\alpha) \cdot \boldsymbol{u}_{\boldsymbol{\theta}^-}(\boldsymbol{z}_s, r, s))\|_2^2 \right], \tag{8}$$

*where $t, r \in [0, 1]$ is the start and end timestep, $s$ is the intermediate timestep: $s = \alpha \cdot r + (1 - \alpha) \cdot t, \alpha \in (0, 1]$ is the consistency step ratio, and $\boldsymbol{z}_s = \boldsymbol{z}_t + (t - s) \cdot \tilde{\boldsymbol{v}}_{s,t}$ is the trajectory value at this timestep $s$. Here, $\tilde{\boldsymbol{v}}_{s,t}$ is the "shift velocity" used to estimate the intermediate variable $\boldsymbol{z}_s$ from $\boldsymbol{z}_t$.*

The $\alpha$-Flow loss is visualized in Figure 4e. Intuitively, it enforces trajectory consistency between $t$ and $r$ by introducing an additional $s$, which is an interpolation between $t, r$ with ratio $\alpha$. More importantly, this definition generalizes previously introduced training objectives such as trajectory flow matching, Shortcut Model training, and MeanFlow training:

**Theorem 1.** *The $\alpha$-Flow loss unifies flow matching, Shortcut Models, and MeanFlow:*

▷ $\mathcal{L}_{\text{TFM}}(\boldsymbol{\theta}) = \mathcal{L}_{\alpha=1}(\boldsymbol{\theta})$ *with* $\tilde{\boldsymbol{v}}_{s,t} = \boldsymbol{v}_t$.

▷ $\mathcal{L}_{\text{SC}}(\boldsymbol{\theta}) = \frac{1}{2}\mathcal{L}_{\alpha=1/2}(\boldsymbol{\theta})$ *with* $\tilde{\boldsymbol{v}}_{s,t} = \boldsymbol{u}_{\boldsymbol{\theta}^-}(\boldsymbol{z}_t, s, t)$.

▷ $\nabla_{\boldsymbol{\theta}}\mathcal{L}_{\text{MF}}(\boldsymbol{\theta}) = \nabla_{\boldsymbol{\theta}}\mathcal{L}_{\alpha\to 0}(\boldsymbol{\theta})$ *with* $\tilde{\boldsymbol{v}}_{s,t} = \boldsymbol{v}_t$.

**Proof sketch:** The connection between $\mathcal{L}_\alpha$ and $\mathcal{L}_{\text{TFM}}$ and $\mathcal{L}_{\text{SC}}$ are straight forward. For the non-trivial relationship showing the convergence from $\nabla_{\boldsymbol{\theta}}\mathcal{L}_\alpha(\boldsymbol{\theta})$ to $\nabla_{\boldsymbol{\theta}}\mathcal{L}_{\text{MF}}(\boldsymbol{\theta})$, we leverage a first-order Taylor expansion on the term $\boldsymbol{u}_{\boldsymbol{\theta}^-}(\boldsymbol{z}_s, r, s)$ around $s = t$. This yields:

$$\boldsymbol{u}_{\boldsymbol{\theta}^-}(\boldsymbol{z}_s, r, s) = \boldsymbol{u}_{\boldsymbol{\theta}^-}(\boldsymbol{z}_t, r, t) - \frac{\mathrm{d}\boldsymbol{u}_{\boldsymbol{\theta}^-}(\boldsymbol{z}_t, r, t)}{\mathrm{d}t}(t - r)\alpha + \mathcal{O}\left(\alpha^2\right),$$

Substituting this expansion into the Equation (8) and taking the limit as $\alpha \to 0$ causes the higher-order terms $\mathcal{O}\left(\alpha^2\right)$ to vanish and recover $\nabla_{\boldsymbol{\theta}}\mathcal{L}_{\text{MF}}(\boldsymbol{\theta})$. The detailed proof is deferred to Section D.2. Furthermore, under Assumption 2, we prove the upper bound of the non-asymptotic distance between $\nabla_{\boldsymbol{\theta}}\mathcal{L}_\alpha(\boldsymbol{\theta})$ and $\nabla_{\boldsymbol{\theta}}\mathcal{L}_{\text{MF}}(\boldsymbol{\theta})$ in Appendix E.3. This bound is linearly related to $\alpha$ and converges to 0 as $\alpha \to 0$, which aligns with the result of Theorem 1.

Moreover, if one considers a $\boldsymbol{z}_0$-parametrization $\boldsymbol{u}_{\boldsymbol{\theta}}(\boldsymbol{z}_t, 0, t) = (\boldsymbol{z}_t - \boldsymbol{f}_{\boldsymbol{\theta}}(\boldsymbol{z}_t)) / t = \hat{\boldsymbol{z}}_0$, $\mathcal{L}_\alpha$ incorporates discrete and continuous consistency training as well. With $\tilde{\boldsymbol{v}}_{s,t} = \boldsymbol{v}_t$ and $r \equiv 0$:

▷ $\mathcal{L}_{\text{CT}_d}(\boldsymbol{\theta}) = \mathcal{L}_{\alpha=\delta}(\boldsymbol{\theta})$ *for* $\delta \in (0, t)$.

▷ $\nabla_{\boldsymbol{\theta}}\mathcal{L}_{\text{CT}_c}(\boldsymbol{\theta}) = \nabla_{\boldsymbol{\theta}}\mathcal{L}_{\alpha\to 0}(\boldsymbol{\theta})$.

This theorem reveals that the ratio $\alpha$ is the key hyperparameter that unifies seemingly different methods, which controls the relative position of the intermediate timestep $s$ within the $(r, t)$ interval. By annealing $\alpha$ from 1 to 0, we obtain a family of models in the interpolation between trajectory flow matching and MeanFlow. Notably, discrete CT is a special case of $\alpha$-Flow with $r \equiv 0$. Unlike discrete CT, $\alpha$-Flow requires no complex timestep partitioning: once $t$ and $r$ are sampled, $s$ is immediately determined with a fixed $\alpha$.

### 4.2 $\alpha$-FLOW MODELS

The $\alpha$-Flow loss enables a curriculum learning strategy that progressively transitions from the trajectory flow matching to MeanFlow objective. As visualize in Figure 2, this approach better disentangles the optimization of the trajectory flow matching and consistency losses, could potentially

reduce reliance on the flow matching objective, and leads to better convergence. The detailed curriculum learning can be summarized into three phases:

- **Trajectory flow matching pretraining** ($\alpha = 1$). To speed-up convergence toward narrow $\mathcal{L}_{\text{TFM}}$ manifold, we prioritize optimizing trajectory flow matching in the early training phase. Additionally, as a low-variance objective, trajectory flow matching quickly establishes a reliable noise-to-data mapping, providing a good initialization for subsequent few-step refinement. Notably, this pretraining strategy is aligned with previous diffusion model pretraining strategy applied on consistency model (Geng et al., 2025b), while we start from different motivations and generalize it into the $\alpha$-Flow framework.

- **$\alpha$-Flow transition** ($\alpha \in (0, 1)$). After initial training establishes a solid foundation with $\mathcal{L}_{\text{TFM}}$, we transition to $\mathcal{L}_{\text{MF}}$ via curriculum learning: progressively decreasing $\alpha$ from 1 to 0. This gradual shift, inspired by discrete CT methods (Song et al., 2023), serves two critical purposes: (a) theoretically, the optimal solution of $\alpha$-Flow smoothly transitions from that of $\mathcal{L}_{\text{TFM}}$ to that of $\mathcal{L}_{\text{MF}}$ as $\alpha$ decreases (Section E.2); (b) curriculum learning steers the training objective from a "high-bias, low-variance" state to the necessary "low-bias, high-variance" state, with theoretical proof showing gradient variance increases as $\alpha$ decreases (Section E.4). This strategy yields significantly better convergence than directly optimizing the high-variance MeanFlow objective.

- **MeanFlow fine-tuning** ($\alpha \to 0$). In the final stage, we focus entirely on the MeanFlow training objective. Unlike the original paper, our improved early-stage optimization of trajectory flow matching significantly reduces the need for the flow matching loss (as shown in Table 2 (b)) and achieves significantly better few-step generation quality.

The overall training code of $\alpha$-Flow is shown in Algorithm 1, where we first sample $t, r$ and obtain the $\alpha$ from the schedule. Based on whether $\alpha = 0$ or not, $\alpha$-Flow will use either $\mathcal{L}_{\text{MF}}$ or $\mathcal{L}_{\alpha}$ to train the model. $\alpha$-Flow applies the same training details as MeanFlow when training $\mathcal{L}_{\text{MF}}$ (except a lower ratio of flow matching). Below, we only show the difference: the schedule of $\alpha$ as well as the design space of $\mathcal{L}_{\alpha}$ when $\alpha > 0$.

**Schedule.** To schedule the training, we use a sigmoid function, $\alpha = \text{Sigmoid}_{k_s \Rightarrow k_e, \gamma, \eta}(k)$, which depends on the training iteration $k$. The function is defined by its starting and ending iterations, $k_s, k_e$, a temperature parameter $\gamma$ (set to be 25) and a clamping value $\eta$. The specific implementation can be found in Algorithm 2. Figure 6 provides a visualization of this scheduler, while Section 5.2 conducts an ablation study over its parameters.

---

**Algorithm 1** $\alpha$-Flow: Training.

```
# fn(z, r, t): function to predict u
# x: training batch, k: training iterations

t, r = sample_t_r()
alpha = sample_alpha(k)
s = alpha * r + (1 - alpha) * t
e = randn_like(x)

zt = (1 - t) * x + t * e
v = e - x

if alpha == 0:
    u, dudt = jvp(fn, (zt, r, t), (v, 0, 1))
    u_tgt = v - (t - r) * dudt
else :
    u = fn(zt, r, t)
    zs = zt - (t - s) * v
    u_tgt = alpha * v + (1 - alpha) * fn(zs, r, s)

error = u - stopgrad(u_tgt)
loss = metric(error)
```

---

**Algorithm 2** $\alpha$-Flow: Curriculum Schedule

```
# k_s, k_e: start/end schedule iterations,
# gamma: temperature parameter
# k: current iteration, eta: clamping value

scale = 1 / (k_e - k_s)
offset = - (k_s + k_e) / 2 / (k_e - k_s)
alpha = 1 - sigmoid((scale * k + offset) * gamma)
alpha = 1 if alpha > (1- eta) else (0 if alpha < eta else
    alpha)
```

---

**Clamping value.** Geng et al. (2025b) show that when $\Delta t = t - s$ approaches 0, the performance of few-step CT model will first increase and then decrease. For $\alpha$-Flow, we observe a similar phenomenon: by training $\alpha$-Flow with a fixed $\alpha$, as $\alpha$ approaches 0, the 1-step generation performance will first increase then decrease. Detailed experiments are shown in Table 5 (c). From the experiment, the optimal performance is achieved when $\alpha = 5 \times 10^{-3}$. Thus, we set a clamping value $\eta = 5 \times 10^{-3}$ for the schedule. $\alpha$ will be set to 0 when $\alpha < \eta$. We also use the same clamping value to set $\alpha$ to 1 when $\alpha > 1 - \eta$, as when $\alpha$ is close to 1, $\mathcal{L}_{\text{TFM}}$ is similar to $\mathcal{L}_{\alpha}$ but more efficient.

| Method | Source | Params | Epochs | NFE 1 | | NFE 2 | | |
|---|---|---|---|---|---|---|---|---|
| | | | | FID | FDD | FID | FDD | FID$^\dagger$ |
| Shortcut-XL/2 | Frans et al. (2025) | 675M | 160 | 10.60 | – | – | – | – |
| IMM-XL/2 | Zhou et al. (2025) | 676M | 3840 | 8.05 | – | 3.88 | – | – |
| MeanFlow-XL/2 | Geng et al. (2025a) | 676M | 240 | 3.43 | – | 2.93 | – | – |
| MeanFlow-XL/2+ | Geng et al. (2025a) | 676M | 1000 | – | – | 2.20 | – | – |
| FACM-XL/2 | Peng et al. (2025) | 675M | $800 + 250 \times 2$ | – | – | – | – | 2.07 |
| FACM-XL/2 | | 675M | $120 \times 2$ | 9.54 | 410.4 | 7.31 | 362.0 | – |
| FACM-XL/2 | Our reproduction | 675M | $240 \times 2$ | 6.59 | 327.7 | 4.73 | 278.6 | – |
| CT-XL/2 | | 676M | 240 | 7.44 | 324.9 | 6.22 | 271.9 | – |
| MeanFlow-B/2 | | 131M | 240 | 6.04 | 312.3 | 5.17 | 232.1 | – |
| MeanFlow-XL/2 | | 676M | 240 | 3.47 | 185.8 | 2.46 | 108.7 | 2.26 |
| $\alpha$-Flow-B/2 | | 131M | 240 | 5.40 | 287.1 | 5.01 | 231.8 | – |
| $\alpha$-Flow-XL/2 | Our methods | 676M | 240 | 2.95 | 164.6 | 2.34 | 105.7 | 2.16 |
| $\alpha$-Flow-XL/2+ | | 676M | 240+60 | **2.58** | **148.4** | **2.15** | **96.8** | **1.95** |

Table 1: **Class-conditional generation on ImageNet-256×256**. The table reports the results for few-step diffusion/flow matching-based methods trained from scratch. "×2" indicates that FACM requires roughly twice the computation per epoch compared to other methods. For a direct "epoch-to-epoch comparison," $\alpha$-Flow-XL/2, MeanFlow-XL/2 and FACM-XL/2 are each trained for 240 epochs. $\alpha$-Flow-XL/2+ is a fine-tuned version of $\alpha$-Flow-XL/2, trained for extra 60 epochs with a batch size of 1024. † FID scores are evaluated with the balanced class sampling (see Section J).

**Training objective.** In the unifying space of $\alpha$-Flow loss, all other few-step models set $\tilde{v}_{s,t} = v_t$ except the shortcut model which uses $\tilde{v}_{s,t} = u_{\theta^-}(z_t, s, t)$. Additionally, we are interested in seeing whether we need exponential moving average (EMA) for $\theta^-$. With ablation study in Table 5 (a), we set $\tilde{v}_{s,t} = v_t$ and do not use EMA for $\theta^-$.

**Adaptive loss weight.** MeanFlow (Geng et al., 2025a) demonstrates the effectiveness of adaptive loss. Basically, let $||\Delta||_2^2$ denote the squared L2 loss. The adaptive loss weight $\omega = 1/(||\Delta||_2^2 + c)$ where $c = 10^{-3}$. And the adaptively weighted loss is $\text{sg}(\omega)||\Delta||_2^2$. Theoretically, we derived an equivalent adaptive loss weight $\omega = \alpha/(||\Delta||_2^2 + c)$ for $\mathcal{L}_\alpha$. We defer the derivation in Section H.2. With ablation study in Table 5 (b), we demonstrate the derived adaptive loss weight is better than other loss weights.

**Classifier-free guidance (CFG).** We apply a similar CFG training strategy as MeanFlow, by setting $\tilde{v}_{s,t}$ in Equation (8) with $\tilde{v}_{s,t} = w \cdot v(z_t, t|x) + \kappa \cdot u_{\theta^-}(z_t, t|c) + (1 - w - \kappa) \cdot u_{\theta^-}(z_t, t|\varnothing)$, where $w, \kappa$ are the guidance scale, $u_{\theta^-}(\cdot|c)$, $u_{\theta^-}(\cdot|\varnothing)$ denotes the class-condition (with class $c$) and class-unconditional prediction. Detailed settings are deferred to Section G.

**Sampling.** We employ both consistency sampling (Song et al., 2023) and ODE sampling for two-step generation. Implementation details are provided in Algorithm 3. Empirically, we observe that consistency sampling outperforms ODE sampling for larger models with better convergence. Consequently, we adopt ODE sampling for all DiT-B/2 architectures and consistency sampling for all DiT-XL/2 architectures, with additional ablation studies on DiT-XL/2 presented in Figure 5.

## 5 EXPERIMENTS

In this section, we employ $\alpha$-Flow on real image dataset ImageNet-1K $256^2$ Deng et al. (2009) (Section 5.1, Section 5.2) and video dataset Kinetics-700 $17 \times 256^2$ Carreira et al. (2019) (Section 5.3). We use exactly the same DiT Peebles & Xie (2023) architecture as MeanFlow Geng et al. (2025a). For evaluation, we use Fréchet Inception Distance (FID) Heusel et al. (2017), Fréchet DINOv2 Distance (FDD) Oquab et al. (2023). We also use Fréchet Video Distance (FVD) Unterthiner et al. (2019) for video generation evalution. We evaluate model performance for both 1 and 2 Number of Function Evaluations (NFE=1, NFE=2). We implement our models in the latent space of the

| Schedule | NFE 1 | | NFE 2 | |
|---|---|---|---|---|
| | FID | FDD | FID | FDD |
| Constant$_{0.0}$ | 44.4 | 844.1 | 42.1 | 836.3 |
| *Trajectory flow matching iterations* | | | | |
| Sigmoid$_{0K \to 100K}$ | 44.3 | 860.3 | 40.8 | 826.9 |
| Sigmoid$_{50K \to 150K}$ | 44.1 | 846.8 | 39.9 | 811.6 |
| Sigmoid$_{100K \to 200K}$ | 42.4 | 828.0 | 38.3 | 795.4 |
| Sigmoid$_{150K \to 250K}$ | 41.3 | 818.8 | 38.1 | 793.1 |
| *Transition iterations* | | | | |
| Sigmoid$_{200K \to 200K}$ | 41.4 | 794.4 | 38.8 | 796.7 |
| Sigmoid$_{150K \to 250K}$ | 41.3 | 818.8 | 38.1 | 793.1 |
| Sigmoid$_{0K \to 400K}$ | **40.0** | **785.4** | **37.1** | **782.9** |

(a) **Consistency step ratio schedule.**

| | Model | NFE 1 | | NFE 2 | |
|---|---|---|---|---|---|
| % $r = t$ | Schedule | FID | FDD | FID | FDD |
| 0% | Constant$_{0.0}$ | 46.0 | 879.6 | 44.3 | 867.7 |
| | Sigmoid$_{0K \to 400K}$ | 40.4 | 822.5 | 38.9 | 811.8 |
| 25% | Constant$_{0.0}$ | 44.4 | 844.1 | 42.1 | 836.3 |
| | Sigmoid$_{0K \to 400K}$ | **40.0** | 785.4 | 37.1 | 782.9 |
| 50% | Constant$_{0.0}$ | 43.9 | 844.1 | 42.1 | 836.3 |
| | Sigmoid$_{0K \to 400K}$ | 40.2 | **781.0** | 37.1 | 775.0 |
| 75% | Constant$_{0.0}$ | 43.1 | 819.2 | 38.5 | 787.6 |
| | Sigmoid$_{0K \to 400K}$ | 42.2 | 810.5 | **36.2** | **754.7** |

(b) **Flow matching ratio.**

Table 2: Ablation study on ImageNet-1K $256^2$ for $\alpha$-Flow-B/2.

Stable Diffusion Variational Autoencoder (SD-VAE) [2]. More details on the experiments settings are in Section G.

## 5.1 COMPARISON WITH BASELINE

In Table 1, we compare $\alpha$-Flow with previous few-step Diffusion and Flow models, demonstrating its superior performance for 1-NFE and 2-NFE generation. Across models trained for 240 epochs, $\alpha$-Flow-XL/2 achieves **2.95** FID (**164.6** FDD), representing a relative improvement of 15% (12%) over MeanFlow-XL/2 and 70% (60%) over FACM-XL/2. Our best model, $\alpha$-Flow-XL/2+, sets a new state-of-the-art 1-NFE generation with an impressive FID of **2.58** (**148.4** FDD), compared with all the other few-step Diffusion and Flow models trained over the SD-VAE. Furthermore, for 2-NFE generation, $\alpha$-Flow-XL/2+ achieves **2.15** FID (**96.8** FDD), outperforms all these baseline methods. It's particularly notable that it surpasses FACM-XL/2's 2.07 FID (achieved with a class-balanced sampling) by reaching 1.95 FID with only 23% of the training epochs. Uncurated samples, shown in Figure 1 and Section L, visually confirm these results. Specifically in Figure 1, $\alpha$-Flow-XL/2 generates more images with better quality, as highlighted in green.

## 5.2 ABLATION STUDY

**Consistency step ratio schedule.** In Table 2 (a), we evaluate our $\alpha$-Flow framework trained with various sigmoid schedules, as visualized in Figure 6. For these experiments, the flow matching ratio is fixed at 25%. We first analyze the impact of the trajectory flow matching pretraining duration. By fixing $k_e - k_s$ to 100K iterations, we progressively increase $k_s$ from 0K to 150K. As the pretraining duration increases, $\alpha$-Flow's performance consistently improves across all metrics. The best-performing schedule, Sigmoid$_{150K \to 250K}$, significantly outperforms the baseline Mean-Flow (Constant$_{0.0}$). This suggests that *optimizing trajectory flow matching is more crucial than optimizing MeanFlow in the early training stages for achieving superior few-step flow modeling*. This finding aligns with our empirical analysis, which shows that because the gradients of the trajectory flow matching and consistency losses conflict, it is more efficient to exclusively optimize the trajectory flow matching objective for faster initial convergence.

Next, we investigate the effect of the transition duration. With the midpoint $(k_s + k_e)/2$ fixed at 200K iterations, we vary the total transition iterations from 0 to 400K. Our results indicate that a longer, smoother transition leads to better generation quality. This highlights the importance of gradually reducing the bias of the training objective by smoothly transitioning between trajectory flow matching and MeanFlow.

**Flow matching ratio.** In Table 2 (b), we compare our $\alpha$-Flow framework with the MeanFlow baseline across various flow matching ratios ($\%r = t$). Our results show that $\alpha$-Flow consistently outperforms MeanFlow for all evaluated ratios, confirming the effectiveness of our proposed method. A key finding is that $\alpha$-Flow achieves its best 1-NFE performance at a relatively low flow matching

---
[2]The EMA version in https://huggingface.co/stabilityai/sd-vae-ft-mse

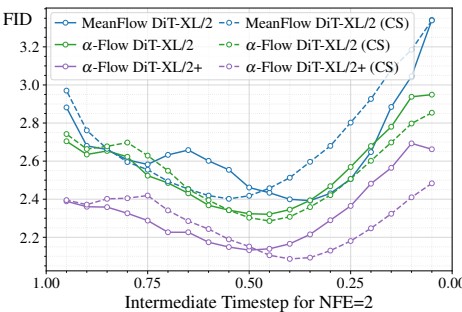 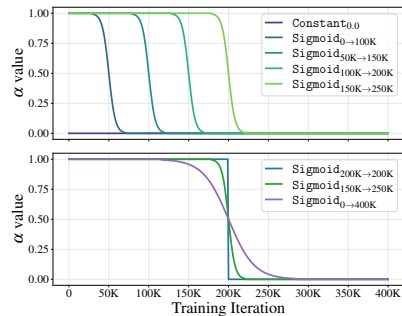

Figure 5: Comparing ODE vs consistency sampling for MeanFlow and $\alpha$-Flow models.

Figure 6: Visualization of consistency step ratio schedule.

ratio. Specifically, it reaches the best FID of 40.0 at 25 % of $r = t$ and the best FDD of 781.0 at 50 % of $r = t$, while MeanFlow requires a higher ratio of 75% to achieve its best FID of 43.1 and FDD of 819.2. This aligns with our motivation: by pretraining on trajectory flow matching, $\alpha$-Flow is less reliant on the flow matching objective and can focus more on the overall MeanFlow objective, leading to superior one-step generation quality.

Furthermore, we observe that for $\alpha$-Flow, the flow matching ratio presents a clear trade-off between 1-NFE and 2-NFE performance. For instance, the 75% ratio yields worse NFE=1 but better NFE=2 generation results compared to the 50%-ratio version. This indicates that a higher proportion of flow matching improves the model's ability to generate images in a slightly higher number of steps.

**Sampling.** As shown in Figure 5, we compare ODE sampling (solid line) and consistency sampling (dotted line) for 2-NFE generation across different intermediate sampling timesteps, using MeanFlow-XL/2, $\alpha$-Flow-XL/2, and $\alpha$-Flow-XL/2+. The results show that consistency sampling yields better generation performance for both $\alpha$-Flow-XL/2 and $\alpha$-Flow-XL/2+, achieving the best FID scores of 2.09 at timestep 0.4 and 2.28 at timestep 0.45, respectively. In contrast, ODE sampling performs better for MeanFlow-XL/2, which attains its best FID of 2.39 at timestep 0.35. In Table 1, we select intermediate sampling timesteps that balance FID and FDD; see Table 3 for details.

### 5.3 EXPERIMENTS ON KINETICS-700 $17 \times 256^2$

We evaluate the generalization capability of $\alpha$-Flow on video generation. As shown in the Figure 7, $\alpha$-Flow consistently outperforms Mean-Flow across all metrics and settings. When training without CFG, $\alpha$-Flow achieved improvements of 2.5 (NFE 1 FID), 21.5 (NFE 1 FDD), 21.5 (NFE 1 FVD), 1.4 (NFE 2 FID), 11.9 (NFE 2 FDD), and 13.5 (NFE 2 FVD); When training with CFG: $\alpha$-

| Method | NFE 1 | | | NFE 2 | | |
|---|---|---|---|---|---|---|
| | FID | FDD | FVD | FID | FDD | FVD |
| MeanFlow-B/2 | 53.2 | 1031.9 | 619.5 | 49.5 | 995.3 | 580.8 |
| $\alpha$-Flow-B/2 | **50.7** | **1010.4** | **598.0** | **48.1** | **983.2** | **567.3** |
| MeanFlow-B/2-cfg | 32.3 | 779.0 | 298.1 | 26.6 | 719.8 | 249.6 |
| $\alpha$-Flow-B/2-cfg | **29.6** | **709.8** | **276.2** | **24.9** | **657.4** | **217.3** |

Figure 7: Kinetics-700 $17 \times 256^2$ experiments.

Flow achieved improvements of 2.7 (NFE 1 FID), 69.2 (NFE 1 FDD), 21.9 (NFE 1 FVD), 1.7 (NFE 2 FID), 62.4 (NFE 2 FDD), and 32.3 (NFE 2 FVD). These experimental results successfully demonstrate the effectiveness and robust generalization of the $\alpha$-Flow framework across more diverse datasets and modalities.

### 6 CONCLUSION

Our work provided a principled analysis of the MeanFlow framework, analyzing its objective and establishing the necessity of flow matching supervision during training. Motivated by this understanding, we proposed the $\alpha$-Flow objective as a generalization of MeanFlow loss, allowing us to train consistently stronger few-step image generation models from scratch.

## 7 REPRODUCIBILITY STATEMENT

We are committed to ensuring the reproducibility of our results. To this end, we include all the necessary implementation details in Section G, ensuring that our methodology can be faithfully reproduced. We will publicly release our source training, inference, and evaluation code, as well as the pre-trained checkpoints for ImageNet-1K $256^2$.

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

## A  RELATED WORK

**Diffusion Models.**  Diffusion models have become a dominant paradigm in generative modeling for vision domains (Sohl-Dickstein et al., 2015; Song & Ermon, 2019; Ho et al., 2020; Song et al., 2021a;b; Dhariwal & Nichol, 2021). The classical diffusion framework defines a forward noising process and a corresponding reverse process that a model learns to approximate. Early works such as DDPM (Ho et al., 2020) and score-based generative modeling (Song & Ermon, 2019) demonstrated high-quality image generation, later extended to continuous-time SDEs and ODEs (Song et al., 2021b). (Dhariwal & Nichol, 2021) further improved sample fidelity with larger architectures and classifier guidance. More recently, the community has explored *flow-based* parameterizations that directly learn continuous velocity fields (Liu et al., 2023; Lipman et al., 2023; Albergo & Vanden-Eijnden, 2022). These flow matching approaches simplify training, unify score- and likelihood-based models, and are used in large-scale systems such as Stable Diffusion 3 (Esser et al., 2024).

**Few-step Diffusion.**  Despite their quality, diffusion models are computationally expensive due to iterative sampling. A large body of work accelerates sampling to a few steps or even one step. Distillation-based approaches include progressive distillation (Salimans & Ho, 2022; Berthelot et al., 2023), and often incorporate adversarial objectives (Yin et al., 2024b;a; Zhou et al., 2024; Sauer et al., 2024). UCGM (Sun et al., 2025) develops a unified training scheme for multi-step and few-step diffusion-based methods.

A closer research direction (which our method follows as well) includes the methods which are trained from scratch and support few- and even one-step generation by design. Consistency Models (CMs) (Song et al., 2023) learn to map noisy inputs directly to clean data by enforcing self-consistency. Extensions improve stability and scalability (Song & Dhariwal, 2024; Lu & Song, 2025; Geng et al., 2025b). *Trajectory-based* methods learn the dynamics of the entire denoising process, enabling arbitrary jumps along the diffusion path. PCM (Wang et al., 2024) scale consistency distillation to large scale models and optimize with preselected time intervals. Shortcut diffusion models (Frans et al., 2025) learn direct mappings with shortcut constraints. MeanFlow (Geng et al., 2025a) predicts time-averaged velocities with continuous consistency, while Guo et al. (2025) explore this idea for discrete consistency. While concurrent work Guo et al. (2025) proposed a similar loss function to our $\alpha$-Flow loss (Equation (8)), our paper offers theoretical and empirical contributions. Theoretically, we provide a deeper analysis comparing the $\alpha$-Flow loss and the MeanFlow loss. Empirically, we introduce novel techniques specifically designed to improve the performance of the $\alpha$-Flow loss, and we successfully demonstrate its efficacy and the benefits of curriculum learning on the large-scale ImageNet dataset. Hybrid approaches combine consistency and flow matching: Consistency-FM (Yang et al., 2024) enforces velocity self-consistency, FACM (Peng et al., 2025) anchors consistency to flow objectives, and IMM (Zhou et al., 2025) matches the output distributions via moment matching instead of exact outputs. Consistency Trajectory Models (CTM) (Kim et al., 2024) generalize consistency training to support transitions between any two timesteps, combining one-step generation with progressive refinement. Boffi et al. (2025b) introduces Flow Matching Mapping (FMM), a unified framework that extends CMs, CTM, and progressive distillation. In particular, it demonstrates that existing methods can be interpreted within a common Euler and progressive perspective, while also proposing a novel class of Lagrangian methods. Building on this foundation, (Boffi et al., 2025a) presents a systematic algorithmic framework and demonstrates the superior performance of the Lagrangian approach. Our work, on the other hand, provides a distinct and detailed unification focused specifically on the Euler method. Within this framework, we unify Flow Matching, Shortcut models, and MeanFlow. Transition Models (TiM) (Wang et al., 2025a) derive an exact continuous-time dynamics equation for arbitrary-step transitions. These methods achieve one- to few-step sampling with steadily improving fidelity.

## B  LIMITATIONS

- Our $\alpha$-Flow loss enables high-quality training of discrete MeanFlow models without requiring JVP computation. However, in practice, the continuous objective (i.e., setting $\alpha \to 0$) remains important, likely due to the bias–variance trade-off inherent in the consistency objective (Song et al., 2023; Song & Dhariwal, 2024).

- We occasionally observed unstable training in large-scale models with guidance integration, both for the vanilla MeanFlow model and our $\alpha$-Flow variant. Thus, our framework should not be viewed as a silver bullet for addressing the well-known instability issues of consistency models Geng et al. (2025b).

- The $\alpha$-Flow objective uses pure flow matching supervision up to $k_s$ iterations, after which the consistency objective is applied. Before this point, the model's few-step performance is weak, which can make progress harder to monitor.

- Our gradient analysis provides actionable insights but remains empirical; it does not fully explain, from a theoretical perspective, why flow matching is so critical for consistency.

- Although we motivate larger batch sizes for fine-tuning by the high variance of the consistency loss, the observed improvements (see Table 4) may instead reflect that small batches are more sensitive to hyperparameters (Marek et al., 2025), and that beyond a certain size, batch-size scaling exhibits diminishing returns (McCandlish et al., 2018).

## C  FAILED EXPERIMENTS

We also wish to share with the community several experiments that did not succeed during the course of this project. Some of these directions were likely underexplored on our side, while others may represent genuine dead-ends. Nevertheless, we believe documenting them may serve as a useful reference for future work.

- We devoted several weeks to exploring decomposed training of the MeanFlow objective with individually tuned weighting functions for each term, drawing inspiration from EDM Karras et al. (2022) to map out the design space. Unfortunately, every configuration we attempted produced worse results than the default adaptive loss heuristic, which was a particularly frustrating outcome.

- Consistency sampling (see Figure 5) did not provide the improvements we had anticipated. Interestingly, the optimal midpoint consistently emerged at $\approx 0.5$, which coincides with the default MeanFlow setting. We suspect this effect is related to the training distribution, which has a mode slightly lower 0.5. Following the original work, we employed a logit-normal distribution with location parameter $-0.4$.

- We experimented with LoRA fine-tuning and introduced separate prediction heads for vanilla velocity and mean velocity. Neither approach yielded promising results.

- We conducted roughly 50 ablations on the train-time noise schedule for vanilla MeanFlow models. None resulted in noticeably better performance, even when factorizing the joint distribution $p(t, r)$ into $p(t)p(r|t)$ and exploring alternative supervision distributions for flow matching in parallel.

- We investigated additional representation alignment losses Yu et al. (2025) with the aim of accelerating convergence in MeanFlow models. However, the observed gains were insufficient to justify the added complexity of the training framework.

- We also experimented with different EMA schedules, but these attempts did not lead to meaningful improvements.

## D  PROOFS OF THINGS

### D.1  LOSS DECOMPOSITION

*Proof.* The MeanFlow loss is given by:

$$\mathcal{L}_{\text{MF}}(\boldsymbol{\theta}) = \mathbb{E}_{t,r,\boldsymbol{z}_t} \left[ \left\| \boldsymbol{u}_{\boldsymbol{\theta}}(\boldsymbol{z}_t, r, t) - \boldsymbol{v}_t + (t - r) \frac{\mathrm{d}\boldsymbol{u}_{\boldsymbol{\theta}^-}(\boldsymbol{z}_t, r, t)}{\mathrm{d}t} \right\|_2^2 \right] \tag{9}$$

(unpacking the norm and regrouping terms yields)

$$= \underbrace{\mathbb{E}_{t,r,\boldsymbol{z}_t} \left[ \|\boldsymbol{u}_{\boldsymbol{\theta}}(\boldsymbol{z}_t, r, t) - \boldsymbol{v}_t\|_2^2 \right]}_{\mathcal{L}_{\mathrm{TFM}}(\boldsymbol{\theta})} + \underbrace{\mathbb{E}_{t,r,\boldsymbol{z}_t} \left[ 2 \cdot (t - r) \cdot \boldsymbol{u}_{\boldsymbol{\theta}}^\top(\boldsymbol{z}_t, r, t) \frac{\mathrm{d}\boldsymbol{u}_{\boldsymbol{\theta}^-}(\boldsymbol{z}_t, r, t)}{\mathrm{d}t} \right]}_{\mathcal{L}_{\mathrm{TC}}(\boldsymbol{\theta})} \quad (10)$$

$$+ \underbrace{\mathbb{E}_{t,r,\boldsymbol{z}_t} \left[ -2 (t - r) \cdot \boldsymbol{v}^\top(\boldsymbol{z}_t, t | \boldsymbol{x}) \frac{\mathrm{d}\boldsymbol{u}_{\boldsymbol{\theta}^-}(\boldsymbol{z}_t, r, t)}{\mathrm{d}t} + (t - r)^2 \left\| \frac{\mathrm{d}\boldsymbol{u}_{\boldsymbol{\theta}^-}(\boldsymbol{z}_t, r, t)}{\mathrm{d}t} \right\|_2^2 \right]}_{\text{Does not depend on } \boldsymbol{\theta}} \quad (11)$$

$\square$

## D.2 $\mathcal{L}_\alpha$ LOSS UNIFICATION

*Proof of theorem 1.* The proof for flow matching and shortcut models is straightforward. We will only show the proof for the third bullet point. For brevity, let's set $\Delta t = t - s$ and $\alpha = \dfrac{\Delta t}{t - r}$.

$$\mathcal{L}_\alpha(\boldsymbol{\theta}) = \mathbb{E}_{t,r,\boldsymbol{z}_t} \left[ \frac{t - r}{\Delta t} \cdot \left\| \boldsymbol{u}_{\boldsymbol{\theta}}(\boldsymbol{z}_t, r, t) - \frac{\Delta t}{t - r} \cdot \boldsymbol{v}_t - \right. \right.$$
$$\left. \left. \frac{t - \Delta t - r}{t - r} \boldsymbol{u}_{\boldsymbol{\theta}^-}(\boldsymbol{z}_{t-\Delta t}, r, t - \Delta t) \right\|_2^2 \right],$$

$$\stackrel{(i)}{=} \mathbb{E}_{t,r,\boldsymbol{z}_t} \left[ \frac{t - r}{\Delta t} \cdot \left\| \boldsymbol{u}_{\boldsymbol{\theta}}(\boldsymbol{z}_t, r, t) - \frac{\Delta t}{t - r} \cdot \boldsymbol{v}_t - \frac{t - \Delta t - r}{t - r} \cdot \right. \right.$$
$$\left. \left. \left( \boldsymbol{u}_{\boldsymbol{\theta}^-}(\boldsymbol{z}_t, r, t) - \frac{\mathrm{d}\boldsymbol{u}_{\boldsymbol{\theta}^-}(\boldsymbol{z}_t, r, t)}{\mathrm{d}t} \Delta t + \mathcal{O}\left(\Delta^2 t\right) \right) \right\|_2^2 \right], \quad (12)$$

$$\stackrel{(ii)}{=} \mathbb{E}_{t,r,\boldsymbol{z}_t} \left[ \frac{t - r}{\Delta t} \cdot \left\| \boldsymbol{u}_{\boldsymbol{\theta}}(\boldsymbol{z}_t, r, t) - \boldsymbol{u}_{\boldsymbol{\theta}^-}(\boldsymbol{z}_t, r, t) - \frac{\Delta t}{t - r} \cdot \right. \right.$$
$$\left. \left. \left( \boldsymbol{v}_t - (t - r) \frac{\mathrm{d}\boldsymbol{u}_{\boldsymbol{\theta}^-}(\boldsymbol{z}_t, r, t)}{\mathrm{d}t} - \boldsymbol{u}_{\boldsymbol{\theta}^-}(\boldsymbol{z}_t, r, t) + \mathcal{O}\left(\Delta^2 t\right) \right) \right\|_2^2 \right],$$

where $(i)$ uses the Taylor expansion over $\boldsymbol{u}_{\boldsymbol{\theta}^-}(\boldsymbol{z}_{t-\Delta t}, r, t - \Delta t)$:

$$\boldsymbol{u}_{\boldsymbol{\theta}^-}(\boldsymbol{z}_{t-\Delta t}, r, t - \Delta t) = \boldsymbol{u}_{\boldsymbol{\theta}^-}(\boldsymbol{z}_t, r, t) - \frac{\mathrm{d}\boldsymbol{u}_{\boldsymbol{\theta}^-}(\boldsymbol{z}_t, r, t)}{\mathrm{d}t} \Delta t + \mathcal{O}\left(\Delta^2 t\right),$$

and $(ii)$ uses the fact that $\dfrac{\mathrm{d}\boldsymbol{u}_{\boldsymbol{\theta}^-}(\boldsymbol{z}_t, r, t)}{\mathrm{d}t} \Delta^2 t = \mathcal{O}\left(\Delta^2 t\right)$. Thus,

$$\lim_{\alpha \to 0} \nabla_{\boldsymbol{\theta}} \mathcal{L}_\alpha(\boldsymbol{\theta}) = \lim_{\Delta t \to 0} \nabla_{\boldsymbol{\theta}} \mathcal{L}_\alpha(\boldsymbol{\theta})$$

$$= \lim_{\Delta t \to 0} \mathbb{E}_{t,r,\boldsymbol{z}_t} \left[ 2 \cdot \frac{t - r}{\Delta t} \cdot \nabla_{\boldsymbol{\theta}}^\top \boldsymbol{u}_{\boldsymbol{\theta}}(\boldsymbol{z}_t, r, t) \cdot \left( \boldsymbol{u}_{\boldsymbol{\theta}}(\boldsymbol{z}_t, r, t) - \boldsymbol{u}_{\boldsymbol{\theta}^-}(\boldsymbol{z}_t, r, t) - \frac{\Delta t}{t - r} \cdot \right. \right.$$
$$\left. \left. \left( \boldsymbol{v}_t - (t - r) \frac{\mathrm{d}\boldsymbol{u}_{\boldsymbol{\theta}^-}(\boldsymbol{z}_t, r, t)}{\mathrm{d}t} - \boldsymbol{u}_{\boldsymbol{\theta}^-}(\boldsymbol{z}_t, r, t) + \mathcal{O}\left(\Delta^2 t\right) \right) \right) \right],$$

$$= \lim_{\Delta t \to 0} \mathbb{E}_{t,r,\boldsymbol{z}_t} \left[ 2 \cdot \frac{t - r}{\Delta t} \cdot \nabla_{\boldsymbol{\theta}}^\top \boldsymbol{u}_{\boldsymbol{\theta}}(\boldsymbol{z}_t, r, t) \cdot \left( -\frac{\Delta t}{t - r} \right) \cdot \right.$$
$$\left. \left( \boldsymbol{v}_t - (t - r) \frac{\mathrm{d}\boldsymbol{u}_{\boldsymbol{\theta}^-}(\boldsymbol{z}_t, r, t)}{\mathrm{d}t} - \boldsymbol{u}_{\boldsymbol{\theta}^-}(\boldsymbol{z}_t, r, t) + \mathcal{O}\left(\Delta^2 t\right) \right) \right],$$

$$= \mathbb{E}_{t,r,\boldsymbol{z}_t} \left[ 2 \cdot \left( \frac{t - r}{\Delta t} \cdot \frac{\Delta t}{t - r} \right) \cdot \nabla_{\boldsymbol{\theta}}^\top \boldsymbol{u}_{\boldsymbol{\theta}}(\boldsymbol{z}_t, r, t) \cdot \left( \boldsymbol{u}_{\boldsymbol{\theta}^-}(\boldsymbol{z}_t, r, t) - \boldsymbol{v}_t + (t - r) \frac{\mathrm{d}\boldsymbol{u}_{\boldsymbol{\theta}^-}(\boldsymbol{z}_t, r, t)}{\mathrm{d}t} \right) \right],$$

$$= \nabla_{\boldsymbol{\theta}} \mathcal{L}_{\mathrm{MF}}(\boldsymbol{\theta}),$$

$$(13)$$

$\square$

*Proof of equivalence with consistency model.* By setting $\tilde{\boldsymbol{v}}_{s,t} = \boldsymbol{v}_t, r = 0$ and $\boldsymbol{u_\theta}(\boldsymbol{z}_t, 0, t) = (\boldsymbol{z}_t - \boldsymbol{f_\theta}(\boldsymbol{z}_t, t))/t, \Delta t = t - s$ and $\alpha = \dfrac{\Delta t}{t}$, we have:

$$
\mathcal{L}_\alpha(\boldsymbol{\theta}) = \mathbb{E}_{t,r,\boldsymbol{z}_t} \left[ \frac{t}{\Delta t} \cdot \left\| \boldsymbol{u_\theta}(\boldsymbol{z}_t, r, t) - \frac{\Delta t}{t - r} \cdot \boldsymbol{v}_t - \right. \right.
$$
$$
\left. \left. \frac{t - \Delta t - r}{t - r} \boldsymbol{u_{\theta^-}}(\boldsymbol{z}_{t-\Delta t}, r, t - \Delta t) \right\|_2^2 \right],
$$
$$
\stackrel{(i)}{\Rightarrow} \mathbb{E}_{t,\boldsymbol{z}_t} \left[ \frac{t}{\Delta t} \cdot \left\| \frac{\boldsymbol{z}_t - \boldsymbol{f_\theta}(\boldsymbol{z}_t, t)}{t} - \frac{\Delta t}{t} \cdot \boldsymbol{v}_t - \right. \right. \tag{14}
$$
$$
\left. \left. \frac{t - \Delta t}{t} \frac{\boldsymbol{z}_{t-\Delta t} - \boldsymbol{f_{\theta^-}}(\boldsymbol{z}_{t-\Delta t}, t - \Delta t)}{t - \Delta t} \right\|_2^2 \right],
$$
$$
\stackrel{(ii)}{=} \mathbb{E}_{t,\boldsymbol{z}_t} \left[ \frac{1}{t\Delta t} \cdot \| \boldsymbol{f_\theta}(\boldsymbol{z}_t, t) - \boldsymbol{f_{\theta^-}}(\boldsymbol{z}_{t-\Delta t}, t - \Delta t) \|_2^2 \right] \stackrel{(iii)}{=} \mathcal{L}_{\mathrm{CT_d}}(\boldsymbol{\theta}),
$$

where $(i)$ plug in the reparameterization and $r = 0$, $(ii)$ uses the fact that $\boldsymbol{z}_t = \boldsymbol{z}_{t-\Delta t} + \Delta t \cdot \boldsymbol{v}_t$. Thus $\mathcal{L}_\alpha(\boldsymbol{\theta})$ could be reparameterized to $\mathcal{L}_{\mathrm{CT_d}}(\boldsymbol{\theta})$ with a loss weighting function $\dfrac{1}{t\Delta t}$. Since the discrete CT uses timestep partition to determine $t$ and $\Delta t$, the $(iii)$ holds for a special timestep partition when $\Delta t = \alpha \cdot t$ given a fixed $\alpha$. From Theorem 2 in Lipman et al. (2023), because $\boldsymbol{u_{\theta^-}}(\boldsymbol{z}_t, r, t)$ is independent of $\boldsymbol{\theta}$, we have:

$$
\mathcal{L}_{\mathrm{MF}}(\boldsymbol{\theta}) = \mathbb{E}_{t,r,\boldsymbol{z}_t} \left[ \left\| \boldsymbol{u_\theta}(\boldsymbol{z}_t, r, t) - \boldsymbol{v}_t + (t - r) \frac{\mathrm{d}\boldsymbol{u_{\theta^-}}(\boldsymbol{z}_t, r, t)}{\mathrm{d}t} \right\|_2^2 \right],
$$
$$
= \mathbb{E}_{t,r,\boldsymbol{z}_t} \left[ \left\| \boldsymbol{u_\theta}(\boldsymbol{z}_t, r, t) - \boldsymbol{v}(\boldsymbol{z}_t, t) + (t - r) \frac{\mathrm{d}\boldsymbol{u_{\theta^-}}(\boldsymbol{z}_t, r, t)}{\mathrm{d}t} \right\|_2^2 \right] + C, \tag{15}
$$

with $C$ a constant independent of $\boldsymbol{\theta}$. Thus,

$$
\nabla_{\boldsymbol{\theta}} \mathcal{L}_{\mathrm{MF}}(\boldsymbol{\theta})
$$
$$
= \mathbb{E}_{t,r,\boldsymbol{z}_t} \left[ 2 \cdot \nabla_{\boldsymbol{\theta}}^\top \boldsymbol{u_\theta}(\boldsymbol{z}_t, r, t) \cdot \left( \boldsymbol{u_\theta}(\boldsymbol{z}_t, r, t) - \boldsymbol{v}(\boldsymbol{z}_t, t) + (t - r) \frac{\mathrm{d}\boldsymbol{u_{\theta^-}}(\boldsymbol{z}_t, r, t)}{\mathrm{d}t} \right) \right],
$$
$$
\stackrel{(i)}{\Rightarrow} \mathbb{E}_{t,\boldsymbol{z}_t} \left[ 2 \cdot \nabla_{\boldsymbol{\theta}}^\top \frac{-\boldsymbol{f_\theta}(\boldsymbol{z}_t, t)}{t} \cdot \left( \frac{\boldsymbol{z}_t - \boldsymbol{f_\theta}(\boldsymbol{z}_t, t)}{t} - \boldsymbol{v}(\boldsymbol{z}_t, t) + \right. \right. \tag{16}
$$
$$
\left. \left. \left( \boldsymbol{v}(\boldsymbol{z}_t, t) - \frac{\mathrm{d}\boldsymbol{f_{\theta^-}}(\boldsymbol{z}_t, t)}{\mathrm{d}t} \right) - \frac{(\boldsymbol{z}_t - \boldsymbol{f_{\theta^-}}(\boldsymbol{z}_t, t))}{t} \right) \right],
$$
$$
= \mathbb{E}_{t,\boldsymbol{z}_t} \left[ 2 \cdot \frac{1}{t} \cdot \nabla_{\boldsymbol{\theta}}^\top \boldsymbol{f_\theta}(\boldsymbol{z}_t, t) \frac{\mathrm{d}\boldsymbol{f_{\theta^-}}(\boldsymbol{z}_t, t)}{\mathrm{d}t} \right] = \nabla_{\boldsymbol{\theta}} \mathcal{L}_{\mathrm{CT_c}}(\boldsymbol{\theta}),
$$

where $(i)$ plug in the reparameterization and $r = 0$, and use the fact that:

$$
\frac{\mathrm{d}\boldsymbol{u_{\theta^-}}(\boldsymbol{z}_t, r = 0, t)}{\mathrm{d}t} = \frac{1}{t^2} \left( t \left( \boldsymbol{v}(\boldsymbol{z}_t, t) - \frac{\mathrm{d}\boldsymbol{f_{\theta^-}}(\boldsymbol{z}_t, t)}{\mathrm{d}t} \right) - (\boldsymbol{z}_t - \boldsymbol{f_{\theta^-}}(\boldsymbol{z}_t, t)) \right) \tag{17}
$$

Thus $\nabla_{\boldsymbol{\theta}} \mathcal{L}_{\mathrm{MF}}(\boldsymbol{\theta})$ could be reparameterized to $\mathcal{L}_{\mathrm{CT_c}}(\boldsymbol{\theta})$ with a loss weighting function $\dfrac{1}{t}$. $\qquad \square$

## E  MORE PROOFS

### E.1  ANALYSIS OF OPTIMAL SOLUTION SPACE BETWEEN TRAJECTORY FLOW MATCHING $\mathcal{L}_{\mathrm{TFM}}$ TRAJECTORY CONSISTENCY $\mathcal{L}_{\mathrm{TC}}$

**Assumption 1.** *Assume that the $\boldsymbol{u_\theta}(\boldsymbol{z}_t, r, t)$ has infinite model capacity, and can approximate any continuous function to an arbitrary level of accuracy based on the Universal Approximation Theorem. And $\boldsymbol{z}_0 = \boldsymbol{x} \in \mathbb{R}^d$ draw from a random distribution $p(\boldsymbol{x})$.*

Assumption 1 has been widely used in previous works Karras et al. (2022); Gu et al. (2023); Zhang et al. (2024) to analysis the optimal solution of diffusion training loss, which simplified specific network architecture constraint without loss of generality. The trajectory flow matching loss defined in Equation (6) could be re-written as:

$$\mathcal{L}_{\text{TFM}}(\boldsymbol{u_\theta}) = \underset{t,r,\boldsymbol{z_t}}{\mathbb{E}} \left[ \|\boldsymbol{u_\theta}(\boldsymbol{z_t}, r, t) - \boldsymbol{v_t}\|_2^2 \right], \tag{18}$$

$$= \int_{t,r} \int_{\mathbb{R}^d} \underbrace{\int_{\mathbb{R}^d} \mathcal{N}\left(\boldsymbol{z_t}; (1-t)\boldsymbol{x}, t^2\boldsymbol{I}\right) \|\boldsymbol{u_\theta}(\boldsymbol{z_t}, r, t) - \frac{\boldsymbol{z_t} - \boldsymbol{x}}{t}\|_2^2 \cdot p(\boldsymbol{x})\mathrm{d}\boldsymbol{x}}_{\mathcal{L}_{\text{TFM}}(\boldsymbol{u_\theta}, t, r, \boldsymbol{z_t})} \ \mathrm{d}\boldsymbol{z_t} \ \mathrm{d}t \ \mathrm{d}r. $$

$$\tag{19}$$

Since Assumption 1 implies that $\boldsymbol{u_\theta}(\boldsymbol{z_t}, r, t)$ can approximate any continuous function, we can minimize $\mathcal{L}_{\text{TFM}}(\boldsymbol{u_\theta})$ by independently minimizing the inner integral $\mathcal{L}_{\text{TFM}}(\boldsymbol{u_\theta}, t, r, \boldsymbol{z_t})$ for each fixed $(\boldsymbol{z_t}, r, t)$.

Given $\boldsymbol{z_t}, r, t$, $\mathcal{L}_{\text{TFM}}(\boldsymbol{u_\theta}, t, r, \boldsymbol{z_t})$ is a convex optimization problem with respect to $\boldsymbol{u_\theta}(\boldsymbol{z_t}, r, t)$ and the optimal solution is uniquely identified by setting the gradient with respect to $\boldsymbol{u_\theta}(\boldsymbol{z_t}, r, t)$ to 0.

$$\nabla_{\boldsymbol{u_\theta}(\boldsymbol{z_t}, r, t)} \mathcal{L}_{\text{TFM}}(\boldsymbol{u_\theta}, t, r, \boldsymbol{z_t}) = 0, \tag{20}$$

$$\Rightarrow \nabla_{\boldsymbol{u_\theta}(\boldsymbol{z_t}, r, t)} \int_{\mathbb{R}^d} \mathcal{N}\left(\boldsymbol{z_t}; (1-t)\boldsymbol{x}, t^2\boldsymbol{I}\right) \|\boldsymbol{u_\theta}(\boldsymbol{z_t}, r, t) - \frac{\boldsymbol{z_t} - \boldsymbol{x}}{t}\|_2^2 \cdot p(\boldsymbol{x})\mathrm{d}\boldsymbol{x} = 0, \tag{21}$$

$$\Rightarrow \int_{\mathbb{R}^d} \mathcal{N}\left(\boldsymbol{z_t}; (1-t)\boldsymbol{x}, t^2\boldsymbol{I}\right) \left( \boldsymbol{u}^*_{\boldsymbol{\theta}, \text{TFM}}(\boldsymbol{z_t}, r, t) - \frac{\boldsymbol{z_t} - \boldsymbol{x}}{t} \right) \cdot p(\boldsymbol{x})\mathrm{d}\boldsymbol{x} = 0, \tag{22}$$

$$\Rightarrow \boldsymbol{u}^*_{\boldsymbol{\theta}, \text{TFM}}(\boldsymbol{z_t}, r, t) = \frac{1}{t} \left( \boldsymbol{z_t} - \frac{\mathbb{E}_{\boldsymbol{x} \sim p(\boldsymbol{x})}\left[ \mathcal{N}\left(\boldsymbol{z_t}; (1-t)\boldsymbol{x}, t^2\boldsymbol{I}\right) \cdot \boldsymbol{x} \right]}{\mathbb{E}_{\boldsymbol{x} \sim p(\boldsymbol{x})}\left[ \mathcal{N}\left(\boldsymbol{z_t}; (1-t)\boldsymbol{x}, t^2\boldsymbol{I}\right) \right]} \right) = \boldsymbol{u}^*_{\boldsymbol{\theta}, \text{TFM}}(\boldsymbol{z_t}, t). \tag{23}$$

This shows that the loss $\mathcal{L}_{\text{TFM}}$ has only one optimal solution. However, the situation is different for the Trajectory consistency $\mathcal{L}_{\text{TC}}$ defined in Equation (6):

$$\mathcal{L}_{\text{TC}} = \underset{t,r,\boldsymbol{z_t}}{\mathbb{E}} \left[ 2(t-r) \cdot \boldsymbol{u}_{\boldsymbol{\theta}}^\top(\boldsymbol{z_t}, r, t) \frac{\mathrm{d}\boldsymbol{u}_{\boldsymbol{\theta}^-}(\boldsymbol{z_t}, r, t)}{\mathrm{d}t} \right]. \tag{24}$$

For fixed $\boldsymbol{z_t}, r, t$, the loss $\mathcal{L}_{\text{TC}}(\boldsymbol{z_t}, r, t)$ is a linear function w.r.t $\boldsymbol{u_\theta}(\boldsymbol{z_t}, r, t)$, so it does not have a lower bound. Any function $\boldsymbol{u_\theta}(\boldsymbol{z_t}, r, t)$ which satisfies the condition $\boldsymbol{u}_{\boldsymbol{\theta}}^\top(\boldsymbol{z_t}, r, t) \frac{\mathrm{d}\boldsymbol{u}_{\boldsymbol{\theta}^-}(\boldsymbol{z_t}, r, t)}{\mathrm{d}t} \to -\infty$ minimizes the loss. This confirms that the unconstrained $\mathcal{L}_{\text{TC}}$ has a large, easily optimizable solution space.

Crucially, this finding does not contradict the difficulty of optimizing $\mathcal{L}_{\text{CT}_c}$ noted in prior work Song et al. (2023); Lu & Song (2025). In those papers, the optimization problem was made difficult by the explicit boundary condition $\boldsymbol{f_\theta}(\boldsymbol{z_0}, 0) = \boldsymbol{z_0}$, which constrains the velocity field as $\boldsymbol{u_\theta}(\boldsymbol{z_t}, 0, t) = (\boldsymbol{z_t} - \boldsymbol{f_\theta}(\boldsymbol{z_t}, t)) / t = \hat{\boldsymbol{z}}_0$.

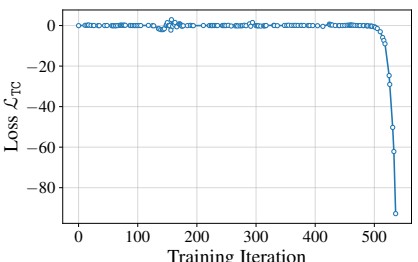

Figure 8: Optimization of $\mathcal{L}_{\text{TC}}$.

### E.2 ANALYSIS OF OPTIMAL SOLUTION BETWEEN TRAJECTORY FLOW MATCHING $\mathcal{L}_{\text{TFM}}$ TRAJECTORY CONSISTENCY $\mathcal{L}_{\text{TC}}$

Following the same derivation as above, minimizing $\mathcal{L}_{\text{MF}}(\boldsymbol{\theta})$ is equivalent to minimizing $\mathcal{L}_{\text{MF}}(\boldsymbol{\theta}, \boldsymbol{z_t}, r, t)$ for every fixed $\boldsymbol{z_t}, r, t$ under Assumption 1, where:

$$\mathcal{L}_{\text{MF}}(\boldsymbol{\theta}, \boldsymbol{z_t}, r, t)$$
$$= \int_{\mathbb{R}^d} \mathcal{N}\left(\boldsymbol{z_t}; (1-t)\boldsymbol{x}, t^2\boldsymbol{I}\right) \|\boldsymbol{u_\theta}(\boldsymbol{z_t}, r, t) - \boldsymbol{v_t} + (t-r) \frac{\mathrm{d}\boldsymbol{u}_{\boldsymbol{\theta}^-}(\boldsymbol{z_t}, r, t)}{\mathrm{d}t}\|_2^2 \cdot p(\boldsymbol{x})\mathrm{d}\boldsymbol{x}. \tag{25}$$

Since the term $u_{\theta-}$ is independent of $u_{\theta}$, $\mathcal{L}_{\mathrm{MF}}(\theta, z_t, r, t)$ remains a convex optimization problem over $u_{\theta}$. Consequently, for all $z_t, r, t$, the optimum satisfies:

$$\nabla_{u_{\theta}(z_t,r,t)} \mathcal{L}_{\mathrm{MF}}(\theta, z_t, r, t) = 0, \tag{26}$$

$$\Rightarrow \int_{\mathbb{R}^d} \mathcal{N}\left(z_t; (1-t)x, t^2 I\right) \left(u^*_{\theta,\mathrm{MF}}(z_t, r, t) - \frac{z_t - x}{t} + (t-r)\frac{\mathrm{d}u^*_{\theta,\mathrm{MF}}(z_t, r, t)}{\mathrm{d}t}\right) \cdot p(x)\mathrm{d}x = 0, \tag{27}$$

$$\Rightarrow u^*_{\theta,\mathrm{MF}}(z_t, r, t) + (t-r)\frac{\mathrm{d}u^*_{\theta,\mathrm{MF}}(z_t, r, t)}{\mathrm{d}t} = u^*_{\theta,\mathrm{TFM}}(z_t, t), \tag{28}$$

$$\Rightarrow u^*_{\theta,\mathrm{MF}}(z_t, r, t) = \frac{1}{t-r}\int_r^t u^*_{\theta,\mathrm{TFM}}(z_t, t)\mathrm{d}t. \tag{29}$$

As detailed in Geng et al. (2025a), the constant term in the last equation is zero. Furthermore, to minimize the $\mathcal{L}_{\alpha}(\theta)$ loss with $\tilde{v}_{s,t} = v_t$, the same assumption in Assumption 1 makes it equivalent to minimizing $\mathcal{L}_{\alpha}(\theta, z_t, r, t)$ for every fixed $z_t, r, t$, where

$$\mathcal{L}_{\alpha}(\theta, z_t, r, t) = \alpha^{-1}\int_{\mathbb{R}^d} \mathcal{N}\left(z_t; (1-t)x, t^2 I\right) ||u_{\theta}(z_t, r, t) -$$
$$(\alpha \cdot v_t + (1-\alpha)\cdot u_{\theta-}(z_s, r, s))||_2^2 \cdot p(x)\mathrm{d}x.$$

Applying the same zero-gradient condition, the optimal solution of $u^*_{\theta,\alpha}(z_t, r, t)$ satisfies:

$$u^*_{\theta,\alpha}(z_t, r, t) - (1-\alpha)\cdot u^*_{\theta,\alpha}(z_s, r, s) = \alpha \cdot u^*_{\theta,\mathrm{TFM}}(z_t, t).$$

Thus, when $\alpha = 1$:

$$u^*_{\theta,\alpha}(z_t, r, t) = u^*_{\theta,\mathrm{TFM}}(z_t, t), \tag{30}$$

when $\alpha < 1$:

$$u^*_{\theta,\alpha}(z_t, r, t) = \alpha \cdot u^*_{\theta,\mathrm{TFM}}(z_t, t) + (1-\alpha)\cdot u^*_{\theta,\alpha}(z_s, r, s), \tag{31}$$

$$= \alpha \cdot u^*_{\theta,\mathrm{TFM}}(z_t, t) + (1-\alpha)\alpha \cdot u^*_{\theta,\mathrm{TFM}}(z_t, s) + (1-\alpha)^2 \cdot u^*_{\theta,\alpha}(z_{t_k}, r, t_k), \tag{32}$$

$$= ..., \tag{33}$$

$$= \alpha \sum_{k=0}^{\infty}(1-\alpha)^k u^*_{\theta,\mathrm{TFM}}(z_{t_k}, t_k), \tag{34}$$

where $t_k = r + (1-\alpha)^k(t-r)$. Let $\lambda_k := (1-\alpha)^k$, thus $\Delta\lambda_k = \lambda_k - \lambda_{k+1} = \alpha(1-\alpha)^k$, so:

$$u^*_{\theta,\alpha}(z_t, r, t) = \sum_{k=0}^{\infty}(\lambda_k - \lambda_{k+1})\, u^*_{\theta,\mathrm{TFM}}(z_{r+\lambda_k(t-r)}, r + \lambda_k(t-r)), \tag{35}$$

$$= \frac{1}{t-r}\sum_{k=0}^{\infty}(\tau_k - \tau_{k+1})\, u^*_{\theta,\mathrm{TFM}}(z_{\tau_k}, \tau_k), \tag{36}$$

where $\tau_k = r + \lambda_k(t-r)$. This is a left Riemann sum, and as $\alpha \to 0$, $\Delta\lambda_k \to 0$ and:

$$\lim_{\alpha\to 0} u^*_{\theta,\alpha}(z_t, r, t) = \frac{1}{t-r}\int_r^t u^*_{\theta,\mathrm{TFM}}(z_t, t)\mathrm{d}t = u^*_{\theta,\mathrm{MF}}(z_t, r, t). \tag{37}$$

**Assumption 2.** *The optimal velocity function $\boldsymbol{u}^*_{\boldsymbol{\theta},\mathrm{TFM}}(\boldsymbol{z}_t, t)$ and the parametric velocity function $\boldsymbol{u}_{\boldsymbol{\theta}}(\boldsymbol{z}_t, r, t)$ satisfy the following properties:*

- *Lipschitz Continuity of Optimal Velocity: There exists a constant $L_1 > 0$ such that for all $t_1, t_2 \in [0, 1]$ and corresponding $\boldsymbol{z}_{t_1}, \boldsymbol{z}_{t_2}$,*

$$||\boldsymbol{u}^*_{\boldsymbol{\theta},\mathrm{TFM}}(\boldsymbol{z}_{t_1}, t_1) - \boldsymbol{u}^*_{\boldsymbol{\theta},\mathrm{TFM}}(\boldsymbol{z}_{t_2}, t_2)||_2 \leq L_1 |t_1 - t_2|. \tag{38}$$

- *Lipschitz Continuity of Time Derivative: There exists a constant $L_2 > 0$ such that for all $t_1, t_2 \in [r, 1]$ (where $r \in [0, 1]$) and corresponding $\boldsymbol{z}_{t_1}, \boldsymbol{z}_{t_2}$,*

$$\left\| \frac{\mathrm{d}\boldsymbol{u}_{\boldsymbol{\theta}}(\boldsymbol{z}_{t_1}, r, t_1)}{\mathrm{d}t} - \frac{\mathrm{d}\boldsymbol{u}_{\boldsymbol{\theta}}(\boldsymbol{z}_{t_2}, r, t_2)}{\mathrm{d}t} \right\|_2 \leq L_2 |t_1 - t_2|. \tag{39}$$

- *Boundedness of Gradient and Time Derivative: There exist constants $C_1, C_2 > 0$ such that for all $r, t \in [0, 1]$ with $r \leq t$, and for the corresponding $\boldsymbol{z}_t$,*

$$||\nabla^{\top}_{\boldsymbol{\theta}} \boldsymbol{u}_{\boldsymbol{\theta}}(\boldsymbol{z}_t, r, t)||_2 \leq C_1, \quad and \quad \left\| \frac{\mathrm{d}\boldsymbol{u}_{\boldsymbol{\theta}}(\boldsymbol{z}_t, r, t)}{\mathrm{d}t} \right\|_2 \leq C_2. \tag{40}$$

The Lipschitz continuity of the score function is a common assumption widely adopted in the theoretical analysis of score functions in diffusion models Chen et al. (2022; 2023). Here, we assume the Lipschitz continuity of the optimal velocity function $\boldsymbol{u}^*_{\boldsymbol{\theta},\mathrm{TFM}}(\boldsymbol{z}_t, t)$ and the derivative of $\boldsymbol{u}_{\boldsymbol{\theta}}$ over $t$, $\frac{\mathrm{d}\boldsymbol{u}_{\boldsymbol{\theta}}(\boldsymbol{z}_t, r, t)}{\mathrm{d}t}$. The boundedness of $\left\| \nabla^{\top}_{\boldsymbol{\theta}} \boldsymbol{u}_{\boldsymbol{\theta}}(\boldsymbol{z}_t, r, t) \right\|_2$ holds in reality as we using gradient clip over each element in $\nabla^{\top}_{\boldsymbol{\theta}} \boldsymbol{u}_{\boldsymbol{\theta}}(\boldsymbol{z}_t, r, t)$ to stablize training. Thus, under a finite $\alpha$:

$$||\boldsymbol{u}^*_{\boldsymbol{\theta},\alpha}(\boldsymbol{z}_t, r, t) - \boldsymbol{u}^*_{\boldsymbol{\theta},\mathrm{MF}}(\boldsymbol{z}_t, r, t)||_2 = \left\| \frac{1}{t - r} \sum_{k=0}^{\infty} \int_{\tau_{k+1}}^{\tau_k} \left( \boldsymbol{u}^*_{\boldsymbol{\theta},\mathrm{TFM}}(\boldsymbol{z}_{\tau_k}, \tau_k) - \boldsymbol{u}^*_{\boldsymbol{\theta},\mathrm{TFM}}(\boldsymbol{z}_{\tau}, \tau) \right) \mathrm{d}\tau \right\|_2, \tag{41}$$

$$\leq \frac{1}{t - r} \sum_{k=0}^{\infty} \int_{\tau_{k+1}}^{\tau_k} \left\| \boldsymbol{u}^*_{\boldsymbol{\theta},\mathrm{TFM}}(\boldsymbol{z}_{\tau_k}, \tau_k) - \boldsymbol{u}^*_{\boldsymbol{\theta},\mathrm{TFM}}(\boldsymbol{z}_{\tau}, \tau) \right\|_2 \mathrm{d}\tau, \tag{42}$$

$$\leq \frac{1}{t - r} \sum_{k=0}^{\infty} \int_{\tau_{k+1}}^{\tau_k} L_1 (\tau_k - \tau) \, \mathrm{d}\tau, \tag{43}$$

$$= \frac{1}{t - r} \sum_{k=0}^{\infty} \frac{L_1}{2} (\tau_k - \tau_{k+1})^2, \tag{44}$$

$$= \frac{L_1(t - r)}{2} \alpha^2 \sum_{k=0}^{\infty} (1 - \alpha)^{2k}, \tag{45}$$

$$= \frac{L_1(t - r)}{2} \cdot \frac{\alpha}{2 - \alpha}, \tag{46}$$

$||\boldsymbol{u}^*_{\boldsymbol{\theta},\alpha}(\boldsymbol{z}_t, r, t) - \boldsymbol{u}^*_{\boldsymbol{\theta},\mathrm{MF}}(\boldsymbol{z}_t, r, t)||_2$ decreases with $\alpha$ and $\lim_{\alpha \to 0} ||\boldsymbol{u}^*_{\boldsymbol{\theta},\alpha}(\boldsymbol{z}_t, r, t) - \boldsymbol{u}^*_{\boldsymbol{\theta},\mathrm{MF}}(\boldsymbol{z}_t, r, t)||_2 = 0$; hence the two optimal solution coincide as $\alpha \to 0$, which align with our previous proof.

### E.3 NON-SYMPTOTIC DISTANCE BETWEEN GRADIENTS FROM $\alpha$-FLOW AND MEANFLOW

From Equation (8), when $\tilde{\boldsymbol{v}}_{s,t} = \boldsymbol{v}_t$ the gradient of $\alpha$-Flow loss is:

$$\nabla_{\boldsymbol{\theta}}\mathcal{L}_{\alpha}(\boldsymbol{\theta}) = \mathop{\mathbb{E}}_{t,r,\boldsymbol{z}_t} \left[ \alpha^{-1} \cdot \nabla_{\boldsymbol{\theta}}^{\top} \boldsymbol{u_{\theta}}(\boldsymbol{z}_t, r, t) \cdot (\boldsymbol{u_{\theta}}(\boldsymbol{z}_t, r, t) - (\alpha \cdot \boldsymbol{v}_t + (1-\alpha) \cdot \boldsymbol{u_{\theta^-}}(\boldsymbol{z}_s, r, s))) \right],$$

$$= \mathop{\mathbb{E}}_{t,r,\boldsymbol{z}_t} \left[ \nabla_{\boldsymbol{\theta}}^{\top} \boldsymbol{u_{\theta}}(\boldsymbol{z}_t, r, t) \cdot (\boldsymbol{u_{\theta}}(\boldsymbol{z}_t, r, t) - \boldsymbol{v}_t) \right]$$

$$\mathop{\mathbb{E}}_{t,r,\boldsymbol{z}_t} \left[ \frac{1-\alpha}{\alpha} \cdot \nabla_{\boldsymbol{\theta}}^{\top} \boldsymbol{u_{\theta}}(\boldsymbol{z}_t, r, t) \cdot (\boldsymbol{u_{\theta}}(\boldsymbol{z}_t, r, t) - \boldsymbol{u_{\theta^-}}(\boldsymbol{z}_s, r, s)) \right],$$

$$= \nabla_{\boldsymbol{\theta}}\mathcal{L}_{\text{TFM}}(\boldsymbol{\theta}) + \mathop{\mathbb{E}}_{t,r,\boldsymbol{z}_t} \left[ (1-\alpha) \cdot \nabla_{\boldsymbol{\theta}}^{\top} \boldsymbol{u_{\theta}}(\boldsymbol{z}_t, r, t) \cdot \frac{1}{\alpha} \int_{s}^{t} \frac{\mathrm{d}\boldsymbol{u_{\theta^-}}(\boldsymbol{z}_\tau, r, \tau)}{\mathrm{d}t} \mathrm{d}\tau \right]. \tag{47}$$

And the gradient of MeanFlow loss could be written as:

$$\nabla_{\boldsymbol{\theta}}\mathcal{L}_{\text{MF}}(\boldsymbol{\theta}) = \mathbb{E}_{t,r,\boldsymbol{z}_t} \left[ \nabla_{\boldsymbol{\theta}}^{\top} \boldsymbol{u_{\theta}}(\boldsymbol{z}_t, r, t) \cdot \left( \boldsymbol{u_{\theta}}(\boldsymbol{z}_t, r, t) - \boldsymbol{v}_t + (t-r)\frac{\mathrm{d}\boldsymbol{u_{\theta^-}}(\boldsymbol{z}_t, r, t)}{\mathrm{d}t} \right) \right]$$

$$= \nabla_{\boldsymbol{\theta}}\mathcal{L}_{\text{TFM}}(\boldsymbol{\theta}) + \mathop{\mathbb{E}}_{t,r,\boldsymbol{z}_t} \left[ (t-r) \cdot \nabla_{\boldsymbol{\theta}}^{\top} \boldsymbol{u_{\theta}}(\boldsymbol{z}_t, r, t) \cdot \frac{\mathrm{d}\boldsymbol{u_{\theta^-}}(\boldsymbol{z}_t, r, t)}{\mathrm{d}t} \right] \tag{48}$$

Thus, the $\ell_2$ distance between $\nabla_{\boldsymbol{\theta}}\mathcal{L}_{\alpha}(\boldsymbol{\theta})$ and $\nabla_{\boldsymbol{\theta}}\mathcal{L}_{\text{MF}}(\boldsymbol{\theta})$ is:

$$\|\nabla_{\boldsymbol{\theta}}\mathcal{L}_{\alpha}(\boldsymbol{\theta}) - \nabla_{\boldsymbol{\theta}}\mathcal{L}_{\text{MF}}(\boldsymbol{\theta})\|_2$$

$$= \left\| \mathop{\mathbb{E}}_{t,r,\boldsymbol{z}_t} \left[ \nabla_{\boldsymbol{\theta}}^{\top} \boldsymbol{u_{\theta}}(\boldsymbol{z}_t, r, t) \frac{1}{\alpha} \int_{s}^{t} \left( (1-\alpha)\frac{\mathrm{d}\boldsymbol{u_{\theta^-}}(\boldsymbol{z}_\tau, r, \tau)}{\mathrm{d}t} - \frac{\mathrm{d}\boldsymbol{u_{\theta^-}}(\boldsymbol{z}_t, r, t)}{\mathrm{d}t} \right) \mathrm{d}\tau \right] \right\|_2$$

$$\leq \|\nabla_{\boldsymbol{\theta}}^{\top} \boldsymbol{u_{\theta}}(\boldsymbol{z}_t, r, t)\|_2 \cdot \mathop{\mathbb{E}}_{t,r,\boldsymbol{z}_t} \left[ \frac{1}{\alpha} \int_{s}^{t} \left\| \frac{\mathrm{d}\boldsymbol{u_{\theta^-}}(\boldsymbol{z}_\tau, r, \tau)}{\mathrm{d}t} - \frac{\mathrm{d}\boldsymbol{u_{\theta^-}}(\boldsymbol{z}_t, r, t)}{\mathrm{d}t} \right\|_2 \mathrm{d}\tau + \int_{s}^{t} \left\| \frac{\mathrm{d}\boldsymbol{u_{\theta^-}}(\boldsymbol{z}_\tau, r, \tau)}{\mathrm{d}t} \right\|_2 \mathrm{d}\tau \right]$$

$$\leq C_1 \cdot \mathop{\mathbb{E}}_{t,r,\boldsymbol{z}_t} \left[ \frac{1}{\alpha} \int_{s}^{t} L_2(t-\tau)\mathrm{d}\tau + \alpha(t-r)C_2 \right]$$

$$= \alpha \cdot C_1 \mathbb{E}_{t,r} \left[ \frac{1}{2}L_2(t-r)^2 + C_2(t-r) \right] \tag{49}$$

Because $t, r \in [0, 1]$, $\mathbb{E}_{t,r}\left[\frac{1}{2}L_2(t-r)^2 + C_2(t-r)\right]$ will be a finite constant no matter the choice of distribution of $r$ and $t$, . Thus the upper bound of $\|\nabla_{\boldsymbol{\theta}}\mathcal{L}_{\alpha}(\boldsymbol{\theta}) - \nabla_{\boldsymbol{\theta}}\mathcal{L}_{\text{MF}}(\boldsymbol{\theta})\|_2$ is linearly depend on $\alpha$, and will vanish to 0 when $\alpha \to 0$, align with our Theorem 1 that $\nabla_{\boldsymbol{\theta}}\mathcal{L}_{\text{MF}}(\boldsymbol{\theta}) = \nabla_{\boldsymbol{\theta}}\mathcal{L}_{\alpha \to 0}(\boldsymbol{\theta})$.

### E.4 VARIANCE OF $\alpha$-FLOW GRADIENT

Let $X \triangleq \nabla_{\boldsymbol{\theta}}^{\top} \boldsymbol{u_{\theta}}(\boldsymbol{z}_t, r, t) \cdot (\boldsymbol{u_{\theta}}(\boldsymbol{z}_t, r, t) - \boldsymbol{v}_t)$ and $Y \triangleq \nabla_{\boldsymbol{\theta}}^{\top} \boldsymbol{u_{\theta}}(\boldsymbol{z}_t, r, t) \cdot \frac{1}{\alpha} \int_{s}^{t} \frac{\mathrm{d}\boldsymbol{u_{\theta^-}}(\boldsymbol{z}_\tau, r, \tau)}{\mathrm{d}t} \mathrm{d}\tau$.

$$\mathbb{V}\left[\nabla_{\boldsymbol{\theta}}\mathcal{L}_{\alpha}(\boldsymbol{\theta})\right]$$

$$= \mathop{\mathbb{V}}_{t,r,\boldsymbol{z}_t}\left[\alpha^{-1}\cdot\nabla_{\boldsymbol{\theta}}^{\top}\boldsymbol{u}_{\boldsymbol{\theta}}(\boldsymbol{z}_t,r,t)\cdot(\boldsymbol{u}_{\boldsymbol{\theta}}(\boldsymbol{z}_t,r,t)-(\alpha\cdot\boldsymbol{v}_t+(1-\alpha)\cdot\boldsymbol{u}_{\boldsymbol{\theta}^-}(\boldsymbol{z}_s,r,s)))\right],$$

$$= \mathop{\mathbb{V}}_{t,r,\boldsymbol{z}_t}\left[X+(1-\alpha)Y\right],$$

$$\leq \left(\sqrt{\mathop{\mathbb{V}}_{t,r,\boldsymbol{z}_t}[X]}+(1-\alpha)\sqrt{\mathop{\mathbb{V}}_{t,r,\boldsymbol{z}_t}[Y]}\right)^2,$$

$$\leq \left(\sqrt{\mathbb{V}[\nabla_{\boldsymbol{\theta}}\mathcal{L}_{\mathrm{TFM}}(\boldsymbol{\theta})]}+(1-\alpha)\sqrt{\mathop{\mathbb{E}}_{t,r,\boldsymbol{z}_t}\left[\|Y\|_2^2\right]}\right)^2,$$

$$\leq \left(\sqrt{\mathbb{V}[\nabla_{\boldsymbol{\theta}}\mathcal{L}_{\mathrm{TFM}}(\boldsymbol{\theta})]}+(1-\alpha)\sqrt{\mathop{\mathbb{E}}_{t,r,\boldsymbol{z}_t}\left[\left\|\nabla_{\boldsymbol{\theta}}^{\top}\boldsymbol{u}_{\boldsymbol{\theta}}(\boldsymbol{z}_t,r,t)\right\|_2^2\cdot\left\|\frac{1}{\alpha}\int_s^t\frac{\mathrm{d}\boldsymbol{u}_{\boldsymbol{\theta}^-}(\boldsymbol{z}_\tau,r,\tau)}{\mathrm{d}t}\mathrm{d}\tau\right\|_2^2\right]}\right)^2,$$

$$\leq \left(\sqrt{\mathbb{V}[\nabla_{\boldsymbol{\theta}}\mathcal{L}_{\mathrm{TFM}}(\boldsymbol{\theta})]}+(1-\alpha)\cdot C_1\cdot C_2\cdot\sqrt{\mathbb{E}_{t,r}\left[(t-r)^2\right]}\right)^2,$$

Thus, the upper bound of the variance of $\nabla_{\boldsymbol{\theta}}\mathcal{L}_{\mathrm{MF}}(\boldsymbol{\theta})$ increases as $\alpha$ approaches 0.

## F    ANALYSIS DETAILS

The detailed implementation of DiT-B/2 is provided in Table 3, where we adopt the DiT-B/2-non-cfg setting. For loss evaluation, at each checkpoint we use a batch size of 128 and run 1000 iterations to compute the mean loss along with its 5% and 95% percentiles, which are reported in the figure. To measure the cosine similarity between different losses, we calculate $\nabla\mathcal{L}_{\mathrm{TFM}}$, $\nabla\mathcal{L}_{\mathrm{FM'}}$, and $\nabla\mathcal{L}_{\mathrm{TC}}$ on the same batch and then compute their pairwise cosine similarities, which is defined as:

$$\mathtt{cosine\ similarity}\,(\boldsymbol{a},\boldsymbol{b})=\frac{\boldsymbol{a}\cdot\boldsymbol{b}}{\|\boldsymbol{a}\|\,\|\boldsymbol{b}\|},$$

given two vectors $\boldsymbol{a},\boldsymbol{b}$. This procedure is also repeated over 1000 iterations to obtain the mean similarity and its 5% and 95% percentiles, as shown in the figure.

## G    IMPLEMENTATION DETAILS

Implementation details for models trained on ImageNet-1K $256^2$ are shown in Table 3. For Kinetics-700 $17\times256^2$, we employ a DiT-B/2 architecture trained for 600K iterations, with and without CFG. For CFG training, we set guidance scale $\omega=1.0$ and classifier weight $\kappa=0.7$.

## H    ADDITIONAL EXPERIMENTS

### H.1    ABLATION STUDY OVER BATCH SIZE

Training diffusion/flow-based models can be challenging due to the high variance of their gradients. Past research Zhou et al. (2025); Karras et al. (2022; 2024) often used large batch sizes (1024 or even 4096) to mitigate this issue. In this section, we fine-tune a MeanFlow-XL/2 model (with implementation details in Table 3) for an additional 60 epochs using a large batch size.

As shown in Table 4, a batch size of 512 achieved the best 1-NFE FID of 3.05 and FDD of 164.3. A batch size of 1024, however, yielded the best FDD of 93.4. Overall, a batch size of 1024 performed well across all metrics, so we designate this configuration as MeanFlow-XL/2+. The same setting is applied to fine-tune the MeanFlow-XL/2 model, leading to the MeanFlow-XL/2+ results in Table 1. Our proposed $\alpha$-Flow-XL/2+ model outperforms MeanFlow-XL/2+ in several key metrics: 1-NFE

Table 3: Configurations on ImageNet 256×256. B/2-non-cfg is our ablation and analysis model in the main text.

| Configs | DiT-B/2-non-cfg | DiT-B/2 | DiT-XL/2 | DiT-XL/2+ |
|---|---|---|---|---|
| *Network Architectures* | | | | |
| Params (M) | 131 | 131 | 676 | 676 |
| FLOPs (G) | 23.1 | 23.1 | 119.0 | 119.0 |
| Depth | 12 | 12 | 28 | 28 |
| Hidden dim | 768 | 768 | 1152 | 1152 |
| Heads | 12 | 12 | 16 | 16 |
| Patch size | 2×2 | $2 \times 2$ | $2 \times 2$ | $2 \times 2$ |
| *Training hyperparameters* | | | | |
| Training steps | 400K | 1.2M | 1.2M | 1.2M |
| Batch size for training | 256 | 256 | 256 | 256 |
| Fine-tuning steps | – | – | – | 75K |
| Batch size for fine-tuning | – | – | – | 1024 |
| Dropout | | 0.0 | | |
| Optimizer | | Adam Kingma & Ba (2014) | | |
| lr schedule | | constant | | |
| lr | | 0.0001 | | |
| Adam $(\beta_1, \beta_2)$ | | (0.9, 0.95) | | |
| Weight decay | | 0.0 | | |
| EMA half-life | | 6931 | | |
| Gradient clipping norm | | 16 | | |
| Autoencoder used | | sd-vae-ft-ema | | |
| *$\alpha$-Flow hyperparameters* | | | | |
| Ratio of $r = t$ | Table 2 (b) | 25% | 50% | 50% |
| $(r, t)$ sampler | | logitnorm(–0.4, 1.0) | | |
| $\tilde{v}_{s,t}$ | Table 5 (a) | $v_t$ | $v_t$ | $v_t$ |
| Whether to use EMA for $u_{\theta^-}$ | Table 5 (a) | No | No | No |
| Adaptive weight | Table 5 (b) | $\omega = \alpha / \left( \|\Delta\|_2^2 + c \right)$ | | |
| *Schedule of $\alpha$* | | | | |
| $\gamma$ | 25 | 25 | 25 | |
| $k_s$ | | 0 | 600K | 600K |
| $k_e$ | Table 2 (b) | 1.2M | 1M | 1M |
| $\eta$ | Table 5 (c) | $5 \times 10^{-3}$ | $5 \times 10^{-3}$ | $5 \times 10^{-3}$ |
| *CFG training* | | | | |
| $w$ | – | 1.0 | 0.2 | 0.2 |
| $\kappa$ | – | 1.0 | 0.92 | 0.92 |
| CFG triggered if $t$ is in | – | [0.0, 1.0] | [0.0, 0.75] | [0.0, 0.75] |
| Whether use EMA for CFG | – | No | No | No |
| *2-NFE Sampling* | | | | |
| Method | ODE | ODE | consistency | consistency |
| Intermediate timestep | 0.5 | 0.5 | 0.55 | 0.5 |

FID (2.58 vs. 3.06), 1-NFE FDD (148.4 vs. 165.7) and 2-NFE FID (2.15 vs. 2.16), only worse in 2-NFE FDD (96.8 vs. 93.4). These results demonstrate the overall effectiveness of our $\alpha$-Flow method. Notably, the results in Table 4 are obtained using labels sampled from the ImageNet dataset distribution, whereas the results in Table 1 use randomly generated labels. In general, sampling labels from the ImageNet distribution leads to lower FID scores compared to using random labels.

## H.2 ABLATION STUDY OVER $\alpha$-FLOW DESIGN SPACE

This section contains an ablation study on $\alpha$-Flow, specifically for $\alpha \in (0, 1)$. We use a DiT-B/2-non-cfg model (see Table 3) that is pre-trained on flow matching for 200k iterations and then fine-tuned on $\alpha$-Flow for another 200k iterations. Across all experiments, $\alpha$ remains a constant, and the ratio of $r = t$ is 25 %.

---

**Algorithm 3** $\alpha$-Flow: Sampling

---

```
# 1 = t1 > t2 > ... > tN = 0 :sequence of
    timesteps
z = randn_like(x)
for n in range(N):
    m = n + 1
    if consistency_sampling:
        z = z - tn * fn(z, r=0, t=tn)
        z = z + tm * randn_like(x)
    elif ODE_sampling:
        z = z - (tn - tm) * fn(z, r=tm, t=tn)
```

---

| Batch Size | NFE 1 | | NFE 2 | |
|---|---|---|---|---|
| | FID | FDD | FID | FDD |
| 256 | 3.13 | 167.2 | 2.31 | 97.1 |
| 512 | **3.05** | **164.3** | 2.21 | 95.2 |
| 1024 | 3.06 | 165.7 | 2.16 | **93.4** |
| 2048 | 3.29 | 169.6 | **2.10** | 96.6 |
| 4096 | 3.13 | 168.9 | 2.16 | 95.1 |

Table 4: Ablation study over the fine-tuning batch size using the data distribution over class labels.

| $\tilde{\boldsymbol{v}}_{s,t}$ | $\boldsymbol{u}_{\boldsymbol{\theta}-}$ | FID | FDD |
|---|---|---|---|
| $\boldsymbol{u}_{\boldsymbol{\theta}-}$ | EMA | 188.1 | 1761.6 |
| $\boldsymbol{u}_{\boldsymbol{\theta}-}$ | Non-EMA | 319.0 | 4009.9 |
| $\boldsymbol{v}_t$ | EMA | 202.8 | 1832.3 |
| $\boldsymbol{v}_t$ | Non-EMA | 59.2 | 964.6 |

(a) **Reformulate the training objective.**

| Loss weight | FID | FDD |
|---|---|---|
| $\omega = 1$ | 59.2 | 964.6 |
| $\omega = 1/\left(\|\|\Delta\|\|_2^2 + c\right)^{0.5}$ | 55.0 | 918.5 |
| $\omega = 1/\left(\|\|\Delta\|\|_2^2 + c\right)$ | 52.2 | 883.6 |
| $\omega = \alpha/\left(\|\|\Delta\|\|_2^2 + c\right)$ | **49.7** | **845.2** |

(b) **Adaptive loss.**

| $\alpha$ | FID | FDD |
|---|---|---|
| $10^{-2}$ | 49.7 | 845.2 |
| $5 \times 10^{-3}$ | **46.2** | 860.8 |
| $2 \times 10^{-3}$ | 50.3 | **833.0** |
| $1 \times 10^{-3}$ | 57.2 | 863.7 |

(c) **Consistency step ratio.**

| Method | FID | FDD |
|---|---|---|
| Shortcut Model | 59.8 | 1017.3 |
| $\tilde{\boldsymbol{v}} = \boldsymbol{v}(\boldsymbol{z}_t, t\|\boldsymbol{x})$ | 59.2 | 964.6 |
| + Adaptive loss | 49.7 | 845.2 |
| + $\alpha = 0.005$ | 45.6 | 857.8 |
| MeanFlow | 43.3 | 822.3 |

(d) **Overall ablation study.**

Table 5: Ablation study over $\alpha$-Flow.

**Training objective.** Here, we set $\alpha = 10^{-2}$. Table 5(a) shows that the model only converge when $\tilde{\boldsymbol{v}}_{s,t}$ was set to $\boldsymbol{v}_t$ and without using EMA for $\boldsymbol{u}_{\boldsymbol{\theta}-}$. This is a key difference from Shortcut Models Frans et al. (2025), which set $\tilde{\boldsymbol{v}}_{s,t} = \boldsymbol{u}_{\boldsymbol{\theta}-}$. We suspect their objective only works when $\alpha$ is larger (e.g., 0.5).

**Adaptive loss.** Geng et al. (2025a) uses an adaptive weight: $\omega = 1/\left(\|\|\Delta\|\|_2^2 + c\right) = 1/\left(\mathcal{L}_{\text{MF}} + c\right)$. From Equation (12), we could derive $\lim_{\alpha \to 0} \mathcal{L}_\alpha = \alpha \mathcal{L}_{\text{MF}}$. When $\alpha$ is close 0, we approximate $\mathcal{L}_{\text{MF}}$ as $\mathcal{L}_\alpha/\alpha$. This gives us a new adaptive weight, $\omega = 1/\left(\mathcal{L}_\alpha/\alpha + c\right) \approx \alpha/\left(\mathcal{L}_\alpha + c\right) = \alpha/\left(\|\|\Delta\|\|_2^2 + c\right)$ as both $c$ and $\alpha$ is very small. As shown in Table 5(b), this new weight performs better empirically, especially compared to the original MeanFlow adaptive weight.

**Consistency step ratio.** Ablating the $\alpha$ in Table 5 (c) reveals that $\alpha = 5 \times 10^{-3}$ to be the optimal consistency step ratio. This value was then used as the clamping value for our schedule.

Table 5(d) shows that by combining these improvements, our discrete $\alpha$-Flow approach significantly reduces the performance gap between Shortcut models and the MeanFlow model.

## H.3 OPTIMAL RATIO OF $r = t$ UNDER DIFFERENT NFE

In this subsection, we extend the experiment results in Table 2 (b) to NFE 5. As shown in Table 6, the optimal ratio of $r = t$ required to achieve the best generation quality for NFE 5 is also 75%. We hypothesize this is because Flow Matching inherently possesses a degree of few-step generation capability. Therefore, when mixing the training of standard Flow Matching and $\alpha$-Flow, varying the ratio $r = t$ introduces a trade-off between single-step and few-step generation.

| | Model | NFE 1 | | NFE 2 | | NFE 5 | |
|---|---|---|---|---|---|---|---|
| $\% \, r = t$ | Schedule | FID | FDD | FID | FDD | FID | FDD |
| 0% | $\texttt{Constant}_{0.0}$ | 46.0 | 879.6 | 44.3 | 867.7 | 44.0 | 843.8 |
| | $\texttt{Sigmoid}_{0K \to 400K}$ | 40.4 | 822.5 | 38.9 | 811.8 | 38.2 | 779.0 |
| 25% | $\texttt{Constant}_{0.0}$ | 44.4 | 844.1 | 42.1 | 836.3 | 41.3 | 817.8 |
| | $\texttt{Sigmoid}_{0K \to 400K}$ | **40.0** | 785.4 | 37.1 | 782.9 | 36.9 | 770.0 |
| 50% | $\texttt{Constant}_{0.0}$ | 43.9 | 844.1 | 42.1 | 836.3 | 38.4 | 783.3 |
| | $\texttt{Sigmoid}_{0K \to 400K}$ | 40.2 | **781.0** | 37.1 | 775.0 | 35.5 | 743.5 |
| 75% | $\texttt{Constant}_{0.0}$ | 43.1 | 819.2 | 38.5 | 787.6 | 36.0 | 752.1 |
| | $\texttt{Sigmoid}_{0K \to 400K}$ | 42.2 | 810.5 | **36.2** | **754.7** | **33.7** | **716.1** |

Table 6: Optimal $\%t = r$ under different NFE.

| Model | NFE 1 | | NFE 2 | |
|---|---|---|---|---|
| | FID | FDD | FID | FDD |
| $\texttt{FM}_{0K \to 200K} + \texttt{MeanFlow}_{200K \to 400K}$ | 42.3 | 798.1 | 37.6 | **772.4** |
| $\texttt{TFM}_{0K \to 200K} + \texttt{MeanFlow}_{200K \to 400K}$ | 42.3 | 821.8 | 37.2 | 780.2 |
| $\texttt{Sigmoid}_{0K \to 400K}$ | **40.2** | **781.0** | **37.1** | 775.0 |
| $\texttt{REPA}_{0K \to 400K} + \texttt{MeanFlow}_{400K \to 600K}$ | 32.0 | 667.4 | 25.5 | **594.3** |
| $\texttt{REPA}_{0K \to 400K} + \texttt{Sigmoid}_{400K \to 600K}$ | **30.7** | **657.1** | **25.3** | 601.5 |

Table 7: Finetuning from pretrained flow models.

## H.4 FINETUNING FROM PRETRAINED FLOW MODELS

In this subsection, we study the performance of $\alpha$-Flow by finetuning from pretrained flow models. Specifically, We have conducted experiments comparing $\alpha$-Flow with standard MeanFlow finetuning when starting from pre-trained Flow Matching (FM), Trajectory Flow Matching (TFM), and FM + Representation Alignment Yu et al. (2025) (REPA) models. We use DiT-B/2-non-cfg configs in Table 3 for backbone models. Both FM, TFM and FM + REPA are pretrained by ourselves. Experiment results are shown in Table 7. A specific configuration for $\alpha$-Flow deserves mention: for the experiment labeled $\texttt{REPA}_{0K \to 400K} + \texttt{Sigmoid}_{400K \to 600K}$, we adjusted the initial $\alpha$ parameter. Since the pre-trained model was provided at the 400K iterations, we set the initial value of $\alpha$ to 0.5 instead of 1.0 at the beginning of the $\alpha$-Flow fine-tuning phase.

Under a direct epoch-to-epoch comparison, starting from the same pre-trained model, $\alpha$-Flow consistently outperforms MeanFlow for one-step generation (NFE 1) and achieves comparable performance for two-step generation. Specifically, under NFE 1, $\alpha$-Flow (FID 40.2, FDD 781.0) demonstrates significant gains: it surpasses the FM + MeanFlow baseline by 2.1 FID and 17.1 FDD, and it surpasses the TFM + MeanFlow baseline by 2.1 FID and 40.8 FDD. Furthermore, when initialized with the same FM+ REPA model, $\alpha$-Flow (FID 30.7, FDD 657.1) still outperforms MeanFlow by 1.3 FID and 10.3 FDD in one-step generation. Since the $\alpha$-Flow method inherently consists of three sequential stages: TFM pre-training, $\alpha$-Flow annealing, and MeanFlow fine-tuning, these results directly demonstrate the effectiveness of the $\alpha$-Flow annealing stage. This stage is particularly beneficial for improving the quality of one-step sampling.

## H.5 DISTILLATION

In this subsection, we conduct the distillation experiment. Specifically, we distilled the $\alpha$-Flow model (using the DiT-B/2 architecture) with a DiT-B/2-REPA teacher model, which we trained for 400K iterations.

The experimental results, summarized in Table 8, clearly demonstrate that distillation improves the performance of $\alpha$-Flow for both NFE 1 and NFE 2 settings. The distilled $\alpha$-Flow model, using the $\texttt{Sigmoid}_{0K \to 100K}$ schedule (the first line in Table above), outperformed the $\alpha$-Flow trained from scratch with the $\texttt{Sigmoid}_{0K \to 400K}$ schedule (the last line of the Table 2 (a)). For NFE 1, the

| Schedule | NFE 1 | | NFE 2 | |
|---|---|---|---|---|
| | FID | FDD | FID | FDD |
| $\text{Sigmoid}_{0K \to 100K}$ | **36.6** | 756.9 | **33.2** | **701.3** |
| $\text{Sigmoid}_{50K \to 150K}$ | 37.1 | 749.2 | 33.4 | 702.9 |
| $\text{Sigmoid}_{100K \to 200K}$ | 37.8 | 757.5 | 33.5 | 704.7 |
| $\text{Sigmoid}_{150K \to 250K}$ | 36.7 | 743.2 | 34.1 | 706.9 |
| $\text{Sigmoid}_{0K \to 400K}$ | 36.9 | **738.9** | 33.6 | 704.3 |

Table 8: Distillation of $\alpha$-Flow-B/2.

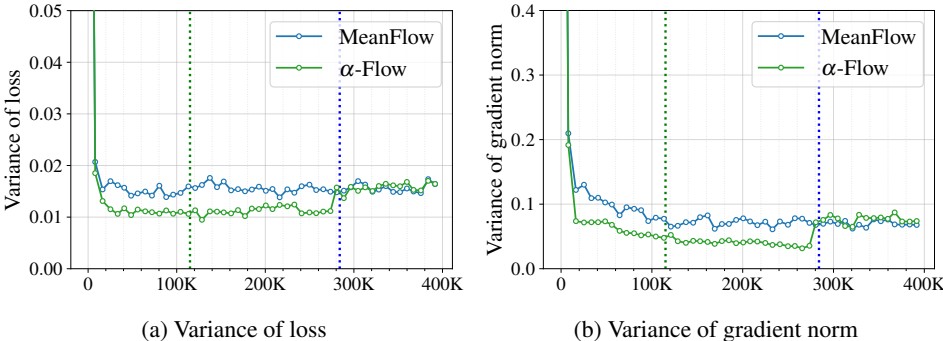

(a) Variance of loss

(b) Variance of gradient norm

Figure 9: Stability comparison of $\alpha$-Flow and MeanFlow. The green dotted line indicates the transition from TFM pretraining to $\alpha$-Flow annealing, while the blue dotted line represents the subsequent transition from $\alpha$-Flow annealing to MeanFlow training.

distilled model achieved improvements of 3.4 (FID) and 28.5 (FDD). For NFE 2, the distilled model showed improvements of 3.9 (FID) and 81.6 (FDD).

It is important to note that the optimal distillation schedule shifted to $\text{Sigmoid}_{0K \to 100K}$. This change is expected, as the teacher model (DiT-B/2-REPA) was already pretrained. This pretrained knowledge allows the distillation process to reducing the initial $\alpha = 1$ training stage.

### H.6 STABILITY ANALYSIS BETWEEN MEANFLOW AND $\alpha$-FLOW

In this subsection, we analyze the training dynamics of MeanFlow and $\alpha$-Flow by evaluating the variance of the MeanFlow loss and the gradient norms throughout the training process. Experiment results are shown in Figure 9, where the variance for each loss and gradient norm is estimated over 500 iterations.

As shown in Figure 9, the variance of the loss and the gradient norm for $\alpha$-Flow are lower than those for MeanFlow during the TFM pretraining and $\alpha$-Flow annealing stages (to the left of the blue dot line). This indicates a more stable training process for $\alpha$-Flow in these early stages. This stability aligns with the statement that $\alpha$-Flow annealing effectively transitions the training objective from a "high-bias, low-variance" state to the necessary "low-bias, high-variance" state.

Despite these early stability advantages, the $\alpha$-Flow formulation ultimately trains the full MeanFlow loss in its final stage (as $\alpha \to 1$). Consequently, $\alpha$-Flow cannot fully circumvent the inherent instability introduced by the MeanFlow objective.

### H.7 MORE GUIDANCE METHOD

Beyond standard Classifier-Free Guidance (CFG) training, we explored an alternative guidance method called $u$-target guidance. This method is based on optimizing the following loss function:

$$\mathcal{L}_{u_t gt}(\boldsymbol{\theta}) = \underset{t, r, \boldsymbol{z}_t, \boldsymbol{c}}{\mathbb{E}} \left\| \boldsymbol{u}_{\boldsymbol{\theta}}(\boldsymbol{z}_t, r, t | \boldsymbol{c}) - (1 + \omega_{u_t gt}) \cdot \boldsymbol{u}_{\text{taget}}(\boldsymbol{z}_t, r, t) - \omega_{u_t gt} \cdot \boldsymbol{u}_{\boldsymbol{\theta}}(\boldsymbol{z}_t, r, t | \varnothing) \right\|_2^2, \quad (50)$$

where $\omega_{u_t gt}$ is the guidance strength. The specific definition of the target vector, $\boldsymbol{u}_{\text{taget}}(\boldsymbol{z}_t, r, t)$, depends on the underlying loss formulation being used:

| Guidance method | NFE 1 | | NFE 2 | |
|---|---|---|---|---|
| | FID | FDD | FID | FDD |
| $u$-target guidance | 25.1 | 645.1 | 9.5 | 392.7 |
| CFG guidance | **10.0** | **445.8** | **7.7** | **365.2** |

Table 9: Ablation of guidance methods.

- For MeanFlow loss, $\boldsymbol{u}_{\texttt{taget}}(\boldsymbol{z}_t, r, t) := \boldsymbol{v}_t - (t-r)\frac{\mathrm{d}\boldsymbol{u}_{\boldsymbol{\theta}^-}(\boldsymbol{z}_t, r, t)}{\mathrm{d}t}$

- For $\alpha$-Flow loss, $\boldsymbol{u}_{\texttt{taget}}(\boldsymbol{z}_t, r, t) := (\alpha \cdot \tilde{\boldsymbol{v}}_{s,t} + (1-\alpha) \cdot \boldsymbol{u}_{\boldsymbol{\theta}^-}(\boldsymbol{z}_s, r, s))$.

The fundamental difference between $u$-target guidance and CFG training lies in the unconditional prediction component. The $u$-target guidance uses $\mathbf{u}_{\boldsymbol{\theta}}(\mathbf{z}_t, r, t|\varnothing)$ for the unconditional prediction conditioned, instead of unconditional prediction $\mathbf{u}_{\boldsymbol{\theta}}(\mathbf{z}_t, t, t|\varnothing)$ employed in CFG training.

We compared $u$-target guidance with CFG guidance using MeanFlow-B/2 architecture, training on ImageNet-1K $256^2$ for 400K iterations. We set $\omega_{\mathbf{u}_{\texttt{rgt}}} = 2.5$. Experiment results are shown in Table 9. The $u$-target guidance performs significantly worse than CFG guidance when using NFE 1 and is still slightly worse even at NFE 2. Thus in the main paper we use the CFG guidance.

### H.8 DISCUSSION OF THE GRADIENT CONFLICT

In this subsection, we discuss the gradient conflict between $\nabla_{\boldsymbol{\theta}}\mathcal{L}_{\text{TFM}}$ and $\nabla_{\boldsymbol{\theta}}\mathcal{L}_{\text{TC}}$ in Figure 3a. In general, for the minimizer $x^*$ of some arbitrary, regular function $f(x) = g(x) + h(x)$, one can trivially show the relation $\nabla_x g(x^*) = -\nabla_x h(x^*)$ from the necessary condition $\nabla_x f(x^*) = 0$.

However, our empirical analysis in Figure 3a shows this gradient relationship from the early start of the optimization process, where the gradients of $\mathcal{L}_{\text{TFM}}$ and $\mathcal{L}_{\text{TC}}$ are already strongly negatively correlated (specifically, $< -0.9$). Crucially, this correlation becomes weaker (around $-0.7$) as training progresses.

This behavior is opposite to what would be expected if the negative correlation simply resulted from convergence to a stationary point. Instead, the decreasing correlation indicates that the two loss components exert conflicting optimization signals early in training, and gradually become more aligned as the model parameters adapt.

This suggests that the observed gradient dissimilarity reflects a different underlying mechanism than the trivial relation implied by $\nabla_x f(x^*) = 0$. This mechanism could be better explained by how far apart the corresponding minimizers $\arg\min g(x)$ and $\arg\min h(x)$ are. This evolution likely reflects the interaction between two loss components whose minima are initially far apart: early in training, their gradients point toward distinct optima, leading to strong opposition, whereas as the model parameters evolve and the shared representation (and with the optimizer's preconditioning reweighting the directions) begins to satisfy both objectives, the corresponding descent directions become partially aligned.

## I LLM USAGE

As requested by the ICLR 2026 policy[3], we disclose the usage of Large Language Models in this section. LLMs were primarily used in two capacities:

- Coding assistance for experiments. LLMs provided code auto-completion functionality to ease the process of implementing and analyzing the experiments.

- Writing assistance for paper writing. We used LLMs to assist with grammar and phrasing validation while working on the submission.

---

[3]https://iclr.cc/Conferences/2026/AuthorGuide

## J   Random vs balanced classes for FID computation

We treat EDM series (Karras et al., 2022; 2024) as the standard in FID (Heusel et al., 2017) evaluations, which use a randomly sampled class label (from 0 to 999) for each sample in constructing 50,000 synthetic examples with the model. We found a curious way to decrease the FID values by up to 10% by using "balanced" class sampling: instead of using 50,000 independently sampled random classes, one can generate 50 samples for each of 1000 classes. This greatly improves FID results, but not FDD (*i.e.*, Fréchet Distance in the DINOv2 (Oquab et al., 2023) feature space) or FCD (Kynkäänniemi et al., 2022) (*i.e.*, Fréchet Distance in the CLIP-L-based (Radford et al., 2021) feature space).

Since it is not a standard practice in the community, we only report it separately from the random class sampling results and with the appropriate notice. But we emphasize that it might be a more reasonable way to evaluate FID since it reduces the variance (we are less likely to sample an unlucky set of classes). We provide the results for it in Table 10.

| Method | Class sampling | Params | Epochs | NFE 1 | | | NFE 2 | | |
|---|---|---|---|---|---|---|---|---|---|
| | | | | FID | FDD | FCD | FID | FDD | FCD |
| MeanFlow-XL/2* | Random $U[1..1000]$ | 676M | 240 | 3.47 | 185.8 | 3.39 | 2.46 | 108.7 | 2.40 |
| $\alpha$-Flow-XL/2 (ours) | Random $U[1..1000]$ | 676M | 240 | 2.95 | 164.6 | 3.14 | 2.32 | 105.7 | 2.42 |
| $\alpha$-Flow-XL/2+ (ours) | Random $U[1..1000]$ | 676M | 240+60 | **2.58** | **148.4** | **3.07** | **2.15** | **96.8** | **2.31** |
| MeanFlow-XL/2* | Balanced | 676M | 240 | 3.33 | 182.8 | 3.34 | 2.26 | 106.1 | 2.36 |
| $\alpha$-Flow-XL/2 (ours) | Balanced | 676M | 240 | 2.81 | 162.4 | 3.10 | 2.16 | 103.2 | 2.37 |
| $\alpha$-Flow-XL/2+ (ours) | Balanced | 676M | 240+60 | **2.44** | **147.2** | **3.04** | **1.95** | **94.6** | **2.30** |

Table 10: Balanced vs random class sampling for FID, FDD and FCD.

It is curious to observe that while it greatly improves FID results, FDD and FCD are barely affected. We believe that this constitutes one more reason for the community to switch from FID to more robust metrics which correlate better with human perception, like FDD and FCD.

# K    ADDITIONAL EXPLORATION OF THE MEANFLOW LOSS

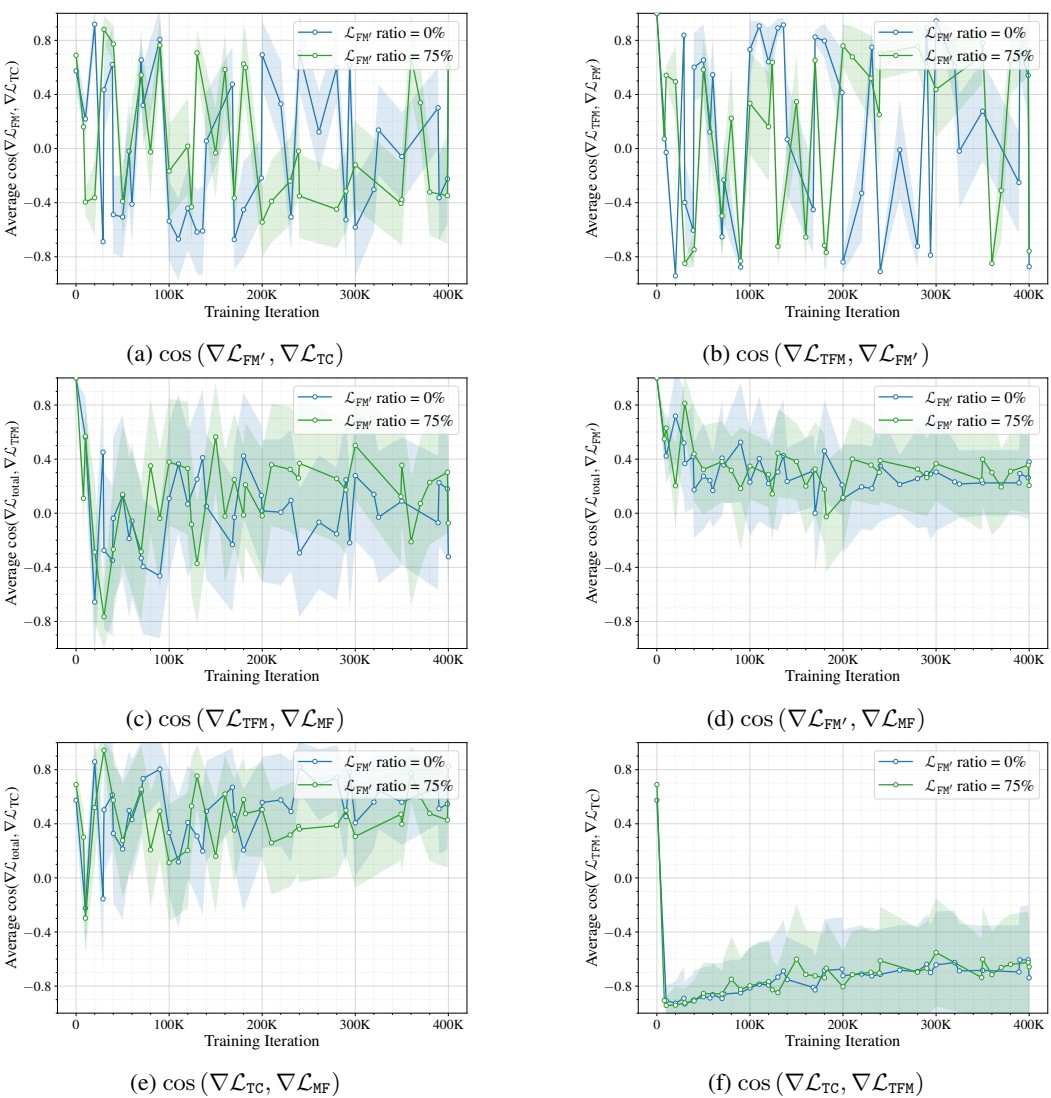

Figure 10:    Average cosine similarities between the gradients of different losses $(\mathcal{L}_{\text{TFM}}, \mathcal{L}_{\text{FM}'}, \mathcal{L}_{\text{CT}_c}, \mathcal{L}_{\text{MF}})$ for DiT-B/2 MeanFlow model trained with 0% and 75% of flow matching.

## L    ADDITIONAL VISUALIZATIONS

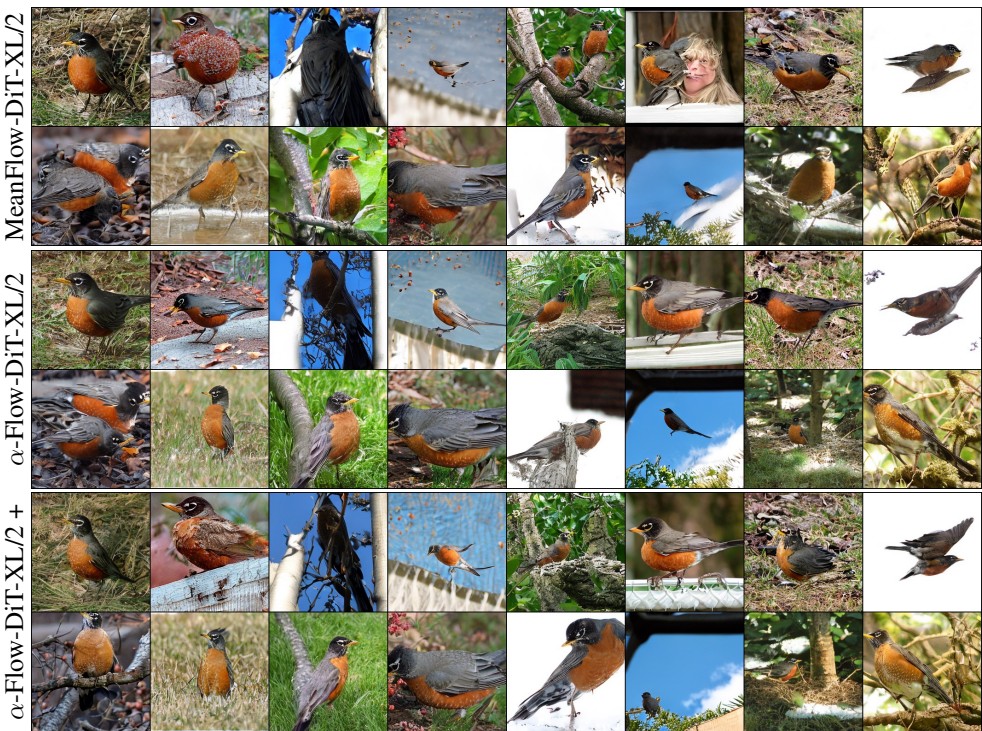

Figure 11: Uncurated samples (seeds 1-16) for Class 15 (robin) for NFE=1.

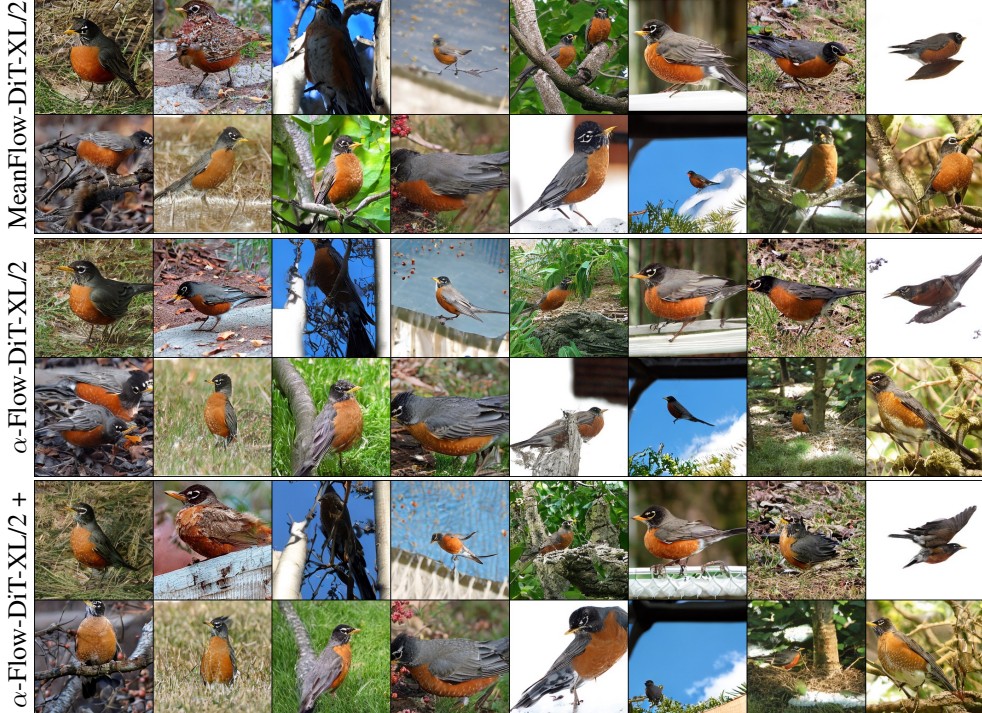

Figure 12: Uncurated samples (seeds 1-16) for Class 15 (robin) for NFE=2.

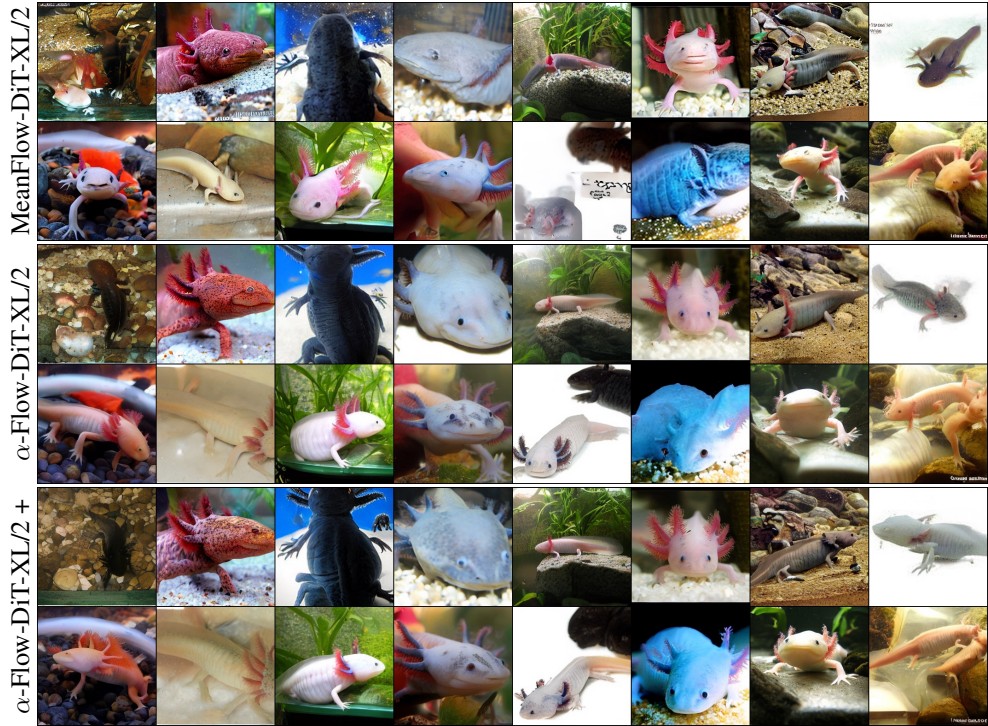

Figure 13: Uncurated samples (seeds 1-16) for Class 29 (axolotl) for NFE=1.

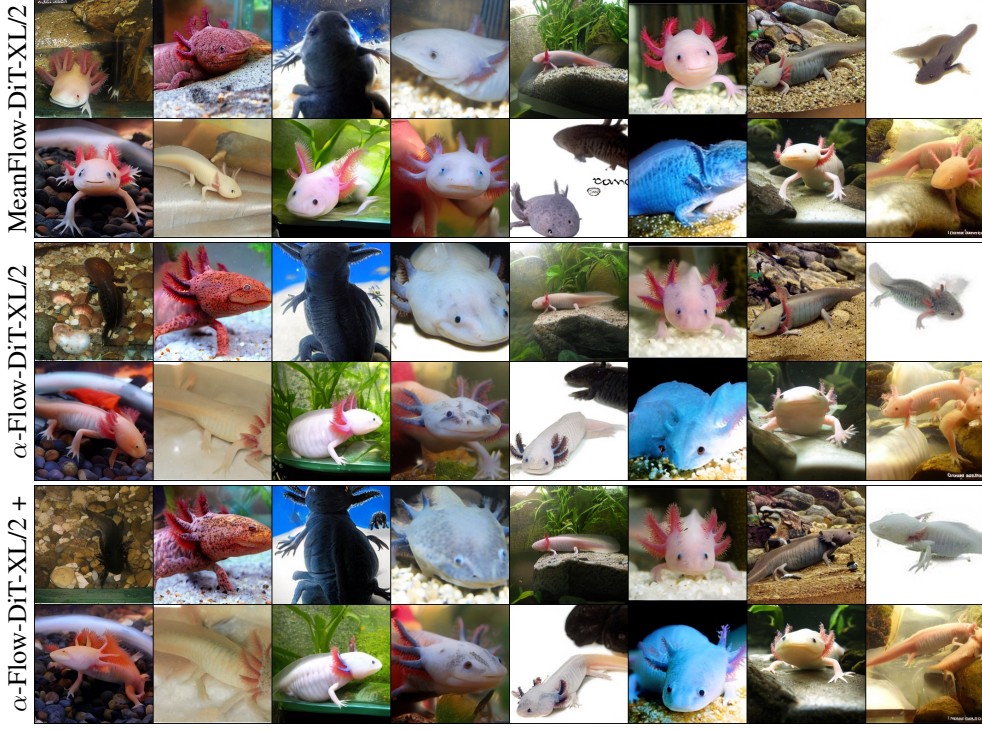

Figure 14: Uncurated samples (seeds 1-16) for Class 29 (axolotl) for NFE=2.

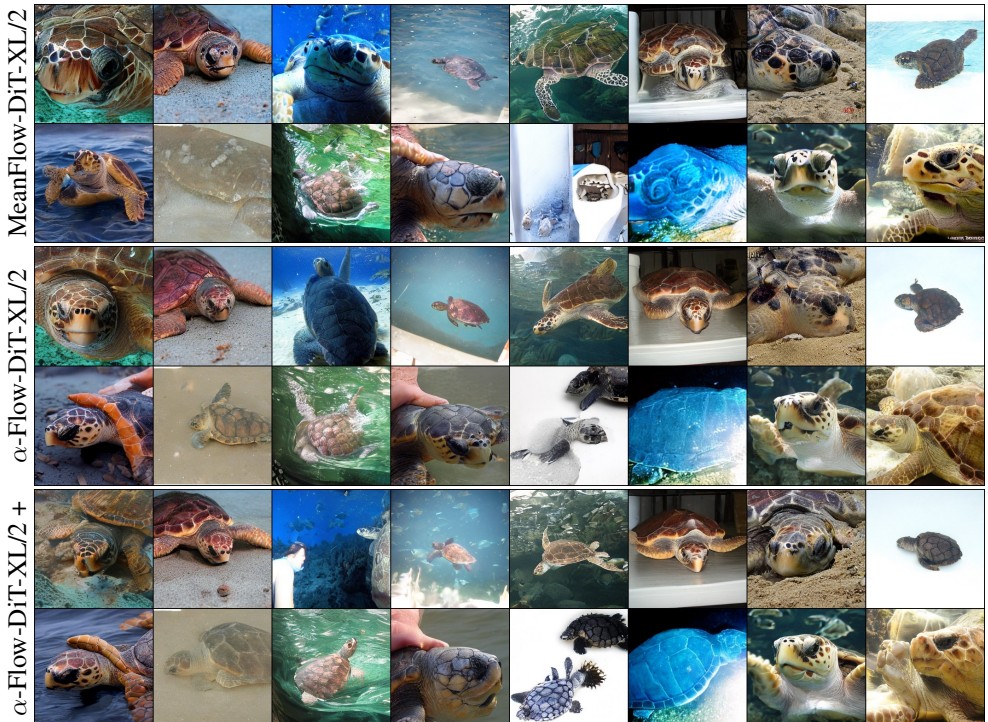

Figure 15: Uncurated samples (seeds 1-16) for Class 33 (loggerhead) for NFE=1.

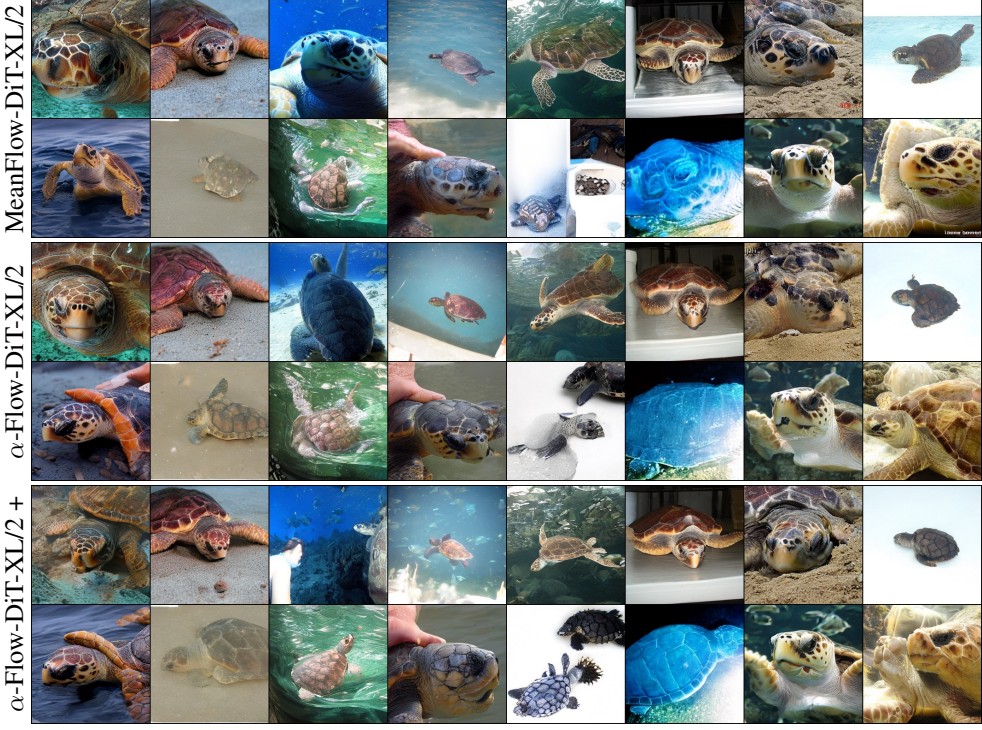

Figure 16: Uncurated samples (seeds 1-16) for Class 33 (loggerhead) for NFE=2.

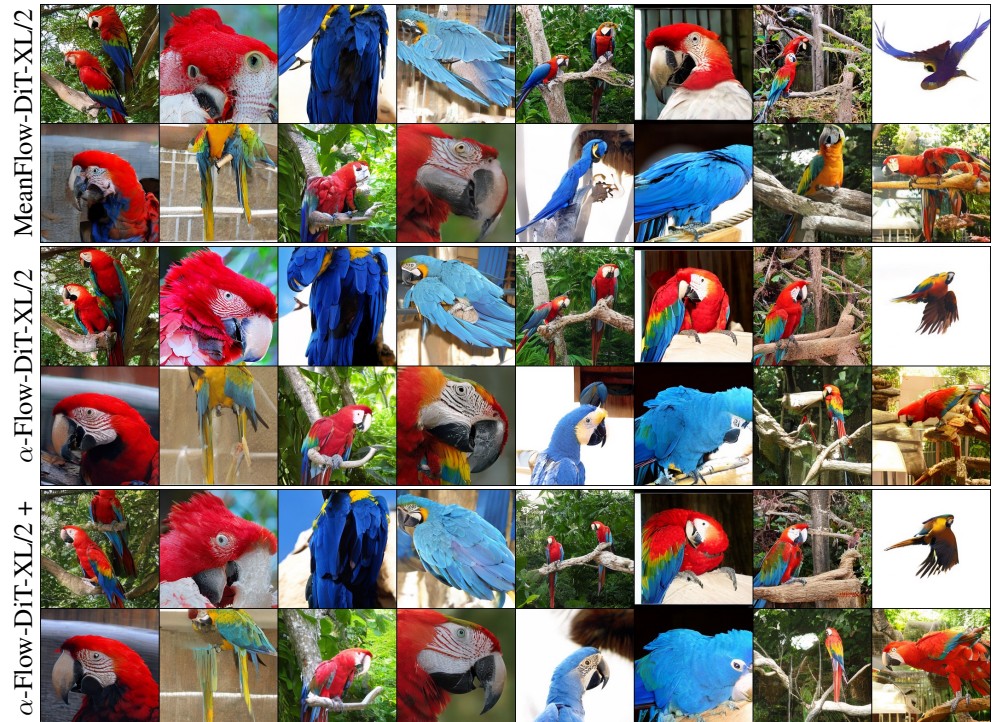

Figure 17: Uncurated samples (seeds 1-16) for Class 88 (macaw) for NFE=1.

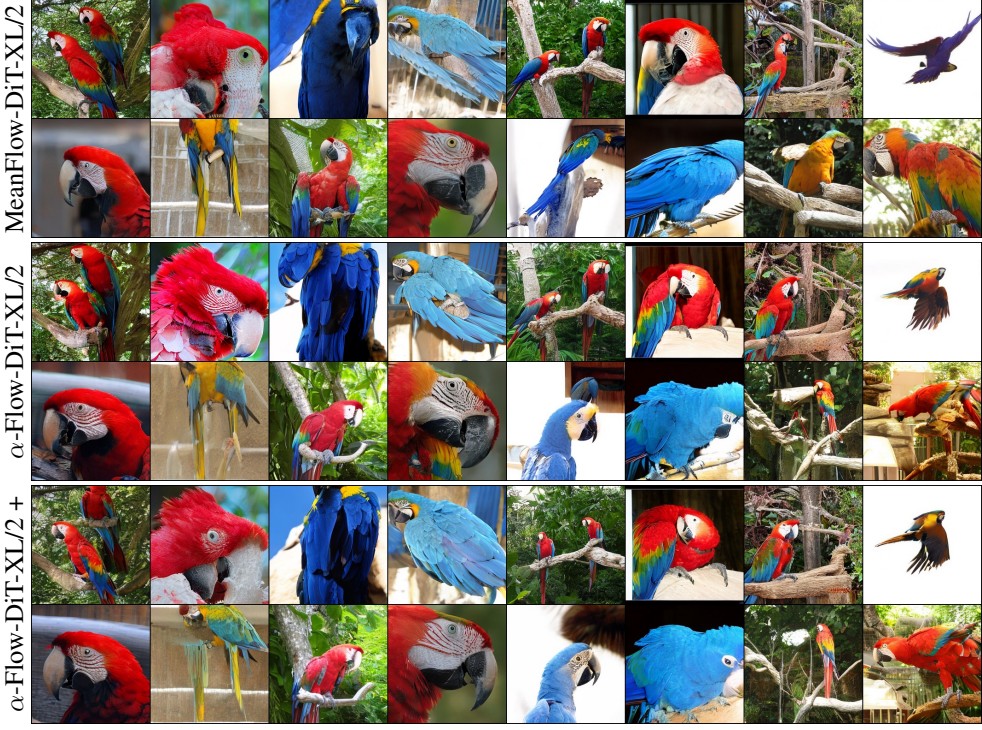

Figure 18: Uncurated samples (seeds 1-16) for Class 88 (macaw) for NFE=2.

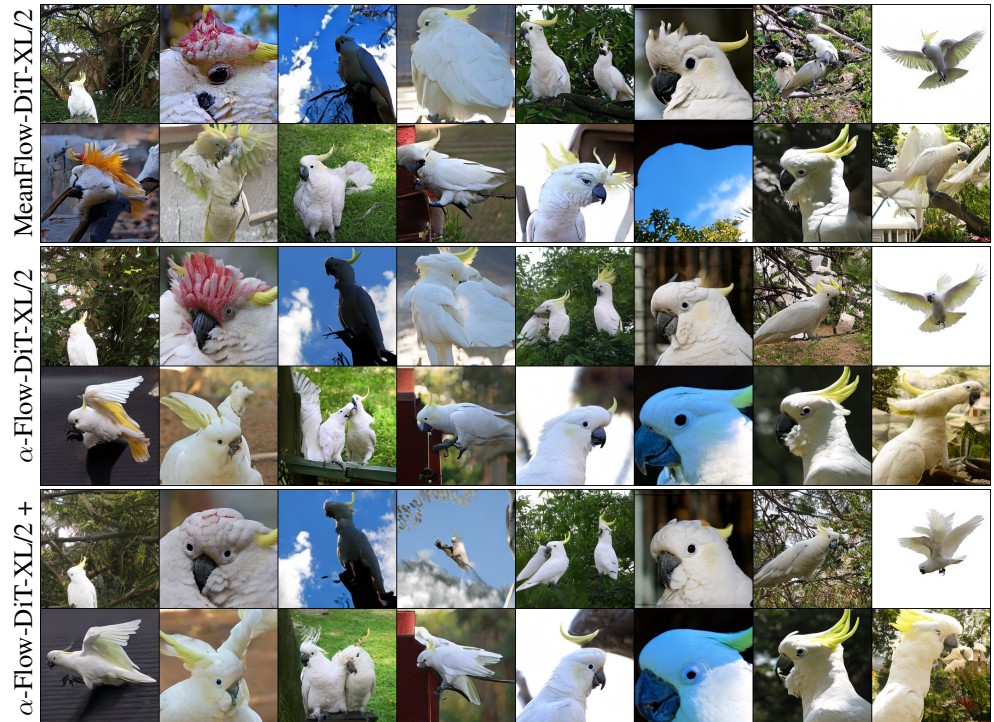

Figure 19: Uncurated samples (seeds 1-16) for Class 89 (cockatoo) for NFE=1.

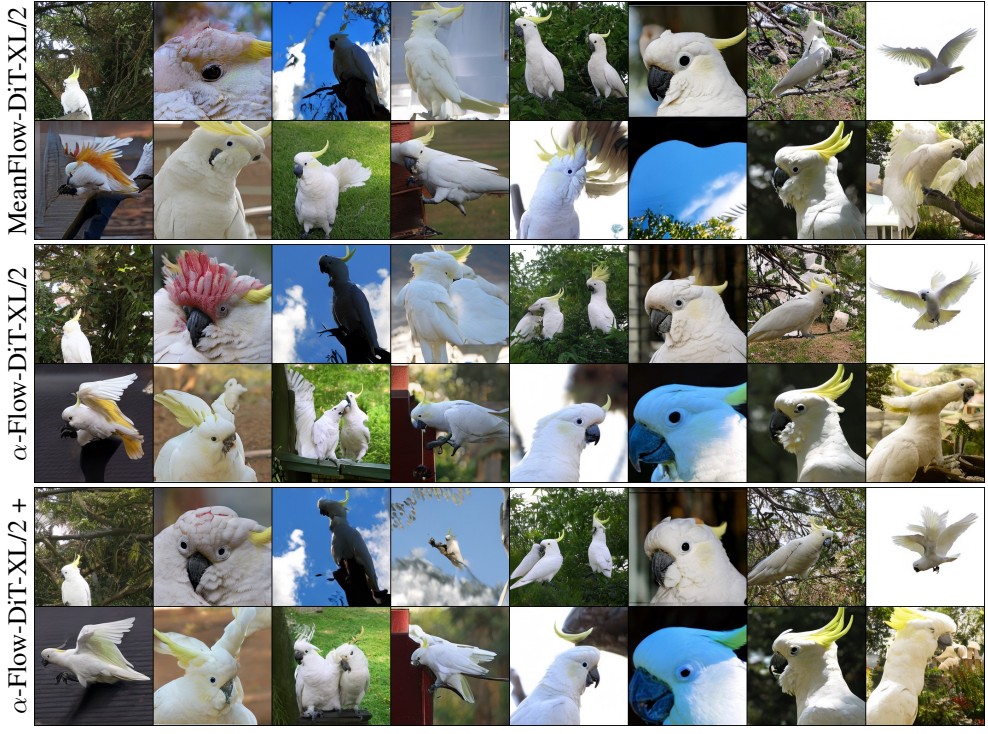

Figure 20: Uncurated samples (seeds 1-16) for Class 89 (cockatoo) for NFE=2.

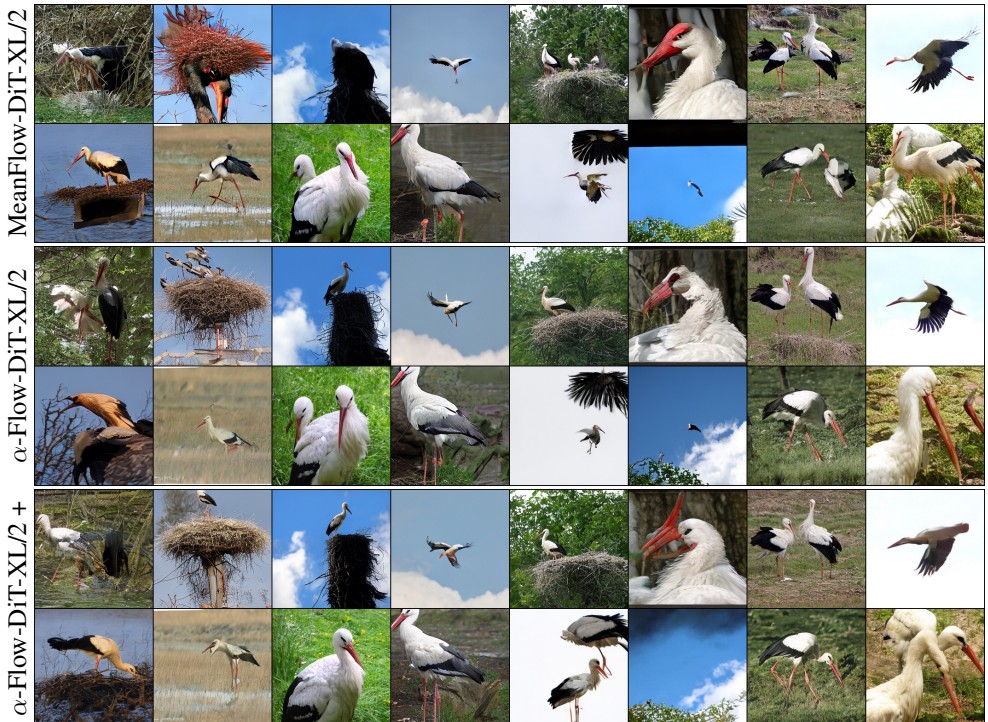

Figure 21: Uncurated samples (seeds 1-16) for Class 127 (white stork) for NFE=1.

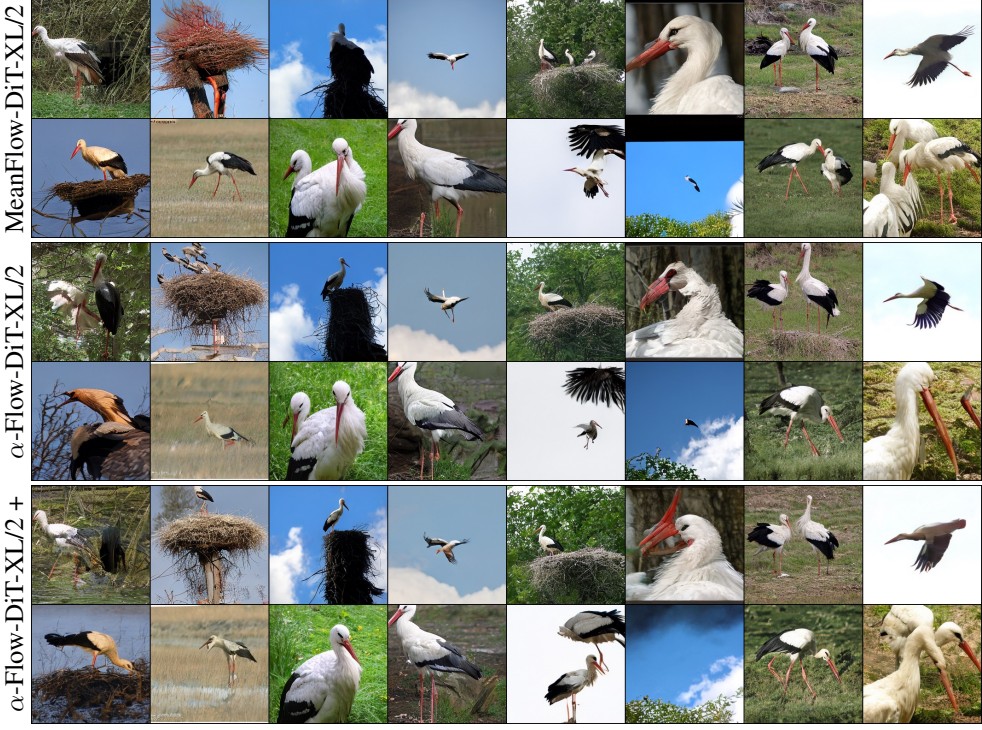

Figure 22: Uncurated samples (seeds 1-16) for Class 127 (white stork) for NFE=2.

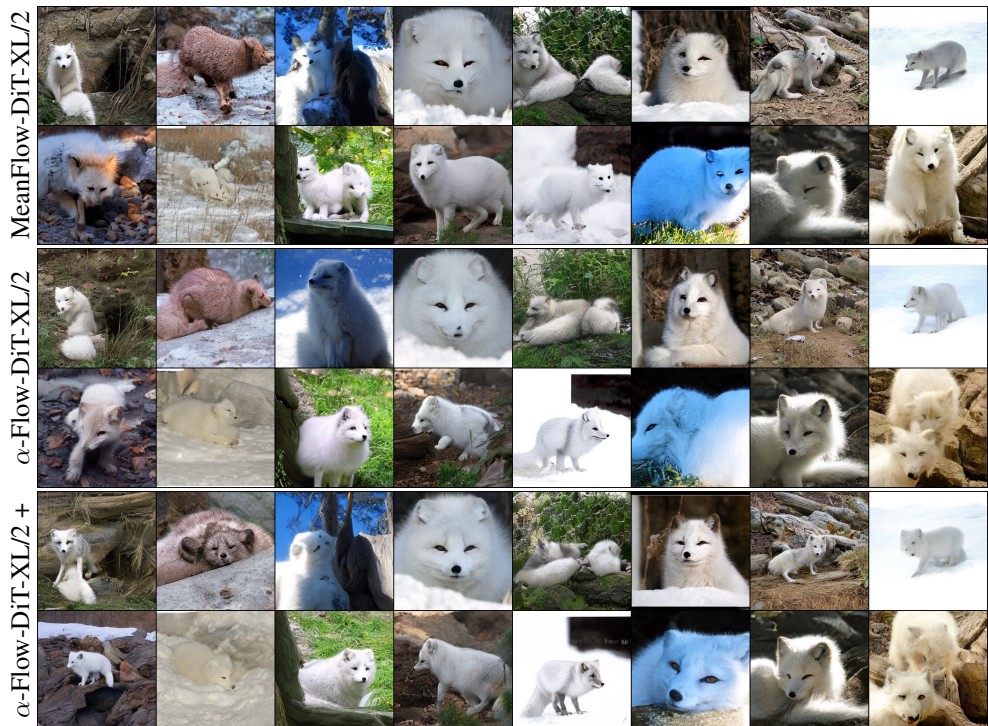

Figure 23: Uncurated samples (seeds 1-16) for Class 279 (arctic fox) for NFE=1.

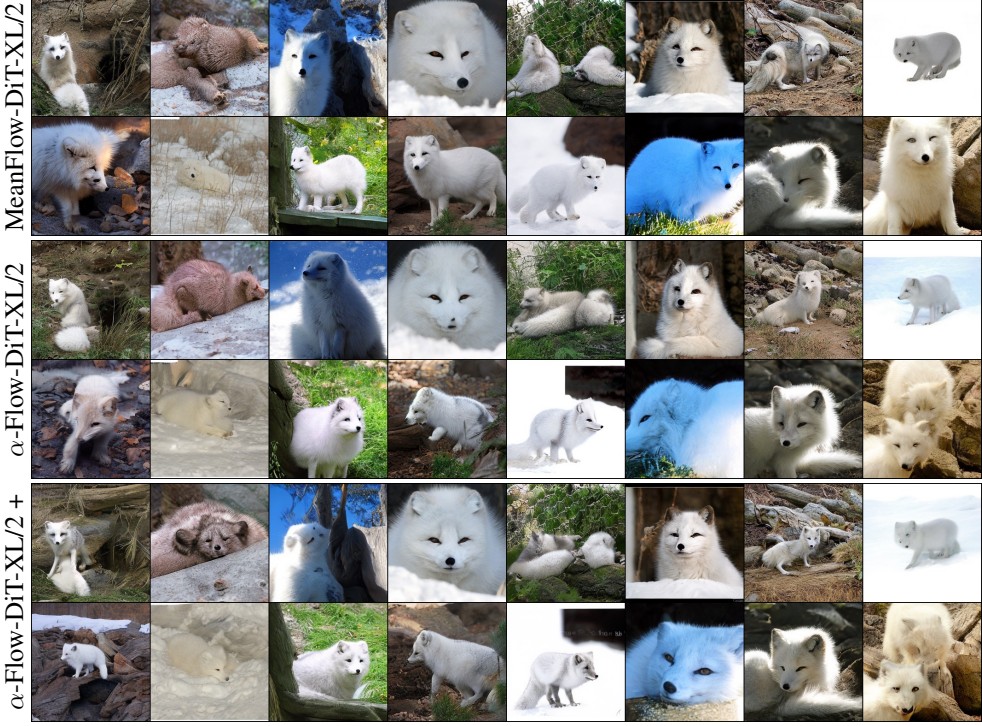

Figure 24: Uncurated samples (seeds 1-16) for Class 279 (arctic fox) for NFE=2.

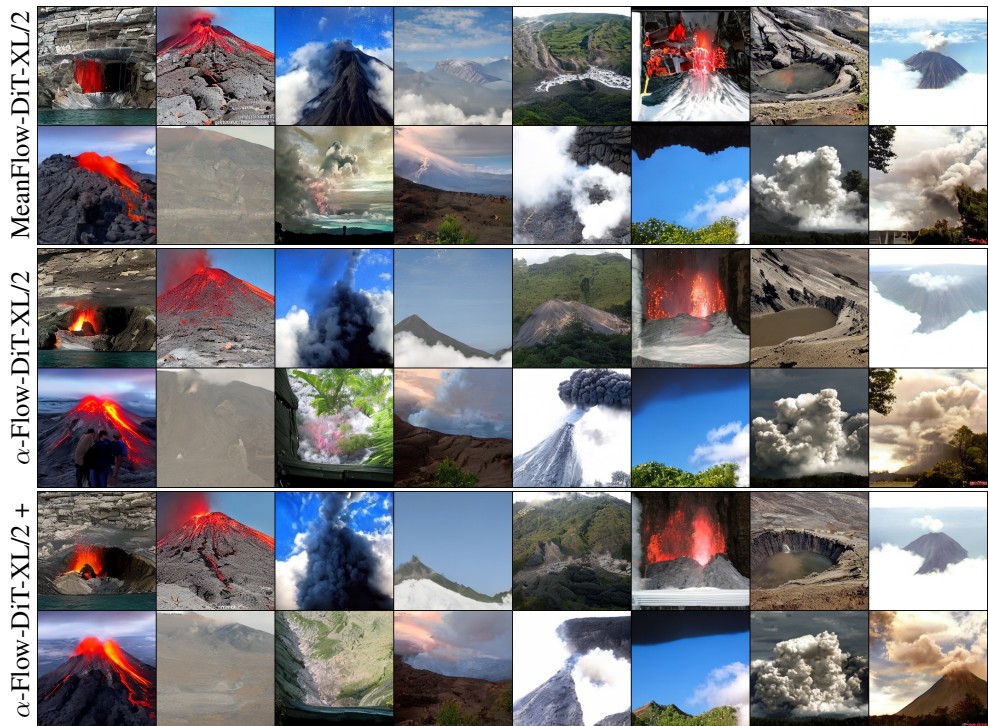

Figure 25: Uncurated samples (seeds 1-16) for Class 980 (volcano) for NFE=1.

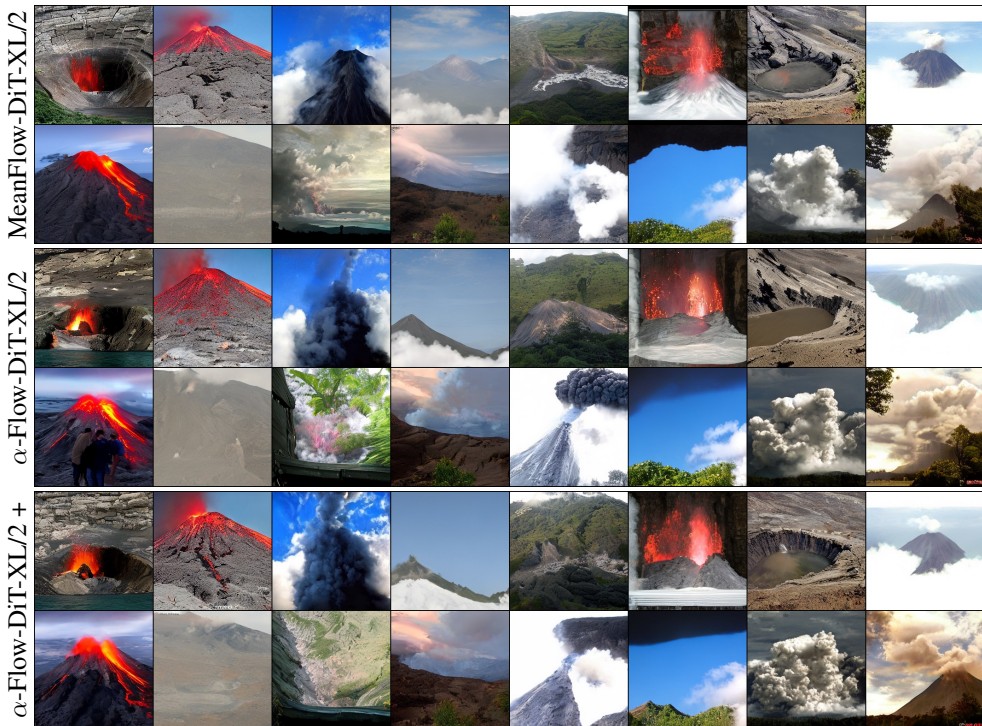

Figure 26: Uncurated samples (seeds 1-16) for Class 980 (volcano) for NFE=2.

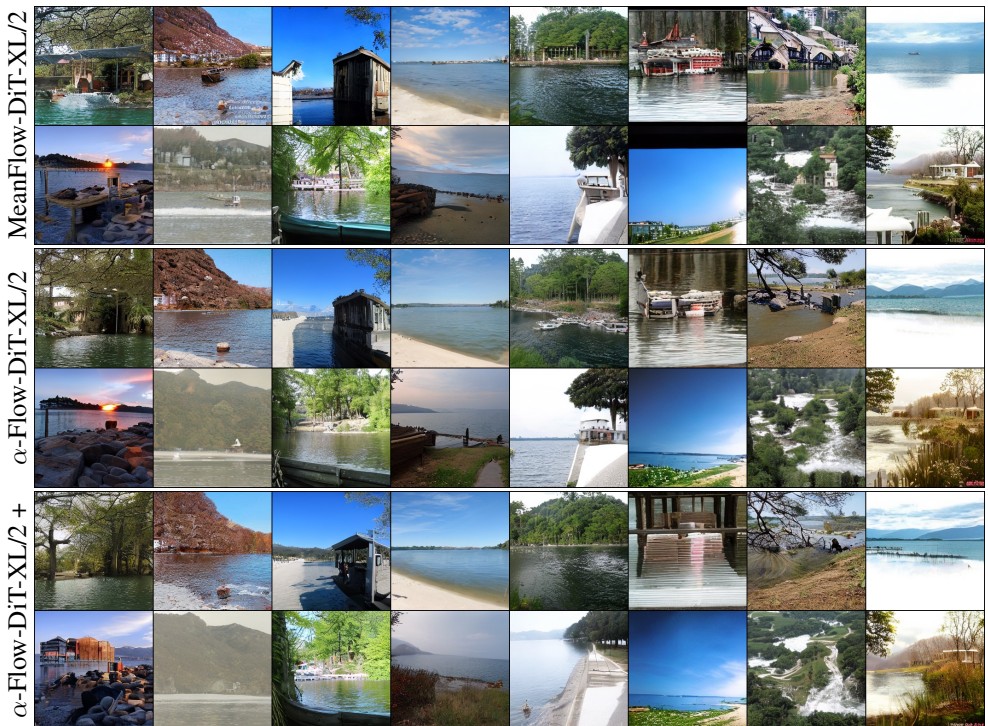

Figure 27: Uncurated samples (seeds 1-16) for Class 975 (lakeside) for NFE=1.

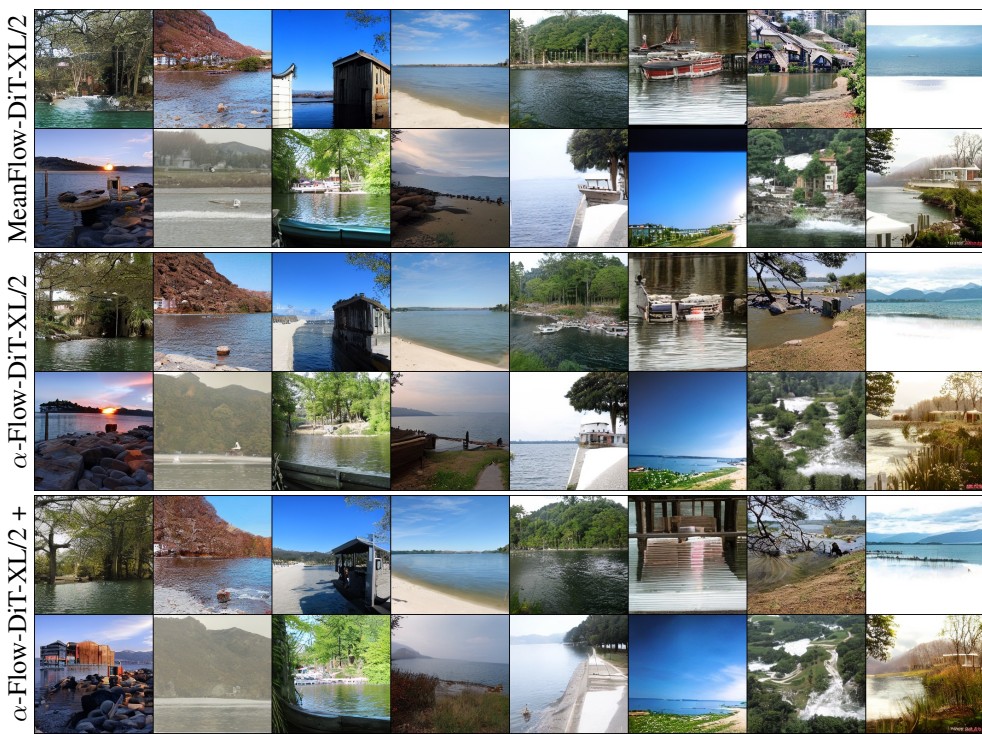

Figure 28: Uncurated samples (seeds 1-16) for Class 975 (lakeside) for NFE=2.

