# OpenReview forum: "AlphaFlow: Understanding and Improving MeanFlow Models"
_ICLR.cc/2026/Conference — ICLR 2026 Poster_

### Official Review · Reviewer_9SDg · 2025-10-26

**Soundness:** 2
**Presentation:** 3
**Contribution:** 2
**Rating:** 4
**Confidence:** 3

**Summary:**

This paper provides a theoretical and empirical analysis of MeanFlow, a recently proposed
framework for few-step generative modeling. The authors decompose the MeanFlow objective
into trajectory flow matching and trajectory consistency components,
revealing that their gradients are strongly negatively correlated during training, causing
optimization conflict. Motivated by this insight, they propose α-Flow, a unified family of
objectives.The paper is well-motivated and clearly written.

**Strengths:**

1. **Clear problem motivation and analysis**: The paper provides valuable insights into *why*
   MeanFlow works by decomposing its objective into interpretable components. The gradient
   analysis in Figure 2 effectively demonstrates the negative correlation between L_TFM and
   L_TCc gradients.


2. **Well-motivated curriculum learning strategy**: The three-phase training approach (trajectory
   flow matching pretraining → α-Flow transition → MeanFlow fine-tuning) is well-justified by
   the observed gradient conflicts and is intuitive.


3. **Thorough ablation studies**: Table 2 and Table 5 provide detailed ablations on curriculum
   schedule, flow matching ratio, training objectives, adaptive loss weights, consistency step
   ratio, and more. This demonstrates systematic investigation.

**Weaknesses:**

**α-Flow is an incremental generalization**
- The core idea is straightforward: interpolate between L_TFM (α=1) and MeanFlow (α→0) via
  curriculum learning.
- While Theorem 1 unifies existing methods, this is largely a mathematical observation rather
  than a conceptual innovation.
- The curriculum learning strategy is intuitive but not fundamentally novel—using easier
  objectives first is a well-established practice.

**Gradient analysis is empirical, not theoretical**
- Figure 2 shows empirical gradient correlations but provides no theoretical explanation for
  *why* the gradients conflict.
- The paper hypothesizes: "L_TCc has a very large solution manifold compared to L_TFM's
  narrow manifold" (Section 3.2) but provides no proof or rigorous analysis.
- No formal characterization of when/why conflicting gradients occur in optimization of
  multiple objectives.

**Curriculum learning for generative models is not new**
- Curriculum strategies in diffusion/flow models have been explored:
  - Progressive distillation (Salimans & Ho, 2022) uses curriculum-like approaches
  - EDM (Karras et al., 2022) employs careful scheduling
  - Consistency model training already uses staged objectives
- The novelty of applying curriculum to MeanFlow is limited.

**Main contribution is improved MeanFlow, not a fundamental advance**
- α-Flow is essentially MeanFlow with curriculum learning + reduced flow matching ratio


**Theorem 1 proof is incomplete**
- The proof in Appendix D.2 shows α-Flow reduces to existing objectives at specific α values
  but doesn't establish theoretical properties (convergence, optimality, bias-variance).
- For MeanFlow: proof shows ∇_θ L_MF(θ) = ∇_θ L_α→0(θ) as α→0, but this is just asymptotic
  equivalence, not equivalence at finite α.
- Missing: formal analysis of how well finite-α approximates α=0.

**Manifold hypothesis (Section 3.2) is unsupported**
- Hypothesis: "L_TCc has a large solution manifold; L_TFM has a narrow manifold."
- No measurement of manifold size, rank, or structure.

**Notation could be clearer**
- L_TFM vs. L_FM′: distinction subtle and easy to confuse
- L_TCc notation for consistency loss is non-standard (why "Cc"?)
- Multiple uses of subscript/superscript make equations hard to parse


**Missing experimental details**
- How is "gradient similarity" computed exactly? (Appendix E brief on this)
- How many iterations used for averaging in Figure 2?

**Questions:**

1. **Why does curriculum learning specifically help?** Can you show that gradient conflict
   decreases during α-annealing? Is it warm-starting or conflict reduction?

2. **How does α-Flow perform on other datasets and resolutions?** ImageNet-256 is the only
   evaluation—generalization unclear.

3. **What if you train MeanFlow with just α-Flow's pretraining phase (L_TFM for 150K steps),
   then fine-tune on L_MF?** Does curriculum truly outperform simple pre-training?

4. **Why is the manifold interpretation important?** The paper claims L_TCc has a large manifold,
   but provides no measurement or proof. Does manifold size actually affect optimization?

5. **Can you provide theoretical guarantees for α-Flow?** Does curriculum learning provably
   reduce optimization difficulty or bias?

6. **How does α-Flow perform with other guidance methods** (not just classifier-free guidance)?
   Why does guidance cause instability?

---

> ### Author Response · Authors · 2025-11-23
>
> > W1 & W3 & W4. α-Flow is an incremental generalization; Curriculum learning for generative models is not new; Main contribution is improved MeanFlow, not a fundamental advance;
>
> We kindly disagree with your concern regarding the incremental nature of the $\alpha$-Flow framework. To the best of our knowledge, this is the first framework that connects the recently proposed MeanFlow formulation with the broader family of consistency-based methods, providing a concise and theoretically grounded unification of multiple state-of-the-art one- and few-step diffusion formulations.
>
> Regarding curriculum learning, we believe there may be a misunderstanding. First, it represents only a small portion of our work. Second, it can still be considered novel, as $\alpha$-Flow performs curriculum learning in a new design space, spanning diffusion frameworks (Flow Matching → $\alpha$-Flow → MeanFlow) in a concise and principled manner, rather than applying curriculum learning over jump sizes as in discrete CT. More concretely, the main contributions of our paper are detailed below:
>
> 1. **Theoretical Contribution**. We introduce a unified theoretical framework that systematically encompasses several existing few-step diffusion models. During the rebuttal phase, we further strengthened this contribution by providing additional theoretical analyses on the optimality, bias and variance of these objectives. This framework offers the research community a principled and comprehensive lens through which to study and develop few-step diffusion models.
>
> 2. **Contribution of the $\alpha$-Space for Curriculum Learning.** While we acknowledge that curriculum learning as a concept is not new, the annealing trajectory defined by the $\alpha$ parameter in $\alpha$-Flow represents a distinct and innovative approach. In the rebuttal, we provide additional theoretical justification for this annealing strategy. Empirically, this theoretically grounded methodology achieves state-of-the-art results on the ImageNet dataset, highlighting its practical effectiveness.
>
> 3. **Contribution on Optimization Challenges in MeanFlow.** Our empirical analysis also identifies key optimization challenges in MeanFlow, particularly the gradient conflict between the TFM and TC components. Recognizing and characterizing this conflict constitutes a significant contribution, as it sheds light on the complex optimization landscape of few-step training and informs future research aimed at improving training stability and convergence.

---

> > ### Author Response · Authors · 2025-11-23
> >
> > > W2 & W6 & Q4. The paper hypothesizes: "L_TC has a very large solution manifold compared to L_TFM's narrow manifold" (Section 3.2) but provides no proof or rigorous analysis. Why is the manifold interpretation important?
> >
> > We appreciate your interest in a formal analysis. In Appendix E.1 of the updated version of our paper, we present a new theoretical result showing that the Trajectory Flow Matching loss ($\mathcal L_{\mathtt{TFM}}$) has a unique optimal solution, whereas the Trajectory Consistency loss ($\mathcal{L}_{\mathtt{TC}}$) admits many optimal solutions. This finding supports our original conjecture, at least for global optima rather than $\epsilon$-sublevel sets.
> >
> > 1. **$\mathcal{L}_{\mathtt{TFM}}$ has only one optimal solution:** The Trajectory Flow Matching loss admits only one optimal solution: (please see Equation 23 in the paper, the equation cannot be rendered in openreview). The detailed proof can be found between lines 970 and 1000.
> >
> > 2. **$\mathcal{L}_{\mathtt{TC}}$ has large solution manifold**:  In contrast, the Trajectory Consistency loss $\mathcal{L}_{\mathtt{TC}}$ is an unbounded optimization objective when no constraints are applied.
> > -   **Infinite Solutions:** Any function $\boldsymbol u_{\boldsymbol \theta}(\boldsymbol z_t, r, t)$ satisfying
> >   $$\boldsymbol u_{\boldsymbol \theta}^{\top}(\boldsymbol z_t, r, t) \frac{\mathrm{d} \boldsymbol u_{\boldsymbol \theta^{-}}(\boldsymbol z_t, r, t) }{\mathrm{d} t} \rightarrow -\infty$$
> > 	       will technically minimize the loss toward negative infinity.
> > -   **Ease of Minimization:** This property creates an extremely large solution space, making $\mathcal{L}_{\mathtt{TC}}$ trivially easy to minimize to a very low value.
> > -   **Experimental Evidence:** **Experimental Evidence:** To empirically support this claim, we conducted an experiment minimizing $\mathcal{L}_{\mathtt{TC}}$ using a DiT architecture. The loss was reduced to $-60$ within just 500 training iterations (see Figure 7), confirming its ease of optimization.
> >
> > This observation does not contradict prior findings that CT is generally difficult to optimize. In practice, CT achieves meaningful solutions by explicitly imposing a boundary condition, namely $f(\boldsymbol x_0) = \boldsymbol x_0$, where $\boldsymbol x_0$ is a data point. Introducing this constraint forces $\boldsymbol u_{\boldsymbol \theta}$ to converge to a meaningful, bounded flow rather than diverging toward an unbounded solution. However, this boundary condition also greatly reduces the size of the solution space, making the optimization problem significantly more challenging, consistent with the well-known difficulties in continuous CT literature.
> >
> > **Why is the manifold interpretation important?** This interpretation provides support for our hypothesis regarding the gradient conflict, as discussed in lines 237–241. "We hypothesize this stems from the fact that TC loss, without any boundary condition, has a very large optimal solution manifold, compared to TFM loss whose manifold is very narrow. Thus the optimization process is getting pulled towards the TC loss manifold, distracting from reaching a narrow intersection. "

---

> > > ### Author Response · Authors · 2025-11-23
> > >
> > > > W5. Theorem 1 proof is incomplete. The proof in Appendix D.2 doesn't establish theoretical properties (convergence, optimality, bias-variance); Missing formal analysis of how well finite-$\alpha$ approximates $\alpha$=0.
> > >
> > > We are grateful for your constructive suggestion for the theoretical proof. We have incorporated more theoretical analysis for the $\alpha$-Flow loss (We are sorry that we cannot put any difficult equations due to the render of openreview), including:
> > >
> > > -   **Optimality of $\alpha$-Flow loss**. In section E.2, we prove the optimal solution of $\alpha$-Flow loss, and show that when $\alpha \to 0$, the optimal solution of $\alpha$-Flow loss converges to the optimal solution of MeanFlow loss. Specifically, under Assumption 1, the optimal solution for Trajectory Flow matching loss is in Line 998, Equation 23.
> > > The optimal solution for MeanFlow loss is in Line 1037, Equation 29.
> > > And the optimal solution for $\alpha$-Flow in Line 1073, Equation 36.
> > >
> > > - Furthermore, we study the **non-asymptotic distance between the optimal solution from $\alpha$-Flow and MeanFlow loss**. In section E.2, under Assumption 2, we have Equation 46 in Line 1125. Specifically, the upper bound of the distance decreases as $\alpha$ decreases, and it vanishes to 0 when $\alpha \to 0$
> > >
> > > - **Non-asymptotic distance between gradient $\alpha$-Flow and Meanflow**. In section E.3, under assumption 2, we proved an upper bound of the distance in Line 1174 Equation 49. The upper bound is linearly dependent on $\alpha$, and will vanish to 0 when $\alpha \rightarrow 0$, align with our Theorem 1.
> > >
> > > - **Variance of the $\alpha$-Flow gradient**. In section E.4, under assumption 2, we theoretically proved an upper bound of the variance of the gradient in Line 1204. This can be further verified by Figure 8 that  the variance of the gradient norm of AlphaFlow is significantly lower than the gradient for MeanFlow. This helps to support the AlphaFlow annealing moves from a “high-bias, low-variance” objective to a “high-variance, low-bias” objective.
> > >
> > > > W7. Notation could be clearer
> > >
> > > Thanks for your helpful comments of notations. We use different notations for $L_{\mathtt{TFM}}$ and $L_{\mathtt{FM}'}$ to emphasize that they represent two distinct loss functions, as shown in Equation 7. Specifically, $L_{\mathtt{TFM}}$ is derived from the decomposition of the MeanFlow loss (Equation 6), while $L_{\mathtt{FM}'}$ corresponds to the case where $r = t$ in the MeanFlow loss.
> > >
> > > Regarding “$L_{\mathtt{TCc}}$,” the “TC” denotes the trajectory consistency loss, and the subscript “c” originally indicated the continuous form, corresponding to the “c” in “$L_{\mathtt{CTc}}$” in Equation 3. For clarity and simplicity, we have updated all instances of “$L_{\mathtt{TCc}}$” to “$L_{\mathtt{TC}}$.”
> > >
> > > > W8. Missing experimental details
> > >
> > > We appreciate your careful reading. We use cosine similarity, which is defined as:
> > > $$\mathtt{cosine\ similarity}\left(\boldsymbol a, \boldsymbol b\right) = \frac{\boldsymbol a \cdot \boldsymbol b}{\left\Vert\boldsymbol a\right\Vert\left\Vert\boldsymbol b\right\Vert},$$
> > > given two vectors $\boldsymbol a, \boldsymbol b$. We have incorporated it in Appendix F. We use 1000 iterations to average the plot in Figure 3 (previously Figure 2).

---

> > > > ### Author Response · Authors · 2025-11-23
> > > >
> > > > > Q1 & Q5. Why does curriculum learning specifically help? Can you show that gradient conflict decreases during $\alpha$-annealing? Is it warm-starting or conflict reduction? Can you provide theoretical guarantees for $\alpha$-Flow?
> > > >
> > > > We thank the reviewer for the thoughtful question towards the mechanism of $\alpha$-Flow training. We note that $\alpha$-Flow resolves the gradient conflict not by directly reducing its magnitude, but by **avoiding** it in the first place: initially minimizing the TFM loss term in isolation during the early stage of training, and then optimizing along an annealing trajectory that gradually approaches the optimal solution of MeanFlow. We emphasize that $\alpha$-Flow achieves this in a principled and theoretically grounded manner by reducing gradient variance while preserving the optimality condition. A visual illustration of the $\alpha$-Flow optimization trajectory is provided in Figure 2 of the updated paper.
> > > >
> > > > During the rebuttal phase, we add detailed theoretical analyses addressing the following points:
> > > >
> > > > - **Preserving the optimality condition.** The optimal solution of $\alpha$-Flow transitions smoothly from that of TFM to that of MeanFlow as $\alpha$ decreases from 1 to 0 (proved in Appendix E.2), confirming that the annealing trajectory continuously transforms from TFM to MeanFlow.
> > > >
> > > > - **Reducing the gradient variance.** The upper bound on the gradient variance for $\alpha$-Flow increases as $\alpha$ decreases (proved in Section E.4), providing theoretical support for the empirical observation by [1] that the annealing trajectory progresses from a “high-bias, low-variance” objective to a “low-bias, high-variance” one. This transition enables more stable convergence compared to directly optimizing a “low-bias, high-variance” objective.
> > > >
> > > > [1] Song, Yang, Prafulla Dhariwal, Mark Chen, and Ilya Sutskever. "Consistency models." (2023).
> > > >
> > > > > Q2. How does $\alpha$-Flow perform on other datasets and resolutions?
> > > >
> > > > |  Model  |  NFE  1  FID  |  NFE  1  FDD  |  NFE  1  FVD  |  NFE  2  FID  |  NFE  2  FDD  |  NFE  2  FVD  |
> > > > |-----------------------|-----------|-----------|-----------|-----------|-----------|-----------|
> > > > |  MeanFlow-B/2  |  53.2  |  1031.9  |  619.5  |  49.5  |  995.3  |  580.8  |
> > > > |  $\alpha$-Flow-B/2  |  **50.7**  |  **1010.4**  |  **598.0**  |  **48.1**  |  **983.2**  |  **567.3**  |
> > > > |  MeanFlow-B/2-cfg  |  32.3  |  779.0  |  298.1  |  26.6  |  719.8  |  249.6  |
> > > > |  $\alpha$-Flow-B/2-cfg  |  **29.6**  |  **709.8**  |  **276.2**  |  **24.9**  |  **657.4**  |  **217.3**  |
> > > >
> > > > We appreciate your valuable suggestion. To demonstrate the generalization capability of $\alpha$-Flow, we extend the framework to a higher-resolution and different modality: video. Specifically, we use the Kinetics-700 dataset [1], where each video has dimensions of $256 \times 256 \times 17$ (resolution $256 \times 256$ with 17 frames). We employ the DiT-B/2 architecture and train the model for 600K iterations, both with and without CFG. For CFG training, we set the guidance scale to $\omega = 1.0$ and $\kappa = 0.7$. In addition to the standard FID and FDD metrics, we use the Fréchet Video Distance (FVD) [2] to evaluate video quality, where lower values indicate better results.
> > > >
> > > > As shown in the table, $\alpha$-Flow consistently outperforms MeanFlow across all metrics and settings. Without CFG, $\alpha$-Flow achieves improvements of 2.5 (NFE 1 FID), 21.5 (NFE 1 FDD), 21.5 (NFE 1 FVD), 1.4 (NFE 2 FID), 12.1 (NFE 2 FDD), and 13.5 (NFE 2 FVD). With CFG, $\alpha$-Flow achieves even larger gains of 2.7 (NFE 1 FID), 69.2 (NFE 1 FDD), 21.9 (NFE 1 FVD), 1.7 (NFE 2 FID), 62.4 (NFE 2 FDD), and 32.3 (NFE 2 FVD).
> > > >
> > > > These results clearly demonstrate the effectiveness and generalization of the $\alpha$-Flow framework across diverse datasets and modalities. We have incorporated the corresponding experiments and discussions in Section H.6.
> > > >
> > > > [1] Carreira, Joao, Eric Noland, Chloe Hillier, and Andrew Zisserman. "A short note on the kinetics-700 human action dataset." arXiv preprint arXiv:1907.06987 (2019).
> > > >
> > > > [2] Unterthiner, Thomas, Sjoerd Van Steenkiste, Karol Kurach, Raphaël Marinier, Marcin Michalski, and Sylvain Gelly. "FVD: A new metric for video generation." (2019).

---

> ### Author Response · Authors · 2025-11-23
>
> > Q3. What if you train MeanFlow with just α-Flow's pretraining phase (L_TFM for 150K steps), then fine-tune on L_MF? Does curriculum truly outperform simple pre-training?
>
> This is really great suggestion! We conduct experiments comparing $\alpha$-Flow with MeanFlow fine-tuning, starting from pre-trained Flow Matching (FM), Trajectory Flow Matching (TFM), and FM + Representation Alignment (REPA) models. For all experiments, we use DiT-B/2-non-cfg configurations in Table 3 as the backbone models. The FM, TFM, and FM + REPA models were all pre-trained by us.
>
> | Pretraining Model | Fine-Tuning Model| NFE 1 FID | NFE 1 FDD | NFE 2 FID | NFE 2 FDD |
> |---|---|---|---|---|---|
> | $\text{FM}_{0K \to 200K}$ | $\text{MeanFlow}_{200K \to 400K}$ | 42.3 | 798.1 | 37.6 | **772.4** |
> | $\text{TFM}_{0K \to 200K}$ | $\text{MeanFlow}_{200K \to 400K}$ | 42.3 | 821.8 | 37.2 | 780.2 |
> | $\text{Sigmoid}_{0K \to 400K}$ |   | **40.2** | **781.0** | **37.1** | 775.0 |
> | $\text{REPA}_{0K \to 400K}$ | $\text{MeanFlow}_{400K \to 600K}$ | 32.0 | 667.4 | 25.5 | **594.3** |
> | $\text{REPA}_{0K \to 400K}$ | $\text{Sigmoid}_{400K \to 600K}$ | **30.7** | **657.1** | **25.3** | 601.5 |
>
> When comparing results epoch by epoch using the same pre-trained initialization, $\alpha$-Flow consistently outperforms MeanFlow in one-step generation (NFE 1) and achieves comparable performance in two-step generation. Specifically, for NFE 1, $\alpha$-Flow (FID 40.2, FDD 781.0) delivers improvements, outperforming the FM + MeanFlow baseline by 2.1 FID and 17.1 FDD, and surpassing the TFM + MeanFlow baseline by 2.1 FID and 40.8 FDD. Furthermore, when initialized with the same FM + REPA model, $\alpha$-Flow (FID 30.7, FDD 657.1) continues to outperform MeanFlow by 1.3 FID and 10.3 FDD in one-step generation.
>
> Because $\alpha$-Flow inherently involves three sequential stages: TFM pre-training, $\alpha$-Flow annealing, and MeanFlow fine-tuning. These results clearly demonstrate the effectiveness of the $\alpha$-Flow annealing stage. This stage plays a crucial role in enhancing the quality of one-step sampling. Full results and detailed experimental configurations are provided in Section H.4.
>
>
> > Q6. How does α-Flow perform with other guidance methods (not just classifier-free guidance)? Why does guidance cause instability?
>
> In our preliminary experiments, we explore ways to simplify and redesign the CFG mechanism in MeanFlow and investigate alternative formulations of the guidance process. Specifically, we experiment with **$u$-target guidance**, whose detailed definition is provided in Equation 50 of Section H.8. The motivation behind this approach is to use $\boldsymbol u_{\boldsymbol \theta}(\boldsymbol z_t, r, t | \varnothing)$ for the unconditional prediction, rather than the unconditional prediction $\boldsymbol u_{\boldsymbol \theta}(\boldsymbol z_t, t, t| \varnothing)$ used in standard CFG guidance. With the reviewer’s kind permission, we provide the corresponding results below:
>
> |  Model  |  NFE  1  FID  |  NFE  1  FDD  |  NFE  1  FVD  |  NFE  2  FID  |
> |---|---|---|---|---|
> |  $u$-target  guidance  |  25.1  |  645.1  |  9.5  |  392.7  |
> |  CFG  guidance  |  **10.0**  |  **445.8**  |  **7.7**  |  **365.2**  |
>
> Although this method yields worse results than the vanilla MeanFlow guidance (which is why we initially omitted these negative results from the submission), it nonetheless demonstrates that $\alpha$-Flow remains compatible with alternative guidance formulations. This finding should addresses the reviewer’s concern about $\alpha$-Flow’s robustness to different guidance mechanisms.
>
> As our paper does not primarily focus on guidance strategies for few-step diffusion models, we adopt the same guidance mechanism used in MeanFlow for consistency. If the reviewer has other guidance approaches in mind, we would be happy to conduct further experiments accordingly.
>
> **Hypothesis for the instability of guidance**. In CFG training, $\boldsymbol v_t$ is replaced to $$\omega \boldsymbol v_t + \kappa \cdot \boldsymbol u_{\boldsymbol \theta^{-}}\left(\boldsymbol z_t, t, t|\boldsymbol c\right) + \left(1 - \omega - \kappa\right) \cdot \boldsymbol u_{\theta^{-}}\left(\boldsymbol z_t, t, t|\varnothing\right)$$
>
> Because the network cannot perfectly capture the complex data manifold, its outputs (the $\boldsymbol u_{\boldsymbol \theta^{-}}$ terms) inevitably contain noise and prediction errors. This substitution makes the overall training target noisier and less stable, which we hypothesize to be a key factor contributing to the instability observed during guidance training.

---

### Official Review · Reviewer_gT18 · 2025-10-27

**Soundness:** 3
**Presentation:** 3
**Contribution:** 2
**Rating:** 6
**Confidence:** 4

**Summary:**

This paper provides a detailed analysis of the MeanFlow objective for few-step generative modeling. Motivated by the observation that MeanFlow dedicates the majority of its training to border-case supervision (i.e., fitting to vanilla flow matching with $r=t$), the authors conduct in-depth investigation into the MeanFlow objective. Specifically, they show that the MeanFlow loss naturally decomposes into two components — trajectory flow matching ($L_{TFM}$) and trajectory consistency ($L_{TC_c}$) — whose gradients are strongly negatively correlated, leading to optimization conflicts. The authors further discover that such issue can be partially mitigated by introducing an auxiliary flow matching loss ($L_{FM'}$).

Intended for a training recipe with higher efficiency, the authors propose $\alpha-$Flow, a unified framework that interpolates between flow matching and MeanFlow via a curriculum parameter $\alpha$. The framework includes several prior objectives (Flow Matching, Shortcut Model, Consistency Training) as special cases, and progressively transitions from $L_{TFM}$ to MeanFlow through an $\alpha-$schedule. This framework achieves improved performance on ImageNet $256\times256$ compared to various previous baselines under comparable compute budgets.

**Strengths:**

1. **Insightful empirical analysis**

The decomposition of the MeanFlow objective and the empirical analysis of gradient correlations in Section 3 provide a novel perspective on the training dynamics of the trajectory-based generative modeling. These analysis not only provide clear motivations for the proposal of $\alpha-$Flow, but also effectively deepens the understanding of optimization behavior in few-step generative models, which is timely and valuable given the growing research focus on this field.

2. **Comprehensive and clear ablation studies**

For each design choice introduced in the proposed $\alpha-$Flow framework, the authors conduct extensive ablation experiments accompanied by clear explanations that justify their settings. These analyses substantially improve the clarity of presentation, strengthen the empirical grounding of their claims, and enhance the reproducibility of the overall training recipe.

3. **Improved empirical results**

Across all comparable settings, $\alpha-$Flow demonstrate superior performance over multiple baselines, including MeanFlow, FACM, and Shortcut Models. The improvements are consistent across different model scales and sampling steps, underscoring the effectiveness and robustness of the proposed framework.

4. **Detailed limitations and failed experiment discussions**

It is commendable that the authors explicitly discuss both the limitations of their framework and the failed experimental attempts (Appendix B–C). Such transparency is rare and valuable: it not only helps future researchers avoid redundant directions but also provides practical insights into the challenges of optimizing few-step generative models at scale. This openness meaningfully contributes to the community’s collective understanding and progress.

**Weaknesses:**

1. **Stability of $L_{TC_c}$**

The proposed method does not directly address the well-known stability issue of the $L_{TC_c}$, which the authors also acknowledge in Appendix B. This instability has been repeatedly identified as the primary bottleneck in scaling consistency-based methods to large-scale applications such as text-to-image or text-to-video generation (Lu & Song, 2024; Chen et al., 2025; Zheng et al., 2025). As a result, while the paper offers valuable insights into the gradient conflict within the MeanFlow objective and presents improved training heuristics, the overall scope remains somewhat limited. That said, since the authors explicitly note that “our framework should not be viewed as a silver bullet for addressing instability,” it would nonetheless strengthen the paper to include an empirical comparison of training stability between α-Flow and vanilla MeanFlow (e.g., visualizations of training losses, gradient norms, or JVP magnitudes as $\alpha \to 0$).

2. **Comparisons with contemporaneous works**

In Section 4.1, the authors demonstrate that $\alpha-$Flow subsumes Consistency Training (CT) under certain network parameterizations. However, the paper does not include corresponding CT reproduction results in Table 1. Including such a baseline would substantiate the claim that $\alpha-$Flow not only unifies but also improves upon prior consistency-based methods.

Moreover, the proposed framework shares conceptual similarities with SplitMeanFlow (Guo et al., 2025). While the authors briefly mention this work in the appendix, a more direct discussion or empirical comparison—highlighting differences in optimization behavior, efficiency, or loss design—would enhance the paper’s soundness and contextual positioning within the recent literature.

3. **Supporting evidence of "$L_{TC_c}$ is easier to optimize"**

The paper claims that $L_{TC_c}$ has a “larger solution manifold” and is therefore easier to optimize, but this assertion appears counterintuitive, as the JVP term in consistency objectives are generally known to suffer from great instability during training. A more rigorous justification (or some empirical results) would make the argument more convincing and clarify the optimization dynamics of $L_{TC_c}$.

Lu, Cheng, and Yang Song. "Simplifying, stabilizing and scaling continuous-time consistency models." arXiv preprint arXiv:2410.11081 (2024).

Chen, Junsong, Shuchen Xue, Yuyang Zhao, Jincheng Yu, Sayak Paul, Junyu Chen, Han Cai, Song Han, and Enze Xie. "Sana-sprint: One-step diffusion with continuous-time consistency distillation." arXiv preprint arXiv:2503.09641 (2025).

Zheng, Kaiwen, Yuji Wang, Qianli Ma, Huayu Chen, Jintao Zhang, Yogesh Balaji, Jianfei Chen, Ming-Yu Liu, Jun Zhu, and Qinsheng Zhang. "Large Scale Diffusion Distillation via Score-Regularized Continuous-Time Consistency." arXiv preprint arXiv:2510.08431 (2025).

Guo, Yi, Wei Wang, Zhihang Yuan, Rong Cao, Kuan Chen, Zhengyang Chen, Yuanyuan Huo et al. "Splitmeanflow: Interval splitting consistency in few-step generative modeling." arXiv preprint arXiv:2507.16884 (2025).

**Questions:**

1. **Distillation applicability**

Could the proposed $\alpha-$Flow framework be extended to a *distillation* setting, where the conditional velocity $v_t$ is replaced by a teacher model’s prediction? If so, would this substitution affect any design choices?

2. **Counterintuitive claim on $L_{TC_c}$'s optimization**

It is well known that the JVP term in the consistency objectives introduces strong training instability and is notoriously difficult to optimize. But in the paper, the authors argue that, without a boundary condition, $L_{TC_c}$ is easier to optimize given its unconstraint solution space. Could the authors elaborate on this seemingly contradiction? Is this claim mainly referring to that it's easy to optimize $L_{TC_c}$ to 0 but not to the solution space intersecting with $L_{TFM}$?

3. **Adaptive weighting strategy**

For $\alpha = 1$, the derived adaptive weighting is applied on the vanilla flow matching objective, which could potentially break the L2 loss required by the flow matching training recipe; yet, for $\alpha \to 0$, no adaptive weighting is not applied to the more challenging JVP-based objective. Although this empirically yields positive results, the design choice seems counterintuitive. Could the authors offer further intuition or justification for why this adaptive weighting behaves favorably in practice?

4. **Fine-tuning the model**

The trajectory flow matching phase in $\alpha-$Flow. effectively serves as a warm-up for stabilizing the model from random initialization. Have the authors experimented with starting from a pretrained FM model instead?

---

> ### Author Response · Authors · 2025-11-23
>
> > W1. Empirical comparison of training stability between α-Flow and vanilla MeanFlow.
>
> We appreciate your suggestion and have included the variance of the loss and gradient norm visualizations as Figure 8 in Appendix H.7.
>
> Figure 8 shows that both the loss variance and gradient norm variance for $\alpha$-Flow are lower than those of MeanFlow during the trajectory flow matching (TFM) pretraining and $\alpha$-Flow annealing stages (to the left of the blue dotted line). This indicates that $\alpha$-Flow exhibits a more stable training process in these early phases.
>
> However, we note that while the $\alpha$-Flow formulation maintains substantially lower loss and gradient variance throughout most of its training, its final stage reverts to pure MeanFlow optimization ($\alpha \rightarrow 0$). As a result, $\alpha$-Flow cannot fully resolve the inherent instability of the MeanFlow objective, leaving this as an open direction for future research.
>
> > W2. Comparisons with CT and Discussion with SplitMeanFlow.
>
> |  Model  |  Epochs  |  NFE  1  FID  |  NFE  1  FDD  |  NFE  2  FID  |  NFE  2  FDD  |
> |--------------------|--------|-----------|-----------|-----------|-----------|
> |  CT-XL/2  |  240  |  7.44  |  324.9  |  6.22  |  271.9  |
> |  $\alpha$-FLow-XL/2  |  240  |  **2.95**  |  **164.6**  |  **2.34**  |  **105.7**  |
>
> We are grateful to your suggestion of related work comparison. We conducted an experiment on the CT method using the DiT-XL/2 architecture by applying the reparameterization and setting $r = 0$, as shown in the table below.
>
> The results indicate that, without the advanced techniques proposed in the sCM paper and when using the DiT backbone, CT training was unstable and yielded performance inferior to that of $\alpha$-Flow. Specifically, $\alpha$-Flow outperforms CT by 4.49 FID and 160.3 FDD for NFE 1, and by 3.88 FID and 166.2 FDD for NFE 2. We have incorporated these experimental results into Table 1.
>
> While concurrent work [1] introduced a loss function similar to our $\alpha$-Flow loss, our paper provides distinct theoretical and empirical contributions. Theoretically, we offer a more in-depth analysis comparing the $\alpha$-Flow loss with the MeanFlow loss. Empirically, we propose novel techniques specifically designed to enhance the performance of the $\alpha$-Flow loss, and we demonstrate its effectiveness and the advantages of curriculum learning on the large-scale ImageNet dataset. We have added this clarification in Appendix A.
>
> [1] Guo, Yi, Wei Wang, Zhihang Yuan, Rong Cao, Kuan Chen, Zhengyang Chen, Yuanyuan Huo et al. "Splitmeanflow: Interval splitting consistency in few-step generative modeling." _arXiv preprint arXiv:2507.16884_ (2025).

---

> ### Author Response · Authors · 2025-11-23
>
> > W3 & Q2. A more rigorous justification that $\mathcal L_{\mathtt{TC}}$ has a “larger solution manifold”.
>
> Thanks for your valuable question. In Appendix E.1 of the updated version of our paper, we present a new theoretical result showing that the Trajectory Flow Matching loss ($\mathcal L_{\mathtt{TFM}}$) has a unique optimal solution, whereas the Trajectory Consistency loss ($\mathcal{L}_{\mathtt{TC}}$) admits many optimal solutions. This finding supports our original conjecture, at least for global optima rather than $\epsilon$-sublevel sets.
>
> 1. **$\mathcal{L}_{\mathtt{TFM}}$ has only one optimal solution:** The Trajectory Flow Matching loss admits only one optimal solution: (please see Equation 23 in the paper, the equation cannot be rendered in openreview). The detailed proof can be found between lines 970 and 1000.
>
> 2. **$\mathcal{L}_{\mathtt{TC}}$ has large solution manifold**:  In contrast, the Trajectory Consistency loss $\mathcal{L}_{\mathtt{TC}}$ is an unbounded optimization objective when no constraints are applied.
> -   **Infinite Solutions:** Any function $\boldsymbol u_{\boldsymbol \theta}(\boldsymbol z_t, r, t)$ satisfying
>   $$\boldsymbol u_{\boldsymbol \theta}^{\top}(\boldsymbol z_t, r, t) \frac{\mathrm{d} \boldsymbol u_{\boldsymbol \theta^{-}}(\boldsymbol z_t, r, t) }{\mathrm{d} t} \rightarrow -\infty$$
> 	       will technically minimize the loss toward negative infinity.
> -   **Ease of Minimization:** This property creates an extremely large solution space, making $\mathcal{L}_{\mathtt{TC}}$ trivially easy to minimize to a very low value.
> -   **Experimental Evidence:** **Experimental Evidence:** To empirically support this claim, we conducted an experiment minimizing $\mathcal{L}_{\mathtt{TC}}$ using a DiT architecture. The loss was reduced to $-60$ within just 500 training iterations (see Figure 7), confirming its ease of optimization.
>
> This observation does not contradict prior findings that CT is generally difficult to optimize. In practice, CT achieves meaningful solutions by explicitly imposing a boundary condition, namely $f(\boldsymbol x_0) = \boldsymbol x_0$, where $\boldsymbol x_0$ is a data point. Introducing this constraint forces $\boldsymbol u_{\boldsymbol \theta}$ to converge to a meaningful, bounded flow rather than diverging toward an unbounded solution. However, this boundary condition also greatly reduces the size of the solution space, making the optimization problem significantly more challenging, consistent with the well-known difficulties in continuous CT literature.
>
> > Q1. Could the proposed $\alpha$-Flow framework be extended to a distillation setting? Would this substitution affect any design choices?
>
> |  Model  |  NFE  1  FID  |  NFE  1  FDD  |  NFE  2  FID  |  NFE  2  FDD  |
> |----------------------|-----------|-----------|-----------|-----------|
> |  $\text{Sigmoid}_{0K \to 100K}$  |  **36.6**  |  756.9  |  **33.2**  |  **701.3**  |
> |  $\text{Sigmoid}_{50K \to 150K}$  |  37.1  |  749.2  |  33.4  |  702.9  |
> |  $\text{Sigmoid}_{100K \to 200K}$  |  37.8  |  757.5  |  33.5  |  704.7  |
> |  $\text{Sigmoid}_{150K \to 250K}$  |  36.7  |  743.2  |  34.1  |  706.9  |
> |  $\text{Sigmoid}_{0K \to 400K}$  |  36.9  |  **738.9**  |  33.6  |  704.3  |
>
>
> We conduct a distillation experiment in which the $\alpha$-Flow model (using the DiT-B/2 architecture) is distilled from a DiT-B/2-REPA teacher model trained for 400K iterations.
>
> The experimental results, summarized in the table, clearly show that distillation improves the performance of $\alpha$-Flow for both NFE 1 and NFE 2 settings. The distilled $\alpha$-Flow model, trained with the Sigmoid schedule from 0K to 100K (first row in the table above), outperforms the $\alpha$-Flow model trained from scratch with the Sigmoid schedule from 0K to 400K  (last row of Table 2(a) in the paper). For NFE 1, the distilled model achieves improvements of 3.4 in FID and 28.5 in FDD. For NFE 2, it achieves even greater gains of 3.9 in FID and 81.6 in FDD.
>
> Notably, the optimal distillation schedule shifts to $\text{Sigmoid}_{0K \to 100K}$. This shift is expected, as the teacher model (DiT-B/2-REPA) is already pretrained. The pretrained knowledge allows the distillation process to bypass the initial $\alpha = 1$ training phase required when training $\alpha$-Flow from scratch, thereby reducing the total number of required TFM pre-training steps.
>
> We integrate the experimental results and corresponding discussion in Section H.5.

---

> > ### Author Response · Authors · 2025-11-23
> >
> > > Q3. Question about the Adaptive weighting strategy for the flow matching objective and the JVP-based objective
> >
> > “the derived adaptive weighting is applied on the vanilla flow matching objective, which could potentially break the L2 loss required by the flow matching training recipe”
> >
> > The adaptive weighting mechanism can be interpreted as a reweighting function based on the loss value. In practice, we observe that the loss values for vanilla Flow Matching exhibit low variance. As a result, the weights produced by the adaptive mechanism remain nearly constant across samples, meaning that the standard $L_2$ loss objective is effectively preserved.
> >
> >
> > |  Model  |  NFE  80  FID  |  NFE  80  FDD  |
> > |---|---|---|
> > |  TFM  with  adaptive  weighting  |  60.9  |  1001.8  |
> > |  TFM  without  adaptive  weighting  |  61.2  |  1002.8  |
> >
> > To further validate this observation, we train the DiT-B/2 architecture using the trajectory flow matching loss with and without adaptive weighting for 100K iterations. The results, shown in the table, indicate that both settings achieve nearly identical performance (60.9 vs. 61.2 for NFE 80 FID and 1001.8 vs. 1002.8 for NFE 80 FDD).
> >
> > Therefore, the adaptive weighting mechanism does not interfere with the standard $L_2$ loss objective. We apply this strategy when $\alpha = 1$ to maintain consistency across the entire $\alpha$-Flow framework for all $\alpha$ values.
> >
> > “no adaptive weighting is applied to the more challenging JVP-based objective.”
> >
> > We respectfully disagree with this remark, as we apply adaptive weighting specifically to the challenging JVP-based objective. The weighting function is defined as $\alpha / \left( |\Delta|_2^2 + c \right)$ (lines 1334–1335 in the paper). After reweighting by $1 / \left( |\Delta|_2^2 + c \right)$, we assign a small weight $\alpha$ as $\alpha \rightarrow 0$. Intuitively, this ensures that the model takes smaller gradient steps, which is crucial for maintaining stability during the final annealing phase.
> >
> >
> > > Q4. Have the authors experimented with starting from a pretrained FM model instead?
> >
> > Thats a really great suggestion! We conduct experiments comparing $\alpha$-Flow with MeanFlow fine-tuning, starting from pre-trained Flow Matching (FM), Trajectory Flow Matching (TFM), and FM + Representation Alignment (REPA) models. For all experiments, we use DiT-B/2-non-cfg configurations in Table 3 as the backbone models. The FM, TFM, and FM + REPA models were all pre-trained by us.
> >
> > | Pretraining Model | Fine-Tuning Model| NFE 1 FID | NFE 1 FDD | NFE 2 FID | NFE 2 FDD |
> > |---|---|---|---|---|---|
> > | $\text{FM}_{0K \to 200K}$ | $\text{MeanFlow}_{200K \to 400K}$ | 42.3 | 798.1 | 37.6 | **772.4** |
> > | $\text{TFM}_{0K \to 200K}$ | $\text{MeanFlow}_{200K \to 400K}$ | 42.3 | 821.8 | 37.2 | 780.2 |
> > | $\text{Sigmoid}_{0K \to 400K}$ |   | **40.2** | **781.0** | **37.1** | 775.0 |
> > | $\text{REPA}_{0K \to 400K}$ | $\text{MeanFlow}_{400K \to 600K}$ | 32.0 | 667.4 | 25.5 | **594.3** |
> > | $\text{REPA}_{0K \to 400K}$ | $\text{Sigmoid}_{400K \to 600K}$ | **30.7** | **657.1** | **25.3** | 601.5 |
> >
> > When comparing results epoch by epoch using the same pre-trained initialization, $\alpha$-Flow consistently outperforms MeanFlow in one-step generation (NFE 1) and achieves comparable performance in two-step generation. Specifically, for NFE 1, $\alpha$-Flow (FID 40.2, FDD 781.0) delivers improvements, outperforming the FM + MeanFlow baseline by 2.1 FID and 17.1 FDD, and surpassing the TFM + MeanFlow baseline by 2.1 FID and 40.8 FDD. Furthermore, when initialized with the same FM + REPA model, $\alpha$-Flow (FID 30.7, FDD 657.1) continues to outperform MeanFlow by 1.3 FID and 10.3 FDD in one-step generation.
> >
> > Because $\alpha$-Flow inherently involves three sequential stages: TFM pre-training, $\alpha$-Flow annealing, and MeanFlow fine-tuning. These results clearly demonstrate the effectiveness of the $\alpha$-Flow annealing stage. This stage plays a crucial role in enhancing the quality of one-step sampling. Full results and detailed experimental configurations are provided in Section H.4.

---

### Official Review · Reviewer_S4Q1 · 2025-10-29

**Soundness:** 3
**Presentation:** 3
**Contribution:** 3
**Rating:** 6
**Confidence:** 4

**Summary:**

The authors propose to train a MeanFlow model along a curriculum that transitions from flow matching to MeanFlow optimization. This curriculum is inspired by two observations: the MeanFlow objective can be decomposed into a flow matching loss and a consistency loss, and the two losses have conflicting gradients during training. By optimizing the flow matching loss (with a narrow solution space) first and then transitioning to the easier MeanFlow objective (with a larger solution space), the authors claim one can achieve better convergence. The curriculum is implemented by optimizing the $\alpha$-Flow objective that interpolates between flow matching (at $\alpha=1$) and MeanFlow (at $\alpha=0$). Various annealing strategies for $\alpha$ are explored in the paper, and $\alpha$-Flow in general outperforms vanilla MeanFlow.

**Strengths:**

- **[S1] This paper is novel, as it presents a new perspective on the slow convergence of MeanFlow.** By decomposing the MeanFlow objective into a sum of flow matching and consistency training losses and showing that the two have conflicting gradients, this paper elucidates a possible source of instability and slow convergence in MeanFlow training.

- **[S2] This paper is significant, as it provides a simple and practical recipe for stable training of MeanFlow.** By pinpointing a source of instability in MeanFlow training, the authors are able to develop a simple curriculum learning strategy for stabilizing convergence of MeanFlow. The authors provide strong evidence (on ImageNet 256$\times$256) that the curriculum indeed yields better MeanFlow models. In particular, experiments show one can use a lower flow matching ratio with $\alpha$-Flow, implying that the proposed curriculum effectively mitigates the tension between flow matching and consistency training.

**Weaknesses:**

- **[W1] Experiments in Section 5 only indirectly support the claim that $\alpha$-Flow mitigates the conflict between trajectory flow matching (TFM) and trajectory consistency (TC) objective.** An experiment such as an analogue of Figure 2 for $\alpha$-Flow training could directly support the hypothesis that the proposed curriculum resolves the tension between TFM and TC.

- **[W2] The paper is missing a simple but crucial baseline: training from a pre-trained diffusion model.** In particular, with diffusion models (DMs) trained via representation alignment offering much better generative performance, initialization from such DMs may be a simpler alternative to $\alpha$-Flow without the added complexity of tuning the alpha decay schedule.

**Questions:**

- **[Q1] Why is there a large gap between FIDs in Table 1 and Table 2?** Did the authors use smaller number of training epochs for ablations?

- **[Q2] How does $\alpha$-Flow training from scratch compare to MeanFlow training initialized from a pre-trained diffusion / flow model?** I am curious how MeanFlow initialized with e.g., REPA [1] or REPA-E [2] DM would perform compared to $\alpha$-Flow. I believe the embedding layer for the second time parameter can easily be implemented by copying embedding modules for the original time parameter in pre-trained DMs.

[1] Representation Alignment for Generation: Training Diffusion Transformers Is Easier Than You Think, ICLR, 2025

[2] REPA-E: Unlocking VAE for End-to-End Tuning with Latent Diffusion Transformers, 2025

---

> ### Author Response · Authors · 2025-11-23
>
> >W1. An experiment analogous to Figure 2 for $\alpha$-Flow training could directly support the hypothesis that the proposed curriculum resolves the tension between TFM and TC.
>
> We note that $\alpha$-flow resolves the gradient conflict not by directly reducing its magnitude, but by **avoiding** it in the first place by minimizing the TFM loss term in isolation during the early stage of training. Optimization then proceeds along an annealing trajectory that gradually approaches the optimal solution of MeanFlow. $\alpha$-flow accomplishes this in a **principled, theoretically justified way** by reducing the gradient variance (proved in Appendix E.4) while preserving the optimality condition (proved in Appendix E.2). We've incorporated a visual illustration of the $\alpha$-flow optimization trajectory in Figure 2 of the updated paper.
>
> > W2 & Q2. How does Alpha-Flow training from scratch compare to MeanFlow training that is initialized from a pre-trained diffusion or flow model?
>
> Thanks for your great suggestion! We conduct experiments comparing $\alpha$-Flow with MeanFlow fine-tuning, starting from pre-trained Flow Matching (FM), Trajectory Flow Matching (TFM), and FM + Representation Alignment (REPA) models. For all experiments, we use DiT-B/2-non-cfg configurations in Table 3 as the backbone models. The FM, TFM, and FM + REPA models were all pre-trained by us.
>
> | Pretraining Model | Fine-Tuning Model| NFE 1 FID | NFE 1 FDD | NFE 2 FID | NFE 2 FDD |
> |---|---|---|---|---|---|
> | $\text{FM}_{0K \to 200K}$ | $\text{MeanFlow}_{200K \to 400K}$ | 42.3 | 798.1 | 37.6 | **772.4** |
> | $\text{TFM}_{0K \to 200K}$ | $\text{MeanFlow}_{200K \to 400K}$ | 42.3 | 821.8 | 37.2 | 780.2 |
> | $\text{Sigmoid}_{0K \to 400K}$ |   | **40.2** | **781.0** | **37.1** | 775.0 |
> | $\text{REPA}_{0K \to 400K}$ | $\text{MeanFlow}_{400K \to 600K}$ | 32.0 | 667.4 | 25.5 | **594.3** |
> | $\text{REPA}_{0K \to 400K}$ | $\text{Sigmoid}_{400K \to 600K}$ | **30.7** | **657.1** | **25.3** | 601.5 |
>
> When comparing results epoch by epoch using the same pre-trained initialization, $\alpha$-Flow consistently outperforms MeanFlow in one-step generation (NFE 1) and achieves comparable performance in two-step generation. Specifically, for NFE 1, $\alpha$-Flow (FID 40.2, FDD 781.0) delivers improvements, outperforming the FM + MeanFlow baseline by 2.1 FID and 17.1 FDD, and surpassing the TFM + MeanFlow baseline by 2.1 FID and 40.8 FDD. Furthermore, when initialized with the same FM + REPA model, $\alpha$-Flow (FID 30.7, FDD 657.1) continues to outperform MeanFlow by 1.3 FID and 10.3 FDD in one-step generation.
>
> Because $\alpha$-Flow inherently involves three sequential stages: TFM pre-training, $\alpha$-Flow annealing, and MeanFlow fine-tuning. These results clearly demonstrate the effectiveness of the $\alpha$-Flow annealing stage. This stage plays a crucial role in enhancing the quality of one-step sampling. Full results and detailed experimental configurations are provided in Section H.4.
>
>
> > Q1. Why is there a large gap between the FID scores in Table 1 and Table 2? Did the authors use a smaller number of training epochs for the ablation studies?
>
> For Table 1, we use CFG training with a larger number of training iterations. In contrast, Table 2 reports results from regular training with fewer iterations. The detailed settings are provided in Table 3, where DiT-B/2-non-cfg is used for Table 2, while DiT-B/2, DiT-XL/2, and DiT-XL/2+ are used for Table 1.

---

> ### Comment · Reviewer_S4Q1 · 2025-11-25
>
> Thank you for the thorough reply! I am satisfied with the rebuttal, and have raise the score from 6 to 8.

---

> > ### Author Response · Authors · 2025-11-25
> >
> > Thank you for your effort and for your thoughtful, constructive review!

---

### Official Review · Reviewer_Tpep · 2025-10-30

**Soundness:** 4
**Presentation:** 3
**Contribution:** 4
**Rating:** 8
**Confidence:** 4

**Summary:**

This paper analyzes the MeanFlow framework for few-step generative modeling and introduces a generalized family of objectives called $\alpha$-Flow. The authors show that the MeanFlow loss decomposes into two competing components—trajectory flow matching and trajectory consistency—whose gradients are strongly negatively correlated, leading to slow convergence. To address this, they propose $\alpha$-Flow, which unifies MeanFlow, Shortcut Models, and flow matching under a single formulation. By gradually annealing the training objective from trajectory flow matching to MeanFlow (a curriculum learning approach), $\alpha$-Flow disentangles the conflicting objectives, improving optimization efficiency. When trained from scratch on class-conditional ImageNet-1K with DiT backbones, $\alpha$-Flow achieves consistent gains over MeanFlow, reaching new state-of-the-art FID scores for 1- and 2-step generation using standard architectures.

**Strengths:**

- The paper introduces a comprehensive framework to train few-step generative models by unifying and streamlining existing approaches. This, combined with the experimental ablations, provides a clear recipe for practitioners.

- The experimental section is complete, and provides a lot of insights.

- Section C (Failed experiments) includes approaches tried by the authors that did not succeed. This is a great idea that should be more common.

**Weaknesses:**

- The authors fail to cite some related works like https://arxiv.org/abs/2406.07507

- The fact that the gradients of L_{TFM} and L_{TC_c} are so negatively correlated is not surprising since adding the two losses and completing the square yields a constant value at the optimal vector field. Meanwhile, no such relationship exists between L_{FM’} and L_{TC_c}. Hence, it is reasonable that the correlation between the gradients of these two losses fluctuates around zero. From this perspective, there is nothing mysterious about the empirical analysis in Section 2. In fact, the negative correlation must hold between the gradients of any decomposition of a least squares loss.

- In Section 4.1, it would be good to devote a paragraph to clarifying/proving how the MeanFlow loss is related to the \alpha-Flow loss. The connection is stated in Theorem 1, but this statement is not obvious at a first glance. While proven in the appendix, having a sketch in the main text would help the reader.

**Questions:**

The authors state that "the 75% ratio yields worse NFE=1 but better NFE=2 generation results compared to the 50%-ratio version”. Extrapolating this observation, I wonder whether the optimal ratio grows with the target inference NFE. Have the authors checked this? More generally, have the authors run evaluations on larger NFE?

---

> ### Author Response · Authors · 2025-11-23
>
> > W1. Related work discussion.
>
> Thank you for pointing out the missing reference [1]. The work in [1] introduces Flow Matching Mapping (FMM), a unified framework that extends CMs, CTM, and progressive distillation. In particular, it demonstrates that existing methods can be interpreted within a common Euler and progressive perspective, while also proposing a novel class of Lagrangian methods. Our work, on the other hand, provides a distinct and detailed unification focused specifically on the Euler method. Within this framework, we unify Flow Matching, Shortcut models, and MeanFlow. We would be happy to incorporate any additional discussions or suggestions you may have. These discussions have been integrated into Appendix A.
>
>
> [1] Boffi, Nicholas Matthew, Michael Samuel Albergo, and Eric Vanden-Eijnden. "Flow map matching with stochastic interpolants: A mathematical framework for consistency models." Transactions on Machine Learning Research (2025).
>
> > W2. The gradients of $\mathcal L_{\mathtt{TFM}}$ and $\mathcal L_{\mathtt{TC}}$ are so negatively correlated is not surprising.
>
> First, we totally agree with you that, in general, for the minimizer $x^{\star}$ of some arbitrary, regular function $f(x) = g(x) + h(x)$, one can trivially show the relation $\nabla_x g(x^{\star}) = - \nabla_x h(x^{\star})$ from the necessary condition $\nabla_x f(x^{\star}) = 0$.
>
> However, our empirical analysis in Section 2 shows this gradient relationship from the early start of the optimization process, where the gradients of $\mathcal L_{\mathtt{TFM}}$ and $\mathcal{L}_{\mathtt{TC}}$ are already strongly negatively correlated (specifically, $<-0.9$). Crucially, this correlation becomes weaker (around $-0.7$) as training progresses.
>
> This behavior is opposite to what would be expected if the negative correlation simply resulted from convergence to a stationary point. Instead, the decreasing correlation indicates that the two loss components exert conflicting optimization signals early in training, and gradually become more aligned as the model parameters adapt.
>
> This suggests that the observed gradient dissimilarity reflects a different underlying mechanism than the trivial relation implied by $\nabla_x f(x^*) = 0$. This mechanism could be better explained by how far apart the corresponding minimizers $\arg\min g(x)$ and $\arg\min h(x)$ are. This evolution likely reflects the interaction between two loss components whose local minima are initially far apart: early in training, their gradients point toward distinct optima, leading to strong opposition, whereas as the model parameters evolve and the shared representation (and with the optimizer’s preconditioning reweighting the directions) begins to satisfy both objectives, the corresponding descent directions become partially aligned.
>
> We have added a clarification regarding this in the updated version of the paper in Appendix H.9.
>
>
> > W3. Devote a paragraph in the main paper to clarifying/proving how the MeanFlow loss is related to the $\alpha$-Flow loss.
>
>  We appreciate your writing suggestion and include the following paragraph after Theorem 1 in Section 4.1, which clarifying the relationship between MeanFlow and $\alpha$-Flow Loss.
>
>  Basically, the connection between $\mathcal L_{\alpha}$ and $\mathcal L_{\mathtt{TFM}}$ and $\mathcal L_{\mathtt{SC}}$ is fairly straightforward. For the non-trivial relationship showing the convergence from $\nabla_{\boldsymbol \theta}\mathcal L_{\alpha}(\theta)$ to $\nabla_{\boldsymbol \theta}\mathcal L_{\mathtt{MF}}(\theta)$, we leverage a first-order Taylor expansion on the term $\boldsymbol u_{\boldsymbol \theta^{-}}(\boldsymbol z_s, r, s)$ around $s =t$. This yields:
>  $$\boldsymbol u_{\boldsymbol \theta^{-}}\left(\boldsymbol z_s, r, s\right) = \boldsymbol u_{\boldsymbol \theta^{-}}\left(\boldsymbol z_t, r, t\right) - \dfrac{\text{d} \boldsymbol u_{\boldsymbol \theta^{-}}(\boldsymbol z_t, r, t)}{\text{d} t} (t - r) \alpha + \mathcal{O}\left(\alpha^2\right),$$
>  Substituting this expansion into the Equation 8 and taking the limit as $\alpha \rightarrow 0$ causes the higher-order terms $\mathcal{O}\left(\alpha^2\right)$ to vanish and recover $\nabla_{\boldsymbol \theta}\mathcal L_{\mathtt{MF}}(\boldsymbol \theta)$.

---

> > ### Author Response · Authors · 2025-11-23
> >
> > > Q1. Does the optimal ratio grow with the target inference NFE?
> >
> > We are grateful for your ablation suggestion. We indeed found that the optimal sampling ratio increases as the inference Number of Function Evaluations (NFE) grows.
> >
> > |  %r=t  |  Model  |  NFE  5  FID  |  NFE  5  FDD  |
> > |------|---------------|-----------|-----------|
> > |  0%  |  MeanFlow  |  44.0  |  843.8  |
> > |  0%  |  $\alpha$-FLow  |  38.2  |  779.0  |
> > |  25%  |  MeanFlow  |  41.3  |  817.8  |
> > |  25%  |  $\alpha$-FLow  |  36.9  |  770.0  |
> > |  50%  |  MeanFlow  |  38.4  |  783.3  |
> > |  50%  |  $\alpha$-FLow  |  35.5  |  743.5  |
> > |  75%  |  MeanFlow  |  36.0  |  752.1  |
> > |  75%  |  $\alpha$-FLow  |  **33.7**  |  **716.1**  |
> >
> > As shown in the table, the optimal ratio of $r=t$ required to achieve the best generation quality for NFE 5 is also 75%. We hypothesize that this occurs because Flow Matching inherently possesses a certain degree of few-step generation capability. Consequently, when mixing the training of Flow Matching and $\alpha$-Flow, varying the ratio $r=t$ introduces a trade-off between single-step and few-step generation. We have integrated the corresponding experimental results and discussion in Section H.3.

---

### Author Response · Authors · 2025-11-23
**Summary of rebuttal**

We thank all reviewers for their thoughtful and constructive feedback and are encouraged by the positive assessments of our work. In particular, the reviewers highlight the our papers are novel and significant (S4Q1), well-motived (9SDg), experiments are comprehensive (gT18, 9SDg) and insightful (Tpep, gT18), limitations and failure case discussion are detailed and valuable (Tpep, gT18). During the rebuttal period, we carefully address all questions raised by the reviewers and made corresponding revisions and additions to the manuscript. All changes are highlighted in blue. Below, we summarize our updates made during the rebuttal:

- We add Figure 2 to illustrate the mechanism explaining why $\alpha$-Flow outperforms MeanFlow.
- Between lines 297–306, we add a proof sketch describing the relationship between $\alpha$-Flow and MeanFlow.
- In Appendix A, we add detailed discussions comparing our work with Flow Matching Mapping (FMM) [1] and SplitMeanFlow [2].
- In Appendix E.1, we provide a theoretical justification of the solution spaces for the Trajectory Flow Matching (TFM) loss and the Trajectory Consistency (TC) loss.
- In Appendix E.2, we present a theoretical analysis of the optimality for MeanFlow and $\alpha$-Flow.
- In Appendix E.3, we provide a non-asymptotic analysis of the distance between gradients from $\alpha$-Flow and MeanFlow.
- In Appendix E.4, we analyze the variance of gradients in $\alpha$-Flow.
- In Appendix H.3, we include experimental results on the optimal ratio of $r = t$ under larger NFE settings.
- In Appendix H.4, we report fine-tuning results of $\alpha$-Flow on pretrained flow models.
- In Appendix H.5, we present distillation results of $\alpha$-Flow using pretrained flow models.
- In Appendix H.6, we extend our experiments to the Kinetics-700 video dataset to evaluate generalization over different dataset.
- In Appendix H.7, we include an empirical analysis of training stability comparing MeanFlow and $\alpha$-Flow.
- In Appendix H.8, we explore alternative guidance methods and their effects on performance.
- In Appendix H.9, we provide a detailed discussion of the gradient conflict between the TFM and TC losses.

We hope these additional theoretical and empirical results comprehensively address the reviewers’ main concerns. In the following, we respond to each reviewer’s specific questions individually.

[1] Boffi, Nicholas Matthew, Michael Samuel Albergo, and Eric Vanden-Eijnden. "Flow map matching with stochastic interpolants: A mathematical framework for consistency models." Transactions on Machine Learning Research (2025).

[2] Guo, Yi, Wei Wang, Zhihang Yuan, Rong Cao, Kuan Chen, Zhengyang Chen, Yuanyuan Huo et al. "Splitmeanflow: Interval splitting consistency in few-step generative modeling." arXiv preprint arXiv:2507.16884 (2025).

---

### Author Response · Authors · 2025-12-04
**Response to AC and all reviewers**

Dear Area Chair and Reviewers,

We sincerely appreciate the time and effort you devoted to reviewing our manuscript and helping us to strengthen it. And it is with regret that we see some fruitful interaction between us getting reset, especially with some of the reviewers revising their evaluations days before the public announcement of the technical issues with the platform. In what follows, we summarize our paper, reviewers’ feedback and the updates made in the revised submission.

Our work proposes $\alpha$-Flow, a unified and theoretically grounded framework for one and few-steps diffusion models trained from scratch. It mitigates the optimization challenges in MeanFlow and establishes new SOTA results. Reviewers commended the work’s novelty, significance, and clear motivation, emphasizing the comprehensive, insightful, and systematic nature of our experiments, along with the transparent discussion of limitations and failed-experiment analyses. They also highlighted the value of the unified $\alpha$-Flow perspective and its practical curriculum strategy. Reviewers additionally offered several constructive suggestions, both theoretical and empirical. And we addressed them in detail during the rebuttal.

 Our detailed reviewer-by-reviewer responses remain fully available in the discussion; none of them were lost during the platform reset. To address the reviewer's questions, we incorporated the following new materials into the revised paper:
- 4 new theoretical analysis (Appendix E.1, E.2, E.3, E.4), strengthening our claims about solution space, optimality, gradient behavior, and the $\alpha$-annealing trajectory;
- 5 new experiments (Appendix H.3, H.4, H.5, H.6, H.7, H.8), covering optimal ratio of $r=t$ under larger NFE settings, fine-tuning, distillation, evaluation on the Kinetics-700 video dataset, training stability analysis, ablation study over guidance methods.
- Figure 2, illustrating the mechanism of why $\alpha$-Flow outperforms MeanFlow.
- 2 new discussions (Appendix A,  Appendix H.9) with more related works and gradient conflict between the $\mathcal L_{\mathtt{TFM}}$ and $\mathcal L_{\mathtt{TC}}$ losses.

All revisions are highlighted in blue in the updated manuscript. We also encourage you to consult the full reviewer-by-reviewer discussions, where our complete responses pages address every point in detail. Together, these additions address all reviewer feedback and consolidate the contributions presented in the paper.

---

### Meta-Review · Area_Chair_17yA · 2025-12-25

**Summary:**

This paper formulates a one parameter loss family that interpolates between mean flow and flow matching (in fact the gradients are interpolated) and uses curriculum learning (schedule of this one parameter during training) to improve mean flow learning.
Overall, aside from one reviewer, all reviewers were positive about this paper.

+Comprehensive framework gracefully combining previous losses.
+Practical algorithm, complete experimental section.
-Initial claims about missing baselines such as CT (addressed in rebuttal).
-One reviewer saw the loss as incremental over FM and MeanFlow.

Overall one reviewer expressed an intent to raise their score from 6 to 8. This paper seems to address reviewers' concerns and above the bar for NeurIPS.

**Reviewer Concerns:**

Please see above.

**Reviewer Scores:**

Please see above.

---

### Decision · Program_Chairs · 2026-01-26

Accept (Poster)